# KPZ fluctuations in finite volume

**Sylvain Prolhac[1]⋆**

**1** Laboratoire de Physique Théorique, UPS, Université de Toulouse, France

⋆ sylvain.prolhac@irsamc.ups-tlse.fr

## Abstract

These lecture notes, adapted from the habilitation thesis of the author, survey in a first part various exact results obtained in the past few decades about KPZ fluctuations in one dimension, with a special focus on finite volume effects describing the relaxation to its stationary state of a finite system starting from a given initial condition. The second part is more specifically devoted to an approach allowing to express in a simple way the statistics of the current in the totally asymmetric simple exclusion process in terms of a contour integral on a compact Riemann surface, whose infinite genus limit leads to KPZ fluctuations in finite volume.

# 1  Introduction

Macroscopic observables of physical systems with a large number of constituents in thermal equilibrium generally exhibit Gaussian fluctuations as a consequence of the central limit theorem, when correlations are short ranged. This is one of the most basic examples of universality, leading to fluctuations whose statistics are independent not only of the precise description at the microscopic scale, but also of the specific physical setting considered. Long range correlations extending over the whole system lead to other universality classes, described by random fields characterized by their joint statistics at multiple points. At equilibrium, this generally requires fine tuning parameters to a critical point. Out of equilibrium, long range correlations may however be generated gradually by the dynamics itself. A prominent example is KPZ universality, initially introduced in the context of interface growth, but which appears also in a variety of other settings.

In one-dimension, KPZ universality is characterized by a random height field $h_\lambda(x, t)$ depending on position $x$ and time $t$, which can for instance be viewed as a solution of the KPZ equation $\partial_t h_\lambda = \frac{1}{2}\partial_x^2 h_\lambda - \lambda(\partial_x h_\lambda)^2 + \xi$ with white noise $\xi$. The real number $\lambda$ quantifies the non-linearity and parametrizes the renormalization group flow, between a fixed point $\lambda \to 0$ describing an interface whose dynamics is reversible in time, and the KPZ fixed point $\lambda \to \infty$ where the interface is driven far from equilibrium. A large number of exact results characterizing the fluctuations of $h_\lambda(x, t)$ have been obtained in the past thirty years, by exploiting a small number of exactly solvable models which are known to exhibit KPZ universality at large

scales.

A notable exactly solvable model displaying KPZ universality at large scales is the asymmetric simple exclusion process (ASEP), a Markov process featuring hard-core particles moving on a lattice. The technically simpler totally asymmetric version (TASEP), where all the particles move in the same direction, is in particular known to describe the KPZ fixed point $\lambda \to \infty$ at large scales. Being closely related to a quantum spin chain, the generator of ASEP can be diagonalized exactly using Bethe ansatz in terms of momenta solution of algebraic equations of high degree.

Two distinct regimes for KPZ fluctuations have received much attention in the past decades: stationary large deviations reached at late times for large but finite systems, which describe how unlikely atypical events are, and typical fluctuations for infinitely large systems, which exhibit scale invariance and retain the memory of the initial condition. In both regimes, the universal KPZ fluctuations are found to be non-Gaussian. The crossover between the two regimes, where KPZ fluctuations in finite volume (i.e. for a system of large but finite length $L$) relax to their stationary state, is the main focus of these notes. At the KPZ fixed point $\lambda \to \infty$, this crossover is associated with the presence of a time scale $t \sim L^{3/2}$, and describes how the correlation length and the amplitude of typical height fluctuations, which grow initially as $t^{2/3}$ and $t^{1/3}$ respectively at short time $t$, eventually saturate to $L$ and $\sqrt{L}$ respectively at late times.

The main picture that has emerged in the past few years for KPZ fluctuations in finite volume is that the analytic structure of large deviations at late times, which can be conveniently understood in terms of a Riemann surface of infinite genus, essentially contains the whole dynamics of the relaxation process. The KPZ fixed point in finite volume with periodic boundaries is the most well understood, at least for simple initial conditions, but recent progress also happened for the more physically relevant case of open boundaries. The full dynamics of KPZ fluctuations in finite volume with arbitrary $\lambda$ is however still open.

In section 2, we give a survey of KPZ fluctuations in one dimension. After a general introduction, we turn to exactly solvable models of KPZ universality, which have been the main drive for exact results about KPZ fluctuations. Exact results for the infinite system are then discussed, both at the KPZ fixed point $\lambda \to \infty$ and for finite $\lambda$. KPZ fluctuations in finite volume are finally reviewed. While we focus mainly on the KPZ fixed point with periodic boundaries, known results for the KPZ fixed point with open boundaries and KPZ fluctuations with finite $\lambda$ are also discussed, starting with stationary large deviations and their late time corrections. Spectral gaps controlling the relaxation to the stationary state of the generating function of the height are then introduced. The full time evolution of the probability of the height interpolating between the infinite system behaviour and stationarity large deviations is finally discussed.

Section 3, which is more technical, is devoted to a recent Riemann surface approach for KPZ fluctuations in finite volume. In a first part, the appearance of Riemann surfaces is explained in the elementary setting of integer counting processes, where the statistics of a quantity $Q_t$ incremented at the transitions of an underlying Markov process is described in terms of the Riemann surface associated with the complex algebraic curve $\det(M(g) - \mu \mathrm{Id}) = 0$, where the parameter-dependent operator $M(g)$ is the generator for $Q_t$, and $\log g$ is conjugate to $Q_t$. This formalism is then applied in a second part to TASEP with periodic boundaries, through a Riemann surface interpretation of its Bethe ansatz solution. Finally, the last part of section 3 outlines progress and difficulties for possible extensions to the KPZ fixed point with open boundaries or with several conserved fields in interaction, and also to the more difficult issue of KPZ fluctuations in finite volume with a finite parameter $\lambda$.

## 2   KPZ fluctuations in one dimension

In this section, we review several aspects of KPZ universality in one dimension, with a focus on exact results, both in the infinite system setting, where the correlation length grows forever, and for systems with a finite volume, where saturation occurs in the long time limit.

### 2.1   KPZ universality

KPZ universality, from Kardar, Parisi and Zhang [1], was introduced more than thirty years ago as a common denomination for a class of systems featuring an interface growing between a stable and a metastable domain. In the one-dimensional case considered here, the interface is a continuous path separating two bi-dimensional domains. Locally, the interface may be represented as a height field $h(x, t)$ depending on space and time. Assuming random growth perpendicular to the interface and expanding to second order in the slope $\partial_x h$ leads to the KPZ equation $\partial_t h = \partial_x^2 h + (\partial_x h)^2 + \xi$ with $\xi$ a noise term, that has given its name to the whole universality class. Higher order non-linear terms in the slope $\partial_x h$ are irrelevant in the renormalization group sense. The fluctuations of the random field $h(x, t)$ are the central object of KPZ universality.

#### 2.1.1   Various settings for KPZ fluctuations

While KPZ fluctuations were first introduced in the context of interface growth [2–6], their universal character goes far beyond: a large number of one-dimensional systems with local interactions and enough randomness and non-linearity fall in fact within KPZ universality. A few are listed below. Experimental observations are discussed instead in section 2.1.4, and exactly solvable models in section 2.2.

A much studied example is a class of driven particle systems [7,8] for which KPZ universality manifests itself in current fluctuations in the appropriate reference frame, see section 2.2.1 for more details on the specific case of the asymmetric exclusion process. Equivalently, density fluctuations $\sigma(x, t)$ obey the stochastic Burgers' equation $\partial_t \sigma = \partial_x^2 \sigma + 2\sigma \partial_x \sigma + \partial_x \xi$ at large scales, which is simply the spatial derivative of the KPZ equation with the identification of $\sigma$ as the slope $\partial_x h$ of the KPZ height. The stochastic Burgers' equation has also been studied in the context of randomly stirred incompressible fluids [9] and as a simplified model for turbulence [10], with $\sigma(x, t)$ interpreted as the velocity field of the fluid.

More generally, one-dimensional fluids with few conservation laws are known to exhibit anomalous fluctuations, with correlations spreading non-diffusively, see e.g. [11]. Non-linear fluctuating hydrodynamics [12–14] leads to coupled Burgers' equations and predicts the emergence of propagating sound modes exhibiting KPZ fluctuations. In the typical setting with three conserved quantities (say density of particles, energy and momentum), one finds generically a single heat mode with dynamical exponent $z$ (see the next section) equal to $5/3$ and two counter propagating KPZ sound modes with $z = 3/2$ [15–17]. KPZ sound modes have also been found in one-dimensional quantum fluids [18–21]. Additionally, the appearance of KPZ fluctuations at high temperature in integrable spin chains with infinitely many conservation laws has received a lot of attention recently [22,23], see also [24,25] for a discussion of the situation when integrability is broken.

Another setting exhibiting KPZ universality is the directed polymer in a random environment [3,26–28], describing the equilibrium fluctuations of a directed random path between two points far from each other, with a potential energy equal to the curvilinear integral of the quenched disorder along the path. The precise mapping to KPZ, see [29] for a discrete version and section 2.2.4 for the continuum version, indicates that the free energy of the directed polymer (i.e. the logarithm of its partition function) has the same statistics as the KPZ height

$h(x, t)$, with $x$ the lateral position of the endpoint of the random path and $t$ the end value for the parametrization of the path. The length of the longest increasing subsequence of a random permutation [30–32] is known to have the same statistics, under an interpretation of the permutation in terms of random points in two dimensions representing the random environment in which the directed polymer lives. KPZ fluctuations are also found in the related context of particle diffusion in a time-dependent random environment [33,34], where the probability that the particle is at position $x$ at time $t$ is interpreted as the partition function of the polymer.

KPZ fluctuations are more generally expected to appear for non-directed paths between two points far away from each other in a random environment, see [35] for a proof in a specific case. In two-dimensional disordered systems, an expansion in terms of non-directed random paths with signed Boltzmann weights also leads to KPZ fluctuations for the logarithm of the conductance between two points far away from each other [36–38] and for the logarithm of the probability density with specific initial conditions [39]. Additionally, random paths are related to random geometry by interpreting optimal paths as geodesics with respect to an appropriate metric, see e.g. [40]. In particular, large balls on a globally flat two-dimensional manifold with random metric have the same fluctuations seen from the Euclidean space as the KPZ height for the growth of a droplet [41], while fluctuations compatible with the growth of a flat interface are seen on the cylinder [42].

KPZ fluctuations have also been observed for quantum systems subject to a classical noise. For a specific type of random unitary dynamics, it was shown that the entanglement entropy $S(x, t)$ at time $t$ for a bi-partition splitting the system at position $x$ exhibit the same fluctuations as the KPZ height $h(x, t)$ [43–45]. Mappings were in particular found to the directed polymer in a random environment, an interface growth model and driven particles. A similar phenomenon was also found for a quantum conformal field theory coupled to noise [46]. The presence of classical noise can be interpreted as a continuous monitoring to which the quantum system is subjected, see e.g. [47–49] for a discussion of KPZ fluctuations in that context.

### 2.1.2 Universal exponents and scaling functions

Systems featuring a fluctuating interface generally exhibit distinct correlation lengths along and perpendicular to the interface increasing with time as power laws. The dynamical length scale $\ell_\parallel$ along the interface grows with time as [1] $\ell_\parallel \sim t^{1/z}$, where $z$ is called the dynamical exponent, while the typical amplitude for the fluctuations perpendicular to the interface grows as $\ell_\perp \sim (\ell_\parallel)^\alpha \sim t^{\alpha/z}$ with $\alpha$ the roughness exponent. If the total length $L$ of the interface is large but finite, this leads to the Family-Vicsek scaling $w \simeq L^\alpha f(t/L^z)$ [50] for the width $w$ of the interface (standard deviation of the height with respect to the position along the interface, averaged over noise). The width grows initially as $w \sim t^{\alpha/z}$ independently of the total length $L$ as long as the dynamical length scale $\ell_\parallel \ll L$, which implies $f(u) \sim u^{\alpha/z}$ for $u \ll 1$. Saturation $\ell_\parallel \simeq L$ at late times when the stationary state is reached leads on the other hand to $w \sim L^\alpha$, which corresponds to $f(u) \sim 1$ for $u \gg 1$. Other estimators for $\ell_\perp$, such as height difference correlations at a fixed time, also follow the Family-Vicsek scaling, albeit with a different function $f(u)$, whose behaviour at small and large $u$ is the same as above.

For a one-dimensional interface with reversible dynamics, the height is Brownian at equilibrium, which implies $\alpha = 1/2$ for the roughness exponent. Additionally, correlations spread diffusively along the interface, $\ell_\parallel \sim t^{1/2}$, which corresponds to the dynamical exponent $z = 2$. The amplitude of fluctuations perpendicular to the interface then grows as $\ell_\perp \sim t^{1/4}$ before saturation. In the interacting particle picture with single file motion (i.e. particles do not cross each other) and reversible dynamics, this corresponds for a tagged particle surrounded

---

[1]The notation $a \sim b$ always means that $a/b$ converges to a finite non-zero value in the appropriate limit, while $a \simeq b$ is used either when $a/b$ converges to $1$ or for perturbative expansions in some parameter, and $a \asymp b$ is used as a shorthand for $\log a \simeq \log b$. The notation $\approx$ is reserved for numerical values.

on both sides by a finite density of particles to a sub-diffusive spreading with displacement proportional to $t^{1/4}$ [51, 52].

It turns out that the non-equilibrium stationary state of a one-dimensional interface with irreversible dynamics described by KPZ universality is still Brownian, see section 2.1.6, so that the roughness exponent is again $\alpha = 1/2$ in the KPZ case. Additionally, at the KPZ fixed point (see section 2.1.3 below), Galilean invariance implies the scaling relation $z + \alpha = 2$, and thus the dynamical exponent $z = 3/2$. Correlations thus spread super-diffusively as $\ell_{\parallel} \sim t^{2/3}$ along the interface, but sub-diffusively as $\ell_{\perp} \sim t^{1/3}$ perpendicularly to it.

Beyond the dynamical and roughness exponents, KPZ fluctuations are characterized by universal scaling functions, for instance the probability density (either at a single point $(x, t)$ or jointly at several points) of the height field $h(x, t)$. From the point of view of exact expressions, generating functions such as $\langle e^{sh(x,t)} \rangle$ are sometimes more convenient. Additionally, in some asymptotic regimes, universal large deviation functions describe the probability of rare events.

The universal scaling functions generally depend on the initial condition for the height field. Additionally, since correlations extend to the whole system at late times, the scaling functions necessarily also depend on boundary conditions for the space variable $x$. The most studied ones are the periodic boundary condition for its simplicity, and open boundaries with slope $\partial_x h$ fixed to $\sigma_a$ and $\sigma_b$ at the ends. In the latter case, subtle issues arise about what it means exactly to fix the slope, which is ill defined, see section 2.1.5. The natural choice in the driven particle picture, which we adopt in the following, is to connect the system to reservoirs of particles in such a way that the slope is fixed on average to $\sigma_a$ and $\sigma_b$ at the boundaries, see section 2.2.2.

Many exact results have been obtained in the past decades for the statistics of the height field in an infinite system, for which the statistics of the height is known quite generally, especially at the KPZ fixed point, see section 2.3. For the finite volume case, which is our main focus here, progress beyond early results for the relaxation rate and stationary large deviations is more recent and many open questions remain, see section 2.4.

### 2.1.3 KPZ universality and KPZ fixed point

In this section, we discuss a bit more precisely what is meant by KPZ universality in one dimension. We emphasize that KPZ universality actually refers to two distinct objects, that are both believed to be universal: the KPZ fixed point, characterized in particular by the exponents $z = 3/2$ and $\alpha = 1/2$, and the KPZ equation, which can be understood as the renormalization group flow between the KPZ fixed point and a fixed point describing an interface with reversible dynamics.

With general units of height, space and time, the one-dimensional KPZ equation with unit Gaussian white noise $\xi$, such that $\langle \xi(x,t) \rangle = 0$ and $\langle \xi(x,t)\xi(x',t') \rangle = \delta(x - x')\delta(t - t')$, reads $\partial_t h = \nu \partial_x^2 h + \lambda(\partial_x h)^2 + \sqrt{D}\,\xi$, where the height $h(x, t)$ is defined for times $t \geq 0$ and in the interval $x \in [0, \ell]$ with $\ell$ the system size. Additional parameters may encode boundary conditions. Using the scaling properties of $\xi$ and writing $h_{\nu,\lambda,D,\ell}$ for the solution of the KPZ equation above, one has the equality in law $h_{a\nu,b\lambda,cD,\ell}(x,t) = \frac{a}{b} h_{\nu,\lambda,D,\ell/\kappa_x}(x/\kappa_x, t/\kappa_t)$ for arbitrary $a, b, c$, with $\kappa_x = \frac{a^3}{b^2 c}$ and $\kappa_t = \frac{a^5}{b^4 c^2}$, under a corresponding rescaling of the initial condition. The parameters $\nu$, $D$ and $\ell$ may then be set to arbitrary values under rescaling. We choose units of height, space and time such that the KPZ equation reads

$$\partial_t h_\lambda = \tfrac{1}{2}\partial_x^2 h_\lambda - \lambda(\partial_x h_\lambda)^2 + \xi \quad \text{with } x \in [0, 1]\,. \tag{1}$$

This choice of parameters is used everywhere in the following for the random field $h_\lambda(x, t)$ with finite system size. We restrict furthermore to a coefficient $\lambda > 0$, which fixes the direction of the growth of the interface toward negative values of the height, a choice that is standard

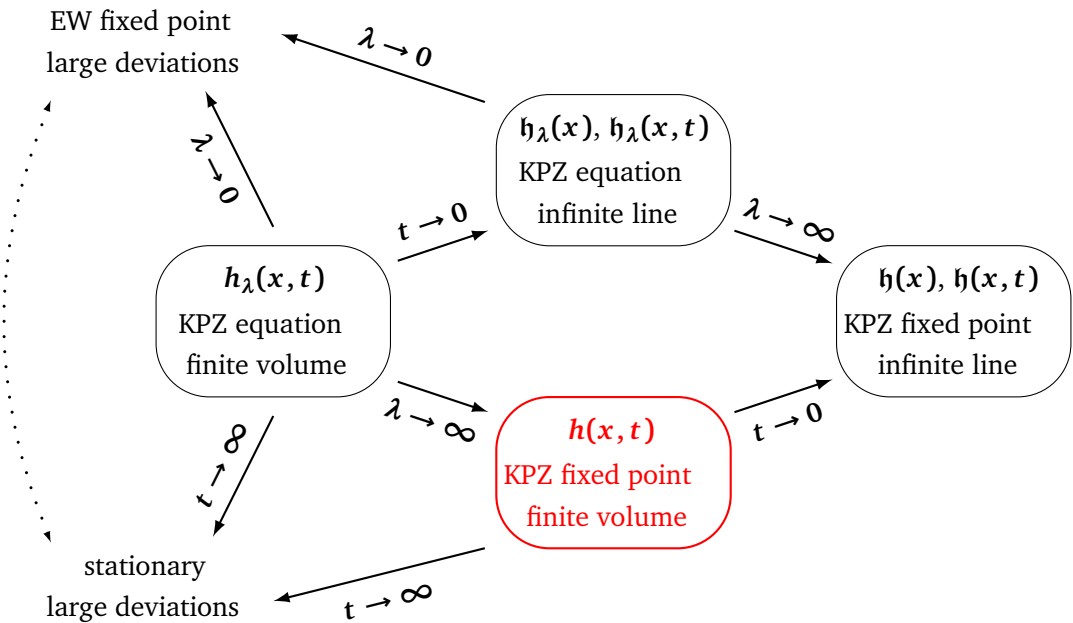

Figure 1: Special limits of interest for the KPZ height $h_\lambda(x,t)$. Appropriate rescalings of the parameters are needed in each case, see the text. The statistics of each height field depends on the initial condition, and additionally on boundary conditions in finite volume.

for the asymmetric exclusion process, a discrete model in KPZ universality, see section 2.2.1 below.

Since a typical solution of the KPZ equation is very rough, the non-linear term $(\partial_x h)^2$ in (1) is not well defined. Various approaches have been considered to make sense of (1), see section 2.1.5. The main outcome is that while it is indeed possible to give a meaning to the solution $h_\lambda(x,t)$ of (1) with some regular enough initial condition $h_\lambda(x,0)$, such a solution is only defined up to a shift by $ct$ with $c$ an arbitrary number independent of $x$ and $t$. For definiteness, we choose here the natural notion of solution defined from the asymmetric exclusion process in (6).

At a given position $x$, the statistics of $h_\lambda(x,t)$ with fixed initial and boundary conditions depends only on the parameter $\lambda$ and on time $t$. We discuss below various limits of interest, see also figure 1. Among those, the KPZ fixed point in finite volume, obtained when $\lambda \to \infty$ is our main focus in the following.

The variable $\lambda$ in front of the non-linear term in (1) characterizes the degree of irreversibility of the dynamics, with $\lambda = 0$ corresponding to the Edwards-Wilkinson (EW) equation [53] describing a reversible dynamics, while $\lambda \to \infty$ corresponds to a strongly irreversible dynamics. Additionally, observing the system at higher scales (under diffusive scaling of space and time) corresponds to increasing $\lambda$ in (1), and one may in fact view $\lambda$ as parametrizing the renormalization group flow from the unstable EW fixed point $\lambda \to 0$, where the interface satisfies the Family-Vicsek scaling with dynamical exponent $z = 2$, to the stable KPZ fixed point $\lambda \to \infty$ with dynamical exponent $z = 3/2$. The EW fixed point, which is not exactly the EW equation but has to be considered as a limit $\lambda \to 0$ of $h_\lambda(x,t)$ after rescaling, see the end of sections 2.3.2 and 2.4.1, has Gaussian typical fluctuations and non-trivial large deviations. On the other hand, the KPZ fixed point $\lambda \to \infty$ defined in (2) below has non-Gaussian fluctuations.

Time, on the other hand describes how the fluctuations of the height $h_\lambda$ (or rather its slope $\partial_x h_\lambda$, see section 2.1.6) relax to their stationary state in the long time limit, where typical fluctuations are Gaussian and non-trivial large deviations describe rare events, see section 2.4.1. At short time, when the dynamical length scale is much smaller than the system size, boundaries are not felt and $h_\lambda(x, t)$ has instead the same statistics as for the infinite system $x \in \mathbb{R}$. The relaxation rate in finite volume thus introduces a time scale in the problem, associated with a crossover between the infinite system and stationary fluctuations.

We consider the solution $h_\lambda(x, t)$ of (1) for some regular enough initial condition independent of $\lambda$ at $t = 0$. Then, the quantity $h_\lambda(x, t/\lambda)$ has a finite limit [2] when $\lambda \to \infty$, and we write

$$h(x, t) = \lim_{\lambda \to \infty} h_\lambda(x, t/\lambda),\qquad(2)$$

which we call in the following the KPZ fixed point in finite volume. From the discussion above, the short time limit of $h(x, t)$ away from boundaries is expected to be the KPZ fixed point on the infinite line. More precisely, the KPZ exponents indicate that the fluctuations of $t^{-1/3} h(t^{2/3}x, t)$ have a finite limit at short time, and one writes

$$\mathfrak{h}(x) = \lim_{t \to 0} t^{-1/3}\big(h(t^{2/3}x, t) - h(0, 0)\big)\qquad(3)$$

for the KPZ fixed point on an infinitely large system (the infinite line $x \in \mathbb{R}$ or the half-line $x \in \mathbb{R}^+$ respectively for $h(x, t)$ corresponding to periodic or open boundaries). For convenience, we generally impose a global shift to the initial condition so that $h(0, 0) = 0$ in the following. While the limit (3) probes the region for $x$ close to $x_0 = 0$, one can more generally take instead the short time limit of $t^{-1/3}(h(x_0 + t^{2/3}x, t) - h(x_0, 0))$ to access the region close to arbitrary $x_0$, and the limit generally depends on $x_0$. Additionally, the limit in (3) does not depend on time anymore, and it is sometimes useful to consider instead $\mathfrak{h}(x, t) = \lim_{\epsilon \to 0} \epsilon^{-1/3}(h(\epsilon^{2/3}x, \epsilon t) - h(0, 0))$. While for a single value of time, $\mathfrak{h}(x, t)$ and $t^{1/3}\mathfrak{h}(x/t^{2/3})$ have the same law under appropriate rescaling of the initial condition, non-trivial correlations in time are expected for distinct values of $t$.

The KPZ fixed point on the infinite line can also be obtained from the solution $h_\lambda(x, t)$ of the KPZ equation in finite volume by exchanging the order of the limits $\lambda \to \infty$ and $t \to 0$, see figure 1. Taking the short time limit first, the quantity $\delta^{-1/4}(h_{\delta^{-1/4}\lambda}(\delta^{1/2}x, \delta t) - h_{\delta^{-1/4}\lambda}(0, 0))$ is equivalent under rescaling to the KPZ height at position $x$ and time $t$ with parameters as in (1) but with system size $\delta^{-1/2}$, and converges when $\delta \to 0$ to a quantity $\mathfrak{h}_\lambda(x, t)$ which is the solution of (1) on the infinite line $x \in \mathbb{R}$, with an initial condition obtained from $h_\lambda(x, 0)$ in the region $x \sim \delta^{1/2}$ (one could also consider a region around arbitrary $x_0$ instead). As before, while the dependency on time (or on $\lambda$) can be essentially removed by rescaling when considering the statistics of the height at a single time, joint statistics at multiple times require both parameters. Finally, $\mathfrak{h}_\lambda(x, t)$ converges again [2] for large $\lambda$ to the KPZ fixed point on the infinite line as $\mathfrak{h}(x, t) = \lim_{\lambda \to \infty} \lambda^{-1/3}\mathfrak{h}_\lambda(\lambda^{2/3}x, t)$. Equivalently, setting $\epsilon = \lambda^{-4}$, one has $\mathfrak{h}(x, t) = \lim_{\epsilon \to 0} \epsilon^{1/3}\mathfrak{h}_1(\epsilon^{-2/3}x, t/\epsilon)$ under rescaling of the KPZ equation, which is the celebrated 1-2-3 scaling at the KPZ fixed point for the height, position and time.

Many exact results are available for the statistics of the KPZ height in one dimension. On the infinite line, see section 2.3, the complete joint statistics at multiple positions and times of the KPZ fixed point is known for specific initial conditions, and variational formulas exist for spatial correlations with general initial condition. Fewer results are available for the KPZ equation on the infinite line. In finite volume, see section 2.4, the KPZ fixed point with periodic boundaries is the most well studied case, although some results have also been obtained about

---

[2]For the notion of solution considered here and defined from the asymmetric exclusion process in section 2.2.1. For other notions of solution, subtracting a divergent deterministic term proportional to $t$ may be required, see section 2.1.5.

stationary large deviations for the KPZ equation with periodic boundaries and for the KPZ fixed point with open boundaries.

### 2.1.4 Experimental observations

The universal character of KPZ fluctuations makes them an appealing subject for experimental studies. So far, experiments have focused on the KPZ fixed point $\lambda \to \infty$. In the context of interface growth, this has the advantage that fine-tuning the rate at which the interface grows is not needed and the system is simply driven far from equilibrium. Additionally, experiments have so far essentially considered the early time regime, where the correlation length is much smaller that the system size, corresponding to KPZ fluctuations on the infinite line, which eliminates any issue about boundary conditions.

The values of the exponents of the KPZ fixed point have been observed in experiments on growing colonies of bacteria [54] and cells [55, 56], combustion fronts in paper [57–59], interface growth between turbulent phases in a liquid crystal [60–63], deposition at the edge of evaporating drops in colloidal suspensions [64, 65], reaction fronts driven through a porous medium [66], see [67] for a detailed survey of those experiments. A common issue has been that the presence of strong enough quenched disorder in the medium may lead to a distinct universality class, where the noise in (1) depends on $x$ and $h(x, t)$ rather than $x$ and $t$. This was circumvented for combustion fronts by soaking the paper beforehand in a solution and for reaction fronts by driving the system with sufficient velocity through the porous medium. Turbulence suppresses the issue altogether in the liquid crystal experiment.

The KPZ exponents have recently been observed in quantum systems as well. In [68], the low energy spectrum of an antiferromagnet whose Hamiltonian is well approximated by a Heisenberg spin chain was probed using neutron scattering, finding the expected scaling as the momentum to the power $3/2$. In [69], superdiffusive spin transport consistent with the KPZ prediction was observed in a system of cold atoms trapped in an optical lattice and mimicking a spin chain. In [70], the KPZ exponents were observed in the coherence decay for a one-dimensional polariton condensate driven out of equilibrium in a semiconductor optical microcavity, see also [71] for a discussion of finite volume effects that should be accessible in this type of systems.

In some experiments, universal scaling functions characterizing KPZ fluctuations have been observed. In the liquid crystal experiment, very clean results were obtained for the probability distribution of the height for droplet growth [60, 62], as well as flat [62], stationary [63] and more general [72, 73] initial conditions. Correlations in time have also been measured [74]. All these experimental results are in full agreement with the theoretical predictions for the KPZ fixed point on the infinite line, summarized in section 2.3.1 below. Experimental results for the distribution of the height were also obtained earlier for burning fronts [59] and aggregation of particles in a two-dimensional fluid [75], while the two-point correlation of the slope of the height was measured for the polariton condensate [70].

### 2.1.5 Mathematical issues for the definition of the KPZ equation

The roughness exponent $\alpha = 1/2$ indicates that a typical realization of the interface corresponds for the KPZ height $h_\lambda(x, t)$ to a continuous but nowhere differentiable path in the variable $x$ at any time $t > 0$, see for instance the next section for explicit representations at late times in terms of conditioned Brownian motions. The KPZ equation is then in principle ill-defined from a mathematical point of view because of the non-linear term. Rigorous ways to make sense of it have however been developed in the past decade, see e.g. [76] for a review of the various approaches.

The KPZ equation (1) can be mapped, at least formally, to the stochastic heat equation with multiplicative noise

$$\partial_t Z_\lambda = \frac{1}{2}\partial_x^2 Z_\lambda - 2\lambda \, \xi Z_\lambda \tag{4}$$

by the Cole-Hopf transformation $Z_\lambda(x,t) = e^{-2\lambda h_\lambda(x,t)}$. The solution of (4) depends on the discretization scheme chosen for the time variable. The standard choice of the Itô prescription for the discretization leads to what is usually called the Cole-Hopf solution of (1) after taking the logarithm (the solution of (4) is known to be positive with probability one for positive initial condition [77]). Mathematical proofs [78–82] have confirmed that for several specific microscopic particle models featuring a discrete Cole-Hopf structure, the corresponding discrete height converges at large scales to the Cole-Hopf solution of the KPZ equation after properly shifting and rescaling the height by model dependent quantities, and under weakly asymmetric scaling between hopping rates in both directions, see section 2.2.1 for the case of the asymmetric exclusion process.

Another approach considers instead the stochastic Burgers' equation

$$\partial_t \sigma_\lambda = \frac{1}{2}\partial_x^2 \sigma_\lambda - \lambda \partial_x \sigma_\lambda^2 + \partial_x \xi \,, \tag{5}$$

formally obtained by taking the spatial derivative of (1) and setting $\sigma_\lambda = \partial_x h_\lambda$. When the initial condition is the stationary state, a martingale approach defining a notion of energy solutions of (5) with well behaved $\partial_x \sigma_\lambda^2$ was introduced [83,84], and their uniqueness was proved subsequently in [85], both on the infinite line and with periodic boundaries. The energy and Cole-Hopf solutions match up to a shift of $\lambda^3 t/3$ for $h_\lambda(x,t)$. Large scale limits of a large class of weakly asymmetric microscopic particle models [86–91] have been shown to lead to the energy solution of (5). Energy solutions have recently been extended to non-stationary initial condition [92].

The KPZ equation can also be defined in the general framework of regularity structures for non-linear stochastic partial differential equations [93–96] with a finite system size, see also [97,98] for an alternative approach in terms of paracontrolled distributions, which allows for an extension to the infinite system [99], and [100,101] for a renormalization group approach. The regularity structures approach has led to rigorous proofs [102–104] that higher order non-linear terms and non-Gaussianities of the noise are indeed irrelevant in the renormalization group sense for the KPZ equation, see also [92,105] with energy solutions. In the regularity structures approach, proper solutions of (1) are defined for arbitrary initial condition by subtracting divergent counter-terms to solutions with regularized noise. In the case of (1) with periodic boundaries [93], the Gaussian white noise $\xi(x,t)$ is regularized with respect to the space variable $x$ and replaced by $\xi_\epsilon(x,t)$ with smooth spatial autocorrelation, which converges to $\xi(x,t)$ when $\epsilon \to 0$. Calling $h_\lambda^\epsilon(x,t)$ the corresponding solution of (1), the quantity $h_\lambda^\epsilon(x,t) - c_\epsilon \lambda t$ with $c_\epsilon$ a coefficient depending on the choice of autocorrelation for $\xi_\epsilon$ and diverging when $\epsilon \to 0$ then has a finite limit $h_\lambda(x,t)$, interpreted as the appropriate solution of (1).

An important point which follows is that solutions of the KPZ equation (1) are only defined in a meaningful way up to a shift $c_\lambda t$ with $c_\lambda$ arbitrary. On the other hand, it is useful in practice to consider a concrete solution $h_\lambda(x,t)$ corresponding to e.g. a specific choice for the asymptotic value of the average $\langle h_\lambda(x,t)\rangle/t$ when $t \to \infty$. We choose here to define $h_\lambda(x,t)$ from a weakly asymmetric exclusion process after minimal subtraction of divergent terms, see section 2.2.1 below. For periodic boundary condition, this solution coincides with the corresponding energy solution but is such that $h_\lambda(x,t) + \lambda^3 t/3$ is the Cole-Hopf solution, and verifies in particular $\langle h_\lambda(x,t)\rangle \simeq \lambda t$ in the long time limit. This choice has the advantage that the limit to the KPZ fixed point in (2) does not require subtracting a divergent term when $\lambda \to \infty$.

The situation is slightly more complicated for the KPZ equation on an open interval. There, one would like to impose Neumann boundary conditions i.e. fix the slopes $\partial_x h_\lambda(0, t) = \sigma_a$, $\partial_x h_\lambda(1, t) = \sigma_b$, which corresponds to the natural choice for driven particles connected to reservoirs, see section 2.2.2, but does not make so much sense since $h_\lambda$ is not differentiable. The height function for the weakly asymmetric exclusion process was however shown [106–108] to converge at large scales to the corresponding Cole-Hopf solution, with boundary conditions $\partial_x Z_\lambda(0, t) = -2\lambda\sigma_a Z_\lambda(0, t)$, $\partial_x Z_\lambda(1, t) = -2\lambda\sigma_b Z_\lambda(1, t)$. Alternative definitions through energy solutions [91] and regularity structures [109] were put forward. A subtle issue is that naive identification for the boundary slopes between different notions of solution fails. For instance, the solution from regularity structures obtained in the limit $\epsilon \to 0$ from a solution of (1) with regularized noise as above and $\partial_x h_\lambda^\epsilon(0, t) = \sigma_a$, $\partial_x h_\lambda^\epsilon(1, t) = \sigma_b$ corresponds to a tilted Cole-Hopf solution with boundary parameters depending on the choice of regularization for the noise and generally distinct from $\sigma_a$ and $\sigma_b$. In the following, we define again $h_\lambda(x, t)$ as the same minimal subtraction for the weakly asymmetric exclusion process and fix the boundary parameters from the slope of the average height, i.e. $\sigma_a = \partial_x\langle h_\lambda(0, t)\rangle$, $\sigma_b = \partial_x\langle h_\lambda(1, t)\rangle$, which is believed to be independent of time, and can be easily computed at late times from the known stationary state, see section 2.2.2. In field theory language, the values of $\partial_x\langle h_\lambda\rangle$ at $x = 0$ and $x = 1$ are the physical boundary slopes, as opposed to the bare values of the slope appearing as boundary conditions in the stochastic heat equation, from which they differ by a shift of $1/2$.

We emphasize that the proper notions of solution discussed above define a KPZ height $h_\lambda(x, t)$ with finite parameter $\lambda$. The height $h(x, t)$ defined from (2) at the KPZ fixed point $\lambda \to \infty$ is an even more singular object, for which the equation (1) makes even less sense.

### 2.1.6  Stationary state and special initial states

The KPZ height grows forever at late times as $h_\lambda(x, t) \sim t$. Subtracting the global velocity by considering instead $h_\lambda(x, t) - h_\lambda(0, t)$ (or equivalently the slope $\sigma_\lambda = \partial_x h_\lambda$ solution of Burgers' equation) leads to a unique stationary state [110–112] (characterized by its joint statistics at multiple positions $x$, called the invariant measure in the probability literature) $h_\lambda^{st}(x)$ with $h_\lambda^{st}(0) = 0$, such that when the system is prepared with initial condition $h_\lambda(x, 0) = h_\lambda^{st}(x)$, then $h_\lambda(x, t) - h_\lambda(0, t)$ has at any time $t$ the same (joint) statistics as $h_\lambda^{st}(x)$. Additionally, the statistics of $h_\lambda(x, t) - h_\lambda(0, t)$ converge when $t \to \infty$ to those of $h_\lambda^{st}(x)$ for arbitrary initial condition, both in finite volume [110] and also locally on the infinite line, within an interval of length going to infinity with time [113, 114].

For the system on the infinite line, the stationary state is Brownian, with the same statistics as a two-sided Wiener process $w(x)$ (more precisely $w(x) + cx$ with arbitrary drift $c$ [115]), independently of $\lambda$. This is still the case on the half-line, but with a drift $c$ fixed by the boundary slope [116]. The dependency of the stationary state on boundary conditions is more pronounced in finite volume. For periodic boundaries, $h_\lambda^{st}(x)$ is a standard Brownian bridge $b(x)$, i.e. a Wiener process conditioned on $b(0) = b(1) = 0$, with mean zero and variance $x(1-x)$. As on the infinite line, the stationary state $h_\lambda^{st}$ with periodic boundaries is independent of $\lambda$.

For open boundaries, the stationary state does depend on $\lambda$, and also on the boundary slopes $\sigma_a = \partial_x\langle h_\lambda(0, t)\rangle$ and $\sigma_b = \partial_x\langle h_\lambda(1, t)\rangle$. At the KPZ fixed point $\lambda \to \infty$ and with $\sigma_a = -\infty$, $\sigma_b = \infty$, one has $h_\infty^{st}(x) = -\frac{w(x) + e(x)}{\sqrt{2}}$ where $w$ is a Wiener process (with $w(1)$ and hence $h_\infty^{st}(1)$ typically non-zero, unlike in the periodic case) and $e$ an independent Brownian excursion (i.e. a standard Brownian bridge conditioned on staying positive) [117]. This leads in particular to the average height $\langle h_\infty^{st}(x)\rangle = -2\sqrt{x(1-x)/\pi}$, which does have the required infinite slopes at $x = 0$ and $x = 1$ (note however that this is only true for the averaged height: the random variable $h_\infty^{st}(x)/\sqrt{x}$ is positive with non-zero probability $\frac{1}{4} - \frac{1}{2\pi} \approx 0.09$

for small $x$ even though $\sigma_a = -\infty$). When $\sigma_a = -\infty$ and $\sigma_b = 0$, the Brownian excursion is replaced by a Brownian meander (i.e. a Wiener process conditioned on staying positive) [118], and the average height is then equal to $\langle h_\infty^{\text{st}}(x) \rangle = -(\sqrt{x(1-x)} + \arcsin\sqrt{x})/\sqrt{\pi}$, which again has the required slopes. When both slopes are equal to zero, the stationary state $h_\infty^{\text{st}}(x)$ is simply the Wiener process $w(x)$. The case of arbitrary finite slopes was obtained in [119, 120], see also [121] for a random path interpretation, by taking the limit $\lambda \to \infty$ of the recent solution for general $\lambda$ [119, 122–126].

Exact results for KPZ fluctuations are generally restricted to a few very special initial conditions $h_\lambda(x, 0) = h_0(x)$, for which computations are tractable, and which we choose for convenience with $h_0(0) = 0$ in the following. Among them is the stationary state $h_0(x) = h_\lambda^{\text{st}}(x)$ described above, which is random. Another one is the very simple, deterministic, flat initial condition $h_0(x) = 0$ for all $x$, for which the interface is initially a flat line, and which corresponds in the driven particle picture to a constant density profile.

The initial condition for which exact expressions are usually the simplest is however the narrow wedge (also called sharp wedge) initial condition, which is deterministic, and is formally equal to $h_0(x) = \frac{|x|}{0} - \infty$. Properly defining the KPZ height for this very singular initial condition requires some care, including generally the subtraction of an extra counter-term in some approaches discussed in section 2.1.5. At the level of the stochastic heat equation (4), the narrow wedge initial condition is simply given by $Z_\lambda(x, 0) = \delta(x)$. The non-linear term of the KPZ equation dominates at short time for the narrow wedge initial condition, which leads to the deterministic solution $h_\lambda(x, t) \simeq \frac{x^2}{4\lambda t}$, after a shift so that $h_\lambda(0, t) = 0$, see e.g. [127], and from (2) to $h(x, t) \simeq \frac{x^2}{4t}$ for small $t$ at the KPZ fixed point. Droplet growth, for which the interface is initially a small circle, which is locally parabolic, is thus described by the narrow wedge initial condition. With periodic boundaries, the discontinuity at $x = 1/2$ (modulo $1$) for $\partial_x h$ is interpreted for driven particles as a leftover from the single shock surviving at late times from Burgers' hydrodynamics, see section 2.2.1 for the asymmetric exclusion process.

## 2.2   Exactly solvable models

The highly singular nature of the KPZ equation and the KPZ fixed point has long required the use of discrete microscopic models to study KPZ universality. Numerical simulations were in particular performed for various models where a cluster grows by random addition of elementary blocks, for instance uniformly on the surface of the cluster (Eden models), by sticking to the cluster at the first point of contact after arriving from a fixed direction (ballistic deposition), or by adding constraints on the allowed height difference between neighbouring points at the surface of the cluster (restricted solid on solid models), see e.g. [3, 128–137].

We consider instead in this section a small number of exactly solvable models exhibiting KPZ fluctuations at large scales, from which many exact results have been obtained (see sections 2.3 and 2.4). Among them, we focus more particularly on the (totally) asymmetric exclusion process, from which a derivation of some exact results for KPZ fluctuations in finite volume is sketched in section 3 in terms of Riemann surfaces.

### 2.2.1   The asymmetric simple exclusion process

Markov processes describing interacting particles on a one dimensional lattice are characterized by time-dependent probabilities solution of a master equation, which is essentially identical to the Schrödinger equation with imaginary time of a quantum spin chain [138–140]. In special cases where the spin chain is integrable, various quantities can then be computed exactly, at least in principle, for the corresponding system of particles.

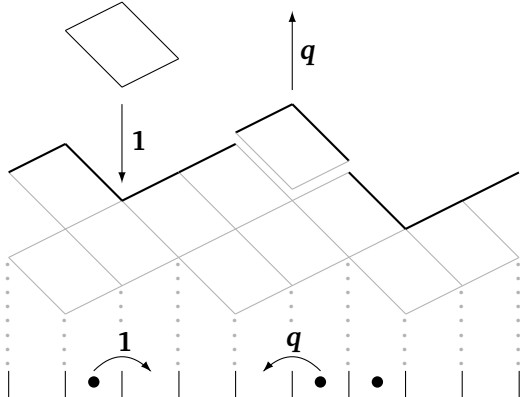

Figure 2: ASEP and the corresponding interface growth model. The totally asymmetric model TASEP corresponds to the case $q = 0$ where particles hop in the same direction. Up and down increments for the height are distinct to enforce periodic boundary condition.

A prominent example exhibiting KPZ fluctuations is the one-dimensional asymmetric simple exclusion process (ASEP) [141–147], introduced initially in [148,149], and used to model e.g. biological transport [150, 151] or traffic flow [152]. ASEP features hard-core particles hopping between neighbouring sites of a lattice (either infinite or with some boundary conditions) with distinct forward and backward hopping rates $1$ and $q < 1$, see figure 2. The Markov generator of ASEP is related by a similarity transformation to the Hamiltonian of an XXZ quantum spin chain with twisted boundary conditions, which is the prototype of Yang-Baxter integrable models solvable by the Bethe ansatz [153–158]. Section 3.2 presents in some details the Bethe ansatz solution with periodic boundary condition, in the technically simpler case of the totally asymmetric model $q = 0$ (TASEP), while open boundaries, variants with multiple species of particles and ASEP with $q \neq 0$ are discussed in section 3.3.

ASEP can be mapped to an interface growth model with a discrete height function $H_{i,t}$ at site $i$ and time $t$. The positions of the particles on the lattice completely fix the increments $H_{i+1,t} - H_{i,t}$, which are positive (respectively negative) when the site $i$ is empty (resp. occupied), see figure 2. The dynamics of ASEP induces local changes of the height by deposition and evaporation with respective rates $1$ and $q$ so that $Q_{i,t} = H_{i,t} - H_{i,0}$ is the time integrated current of particles between sites $i$ and $i + 1$, i.e. the number of particles that have moved from $i$ to $i + 1$ minus the number of particles that have moved from $i + 1$ to $i$ between time $0$ and time $t$. We consider in the rest of this section ASEP with periodic boundaries. Calling $L$ the number of lattice sites and $N$ the conserved number of particles, the height increments $H_{i+1,t} - H_{i,t}$ are then chosen equal to $N/L$ and $-(1 - N/L)$ to ensure the periodicity of the height function, see figure 2.

The large $L$ behaviour of ASEP with fixed $q < 1$ and density of particles $\overline{\rho} = N/L$ exhibits several regimes of interest depending on how the time $t$ scales with $L$. For finite times, the evolution depends precisely on the graph of allowed transitions between microstates, which has a periodic structure since the sum of the positions of the particles increases by one modulo $L$ each time a particle moves forward, leading to oscillations on the scale $t \sim L^0$ for typical initial conditions [159]. Long time scales are needed for a collective behaviour to emerge [160]. On the Euler scale $t \sim L$, the dynamics of the system is at leading order deterministic, and describes the hydrodynamic evolution of the density field $\rho(x, t)$, which becomes at late times $t/L \gg 1$ a travelling wave moving with velocity $(1 - q)(1 - 2\overline{\rho})$ and relaxing to the constant value $\overline{\rho}$. The last time scale of interest happens when the residual deterministic density profile has become of the same amplitude as the random fluctuations, which happens on the time scale

$t \sim L^{3/2}$, and KPZ fluctuations in finite volume are observed in the moving reference frame with velocity $(1-q)(1-2\overline{\rho})$.

Deterministic hydrodynamics is obtained for ASEP with fixed $q < 1$ and density of particles $\overline{\rho} = N/L$ when the the number of lattice sites $L$ goes to infinity on the Euler time scale $t \sim L$. Defining the rescaled time $\tau = (1-q)t/L$, the density of particles $\rho(x,\tau)$, which counts the number of particles per site around the position $i = xL$ on the lattice then follows Burgers' equation without noise and viscosity (the term with the second derivative with respect to space in (5)), $\partial_\tau \rho + \partial_x j = 0$ with current $j = \rho(1-\rho)$, see e.g. [161, 162].

Starting with any non-constant initial condition $\rho_0(x)$, the absence of viscosity leads after a finite time $\tau$ to the formation of discontinuities in $x$ for $\rho(x,\tau)$ called shocks, at which characteristics (lines where $\rho$ is constant in time) disappear. The dynamics of shocks (creation and merging) for finite $\tau$ may be quite intricate. For long times, however, their behaviour is controlled by divides [163], i.e. characteristics issued from global minima of the initial height function $\mathcal{H}_0(x) = \int_0^x \mathrm{d}y \, (\overline{\rho} - \rho_0(y))$. Divides never meet shocks and move at constant velocity $1 - 2\overline{\rho}$. At long time $\tau$, a single shock survives between two neighbouring divides, and the density profile between two neighbouring shocks decays as $\rho(x+(1-2\overline{\rho})\tau) \simeq \overline{\rho} - \frac{x-a}{2\tau}$. Beside shocks, anti-shocks, from which infinitely many characteristics are issued, are also solution of Burgers' equation in the absence of viscosity. However, they are only unphysical solutions (also called weak solutions) and do not describe the density $\rho(x,\tau)$ of ASEP. Indeed, anti-shocks are unstable: in the presence of a vanishingly small viscosity, they decay instantaneously into rarefaction fans where $\rho(x,\tau)$ is linear in $x$. With the convention $q < 1$, shocks (respectively anti-shocks) are identified as discontinuities at which the density profile $\rho$ increases (resp. decreases) with $x$.

On the Euler time scale, typical fluctuations beyond deterministic hydrodynamics have a correlation length of order $\ell_\parallel \sim t^{2/3} \sim L^{2/3}$, i.e. much smaller than the system size $L$, and fluctuations are thus described by the KPZ fixed point on the infinite line [164]. Longer time scales are needed to observe KPZ fluctuations correlated on the whole system. Additionally, rare events where a different density profile is observed for a finite duration $\tau$ lead to large deviations, for which unphysical weak solutions of Burgers' equation are selected [165–167], see also [168, 169] for an application to current fluctuations, and [170] for a study of even rarer events.

The KPZ fixed point in finite volume with periodic boundaries is obtained for ASEP with finite density $\rho = N/L$ and fixed $q < 1$ on the time scale $t \sim L^{3/2}$. For an initial condition with density profile $\rho_0(x)$, defining the rescaled time $\tau$ through $(1-q)t = \tau L^{3/2}/\sqrt{\rho(1-\rho)}$ and observing the system in the moving reference frame $i = (1-q)(1-2\rho)t + xL$ suggested by the long time behaviour of Burgers' hydrodynamics, the ASEP height function behaves for large $L$ as [3]

$$H_{i,t} \simeq (1-q)\rho(1-\rho)t + CL + \sqrt{\rho(1-\rho)L} \, h(x,\tau), \tag{6}$$

and the fraction of occupied sites around the position $x$ as $\rho - \sqrt{\rho(1-\rho)}\,\sigma(x,\tau)/\sqrt{L}$ with $\sigma = \partial_x h$. Here, $h(x,\tau)$ is the random height field from section 2.1.3 representing the KPZ fixed point with periodic boundary condition. For an initial condition with density profile of the form $\rho_0(x) \simeq \rho - \sqrt{\rho(1-\rho)}\,\sigma_0(x)/\sqrt{L}$, the extra counter-term $C$ in (6) is equal to zero and the initial condition for the KPZ height is $h(x,0) = \int_0^x \mathrm{d}x \, \sigma_0(x)$. For a finite density profile $\rho_0(x)$ on the other hand, one has instead $C = \min[\mathcal{H}_0]$ with $\mathcal{H}_0(x)$ as above, which corresponds for the KPZ height to the narrow wedge initial condition if $\mathcal{H}_0$ has a unique global minimum. In particular, we note that the decay $\overline{\rho} - \frac{x}{2\tau}$ of the solution of Burgers' equation at

---

[3]Although $H_{i,t}$ grows toward positive values, subtracting the deterministic term proportional to $t$ in (6) gives a KPZ height $h(x,\tau)$ growing toward negative values, in accordance with the minus sign in front of the non-linear term in the KPZ equation (1).

long time does match with the parabola $h(x, \tau) \simeq \frac{x^2}{4\tau}$ obtained at short time for the narrow wedge initial condition, see section 2.1.6.

The scaling $L^{3/2}$ for the relaxation time means that spectral gaps are of order $L^{-3/2}$, see sections 3.2.4 and 2.4.3. More generally, this scaling is also observed for the mixing time, which considers the convergence of the probabilities of all the configurations to their stationary values and not just the convergence of the statistics of the height [171, 172]. An intriguing synchronization behaviour for the trajectories of microstates [173] is also observed numerically on the same time scale.

Beside the KPZ fixed point, the KPZ equation with parameter $\lambda$ as in (1) can also be obtained from ASEP as in (6), under weakly asymmetric scaling (WASEP) [78, 127, 174],

$$1 - q \simeq \frac{2\lambda}{\sqrt{\rho(1-\rho)L}} \tag{7}$$

and on the time scale $t = \tau L^2 / 2$. This is in agreement with the definition (2) of the KPZ fixed point $h(x, \tau)$ from $h_\lambda(x, \tau)$. We emphasize however that the EW fixed point, i.e. the limit $\lambda \to 0$ of $h_\lambda(x, \tau)$, does not coincide with the symmetric exclusion process $q = 1$: there is indeed another [4] non-trivial crossover scale $1 - q \simeq \mu/L$, where the system is well described by a path integral approach as in section 2.2.5 and exhibits phase separation when conditioned on low current [175]. The EW fixed point is then obtained either by taking $\mu \to \infty$ or $\lambda \to 0$ depending on the weakly asymmetric scaling chosen for ASEP.

### 2.2.2 ASEP with open boundaries

The KPZ fixed point on an interval with fixed slopes $\partial_x h$ at the boundaries as in section 2.1.5 can be understood as a large scale limit of ASEP on a chain of $L$ sites with fixed $q < 1$ and particles entering and leaving the system at the boundaries, which models a system connected to reservoirs of particles. For simplicity, one can always restrict to TASEP $q = 0$ where particles enter the system at site $1$ (if it is empty) with rate $\alpha$ and leave the system from site $L$ with rate $\beta$, see figure 3.

At large $L$, a mean field domain wall picture [176–179] predicts a non-trivial phase diagram, see figure 3, confirmed by an exact solution [180, 181] for the stationary probabilities of TASEP, see also [182] for an interpretation in terms of Lee-Yang zeroes [183]. The low and high density phases (which are further split in two by a dynamical transition [184]) have respective average density of particles $\rho = \alpha$ and $\rho = 1 - \beta$, which are different from $1/2$, and the reference frame $i = (1 - 2\rho)t + xL$ does not make sense on the time scale $t \sim L^{3/2}$ in these two phases. The finite volume regime for KPZ fluctuations can then only be observed in the maximal current phase, for which the average density of particles is $\rho = 1/2$. There, (6) still holds, under the same scalings as before, with $h(x, \tau)$ again the KPZ fixed point, but now with open boundaries.

The corresponding boundary slopes $\sigma_a = \partial_x \langle h(x, t) \rangle_{|x=0}$ and $\sigma_b = \partial_x \langle h(x, t) \rangle_{|x=1}$ of the KPZ height field can be extracted from the known average density of particles at the boundaries in the stationary state computed in [181]. There, the average occupation of the first site (respectively last site) was found to be equal to $1 - \alpha^{-1}\kappa^{-1}$ (resp. $\beta^{-1}\kappa^{-1}$) with $\kappa = 4$, $\alpha^{-1}(1-\alpha)^{-1}$ and $\beta^{-1}(1-\beta)^{-1}$ respectively in the maximal current, low density and high density phase, up to a correction of order at most $L^{-1}$. It follows that the interior of the maximal current phase, $\alpha, \beta > 1/2$, corresponds to the KPZ fixed point with $\sigma_a = -\infty$, $\sigma_b = \infty$, while finite slopes correspond to a region close to the tripe point, with

$$\alpha \simeq \frac{1}{2} - \frac{\sigma_a}{2\sqrt{L}} \qquad \text{and} \qquad \beta \simeq \frac{1}{2} + \frac{\sigma_b}{2\sqrt{L}}. \tag{8}$$

---

[4]Confusingly, both scalings $1 - q \sim L^{-1/2}$ and $1 - q \sim L^{-1}$ are called WASEP in the literature.

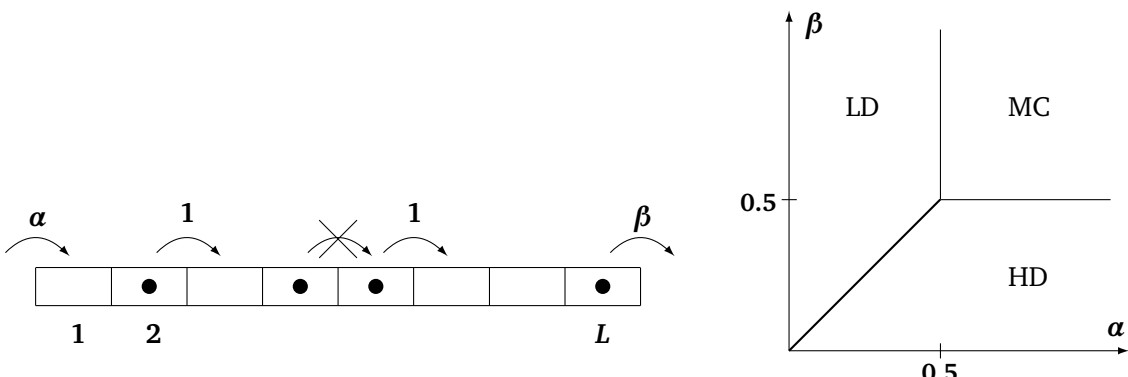

Figure 3: TASEP with open boundaries (left) and its phase diagram at large $L$ (right). The average density and current of particles are discontinuous across the transition between the low density (LD) and high density (HD) phases, while the transition to the maximal current (MC) phase is continuous. KPZ fluctuations with slope $\partial_x h$ equal to finite values at $x = 0$ and $x = 1$ are obtained in the vicinity (8) of the triple point, while the inside of the maximal current phase corresponds to infinite boundary slopes.

As in the periodic case, the KPZ equation with open boundaries is obtained for ASEP with hopping rates $1$ and $q$ under weakly asymmetric scaling (7) with $\rho = 1/2$ [106, 107]. Considering more general boundary conditions where particles can additionally exit the system at site $1$ with rate $\gamma$ and enter the system at site $L$ with rate $\delta$, the scalings $\alpha \simeq \frac{1}{2} - \frac{\sigma_a^- - \lambda}{\sqrt{L}}$, $\gamma \simeq \frac{1}{2} + \frac{\sigma_a^+ - \lambda}{\sqrt{L}}$, $\beta \simeq \frac{1}{2} + \frac{\sigma_b^+ + \lambda}{\sqrt{L}}$, $\delta \simeq \frac{1}{2} - \frac{\sigma_b^- + \lambda}{\sqrt{L}}$ lead to boundary slopes $\sigma_a = \sigma_a^+ + \sigma_a^-$ and $\sigma_b = \sigma_b^+ + \sigma_b^-$ for the KPZ height $h_\lambda$.

### 2.2.3 Other integrable models for KPZ universality

Several other integrable models are known to lead to KPZ fluctuations at large scales. In particular, a whole hierarchy of integrable extensions of ASEP exist, see e.g. [185], including models in discrete time [186], with long range hopping [187], inhomogeneities [188–190], multiple particles allowed per site [191–193] or several types of particles (see section 3.3.2). Non-trivial dualities exist between some of these models, see e.g. [194–198].

Other integrable models from KPZ universality include the polynuclear growth model [199], which features random deposition of portions of interface and deterministic lateral growth, Brownian motions with contact interactions [200–202], which can be obtained from models in discrete space after a suitable continuum limit, non-intersecting Brownian paths [203–206], whose top line has KPZ statistics, vertex models [207–211], whose lattice path representation can be interpreted as trajectories of particles, random tiling models [212–214], where KPZ fluctuations are observed at the boundary of the frozen region, and various directed polymer models [215–218]. It should be noted that the classification above into various classes of models is somewhat arbitrary, since a given model may be interpreted in several ways thanks to the existence of various mappings.

On the quantum side, integrable models such as the Heisenberg or the XXZ quantum spin chains are known from numerical studies [219] to exhibit KPZ fluctuations. Despite the fact that their Hamiltonian is related by a similarity transformation to the Markov generator of ASEP, KPZ fluctuations have not been confirmed so far from exact calculations for quantum integrable models with interactions (see however [220–222], where scaling functions for free fermions matching with the ones appearing for KPZ fluctuations on the infinite line are obtained). Several additional difficulties are indeed present in the quantum setting. First and

foremost, while the convergence of Markov processes to their stationary state is well understood, with contributions of higher eigenstates decaying exponentially fast with time in the presence of a finite spectral gap, see section 2.4.3, the contribution of each separate eigenstate lives forever under the unitary evolution of closed quantum system. This lack of ergodicity in finite quantum systems means in particular that special initial states must be avoided, unlike in the Markov setting, especially for quantum integrable models, which have infinitely many conserved quantities. Additionally, the fact that averages are quadratic in the wave function solution of the Schrödinger equation and not simply linear as in the Markov case adds another layer of complications.

### 2.2.4 The replica solution of the KPZ equation

Another exactly solvable model for KPZ fluctuations, defined directly in the continuum, is the KPZ equation itself, through a mapping [223, 224] to a quantum integrable Bose gas obtained by the replica method, which we explain briefly below.

The stochastic heat equation (4) can be solved formally by the Feynman-Kac formula as the path integral $Z_\lambda(x, t) = \int_{}^{x(t)=x} \mathcal{D}x \, \exp\left(-\int_0^t d\tau \left(\frac{1}{2}(\partial_\tau x(\tau))^2 + 2\lambda \xi(x(\tau), \tau)\right)\right)$, which can be interpreted as the partition function of a directed path $x(\tau)$ in a random medium. The integration measure for the initial position $x(0)$ of the path is determined from the initial condition $Z_\lambda(x, 0)$, with for instance $x(0)$ set equal to $0$ for narrow wedge initial condition, and integrated uniformly on space for flat initial condition. Introducing $n$ replicas $x_j(\tau)$ of the path, the average over the noise $\xi$ of the product $Z_\lambda(x_1, t) \ldots Z_\lambda(x_n, t)$ can be computed, after subtracting a divergent term corresponding to the interaction of a replica with itself. Using again the Feynman-Kac formula, the average then solves the Schrödinger equation in imaginary time $-\partial_t \langle Z_\lambda(x_1, t) \ldots Z_\lambda(x_n, t) \rangle = H_n \langle Z_\lambda(x_1, t) \ldots Z_\lambda(x_n, t) \rangle$. Here,

$$H_{n,\lambda} = -\frac{1}{2} \sum_{j=1}^{n} \partial_x^2 - 4\lambda^2 \sum_{i<j} \delta(x_i - x_j) \tag{9}$$

is the Hamiltonian of the Lieb-Liniger delta-Bose gas [225] with attractive interaction, which is known to be integrable by Bethe ansatz.

Taking all the $x_j$ equal to $x$, the generating function of the KPZ height for e.g. narrow wedge initial condition is then equal to $\langle e^{-2n\lambda h_\lambda(x,t)} \rangle = \langle x, \ldots, x | e^{-tH_{n,\lambda}} | 0, \ldots, 0 \rangle$. In principle, the knowledge of this generating function for all $n \in \mathbb{N}$ does not allow to determine the statistics of $h_\lambda(x, t)$ in a unique way. In particular, extracting the first cumulants of the height would require the limit $n \to 0$ and thus treating $n$ as a continuous variable. Nevertheless, it can be hoped that an expression analytic in the variable $n$ for the generating function could still be valid for non-integer $n$, especially if that expression was obtained in a "natural way", which is known as the replica trick in the context of systems with quenched disorder. The method has been successfully used for KPZ fluctuations with periodic boundaries at late times [226, 227], where only the smallest eigenvalue of $H_{n,\lambda}$ contributes, and on the infinite line [228–230], after an expansion of the generating function over Bethe eigenstates. The results found are in perfect agreement with those obtained from ASEP. Additionally, duality relations for discrete models (including ASEP) on the infinite lattice leads to expressions that have a natural interpretation as a discretization of those obtained from the replica method, thus providing a mathematical justification of their validity [231, 232].

### 2.2.5 Path integral approach

General correlation functions of the KPZ height field $h_\lambda(x, t)$ can formally be written as a functional integral over the realizations of the Gaussian white noise $\xi(x, t)$ in the KPZ equation

(1). A functional integral over $h_\lambda(x, t)$ can then be added, at the price of introducing Dirac distributions to enforce the KPZ equation. Expressing those Dirac distributions in terms of integrals over a response field then allows to perform explicitly the integral over the noise, leading to a functional integral over the height and response fields only, with an explicit action which depends on the correlation function studied.

In the limit where the amplitude of the noise is small (or equivalently the limit $\lambda \to 0$ to the EW fixed point with our conventions in (1)), the functional integral leads after a saddle point to a system of partial differential equations for the height and response fields maximizing the action, which can be rewritten as two coupled non-linear heat equations after some transformations [233], see also [234] for a rigorous mathematical approach from the Freidlin-Wentzell large deviation principle for the stochastic heat equation (4).

It turns out that the system of coupled heat equations mentioned above is actually integrable [5], which has allowed to compute exactly large deviations of the height at the EW fixed point [235, 236], see at the end of section 2.3.2. A similar system of integrable coupled heat equations also appears in the framework of the macroscopic fluctuation theory [237] for several models of diffusive particles [238, 239], as well as in a crossover between the EW fixed point and a diffusive regime [240].

Interestingly, the same system of coupled heat equations as at the EW fixed point (but in the variables position and height instead of time and position) also appears (in relation with the Kadomtsev–Petviashvili equation, see at the end of section 2.3.1) for the exact one-point probability of the KPZ fixed point $\lambda \to \infty$ with narrow wedge initial condition [241], both on the infinite line and with periodic boundaries. It is not clear whether this is just a coincidence or a hint to a possible extension of the path integral approach beyond the saddle point approximation of the weak noise regime, at least at the KPZ fixed point.

## 2.3 Exact results for the infinite system

Most exact results for KPZ fluctuations in the past twenty years have been obtained for the system on the infinite line, see the general surveys [242–245], and also reviews with a focus on numerical approaches [246], random tilings and line ensembles [247, 248], connections to random matrices [249, 250] and random geometry [40], exact one-point statistics for the KPZ equation with narrow wedge initial condition [251–253], joint statistics at the KPZ fixed point [254–256], mathematically rigorous approaches for the stochastic heat equation [257], or integrable aspects [258–260].

### 2.3.1 KPZ fixed point

The KPZ fixed point on the infinite line $x \in \mathbb{R}$ is described by a random field $\mathfrak{h}(x, t)$, see figure 1, which is known to be Markovian in time [261]. The statistics of $\mathfrak{h}(x, t)$ for which exact results have been obtained can be divided into three types, by increasing degree of generality (and of difficulty for exact calculations): one-point statistics, which characterize the probability density of $\mathfrak{h}(x, t)$ for given $x$ and $t$, multiple-point statistics, which consider joint distributions at various positions $x_j$ but the same time $t$, and finally joint statistics at multiple times and positions, which gives the most general characterization of the random field $\mathfrak{h}(x, t)$, and are in particular relevant for the issue of aging of KPZ fluctuations [262–264]. As explained in section 2.1.3, the time variable can be eliminated by rescaling in the first two cases, when $\mathfrak{h}(x, t)$ is only observed at a single time $t$, and one can consider instead $\mathfrak{h}(x) = \mathfrak{h}(x, 1)$ without loss of generality.

---

[5]The *classical integrability* of this system of coupled heat equations and the *quantum integrability* of the Lieb-Liniger delta-Bose gas and of ASEP, see sections 3.2.1 and 3.3.3, are quite distinct notions.

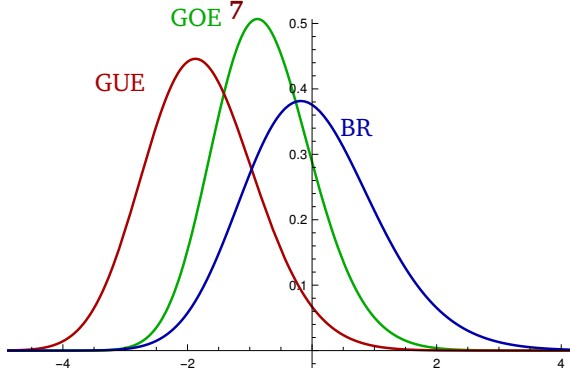

| | GUE | GOE [7] | BR |
|---|---|---|---|
| $c_1$ | -1.771087 | -0.760069 | 0 |
| $c_2$ | 0.813195 | 0.638048 | 1.150394 |
| $c_3$ | 0.164325 | 0.149567 | 0.443468 |
| $c_4$ | 0.061796 | 0.067271 | 0.382673 |

Figure 4: Probability density functions (left) and first cumulants $c_k$ (right) of the GUE and GOE Tracy-Widom distributions (up to a rescaling by a factor $2^{2/3}$ in the GOE case compared with the usual definition from random matrix theory) and the Baik-Rains distribution with parameter $x = 0$, according to which (minus) the height is distributed respectively for a curved, flat or Brownian interface at the KPZ fixed point. The distributions have distinct left and right tails, with respective exponents $3$ and $3/2$ for the logarithm of their probability density function, reflecting the fact that the interface grows in a specific direction.

We begin with the one-point statistics, which depends on the initial condition for the height $\mathfrak{h}_0(x) = \mathfrak{h}(x, 0)$, or equivalently, under rescaling, on the large scale behaviour of $\mathfrak{h}(x)$, see section 2.1.3. The narrow wedge, flat and stationary initial conditions discussed in section 2.1.6 have been identified as domains of attraction under rescaling for interface growth processes [199] and in particular for the KPZ equation [265], corresponding respectively to a curved, flat or stationary interface. The corresponding probability density of $\mathfrak{h}(x)$ have been obtained exactly.

In the narrow wedge case, which describes in particular droplet growth, works on various exactly solvable microscopic models have shown [266–268] that the distribution of $x^2/4 - \mathfrak{h}(x)$, which is independent of $x$, is the GUE Tracy-Widom distribution [6]. That distribution was initially introduced in [269] in the context of random matrix theory for the fluctuations of the largest eigenvalue of a large complex valued Hermitian random matrix distributed according to the Gaussian unitary ensemble (GUE). In the context of driven particles, the GUE Tracy-Widom distribution thus describes the fluctuations of the current [267] for some initial conditions, but interestingly also controls the convergence of the time-dependent probabilities of microstates to their stationary values in a finite system [270, 271].

For flat initial condition, one finds instead [272–274] that $-\mathfrak{h}(x)$, whose statistics is translation invariant in $x$, has the same GOE Tracy-Widom distribution [7] [275] as the largest eigenvalue of large real valued symmetric random matrices distributed according to the Gaussian orthogonal ensemble (GOE). Both GUE and GOE Tracy-Widom distributions have exact expressions as Fredholm determinants with a kernel involving the Airy function, or can alternatively be written in terms of the Hastings-McLeod solution of the Painlevé II equation.

For stationary initial condition [276, 277], $\mathfrak{h}_0(x)$ is a two-sided Wiener process, with $\mathfrak{h}_0(0) = 0$, and $-\mathfrak{h}(x)$ is distributed according to the Baik-Rains distribution, which depends on the position $x$ because of the fluctuations of the initial condition away from $x = 0$. The Baik-Rains distribution has a similar expression as the GUE and GOE Tracy-Widom distribu-

---

[6]The minus sign in front of $\mathfrak{h}(x)$ comes from the minus sign in front of the non-linear term in (1), while the factor $1/4$ for the parabola is a consequence of our choice of parameters in (1).

[7]With the standard convention in random matrix theory, it is rather $-2^{2/3}\mathfrak{h}(x)$ that has GOE Tracy-Widom distribution. The distribution plotted in figure 4 is that of $-\mathfrak{h}(x)$ without further rescaling.

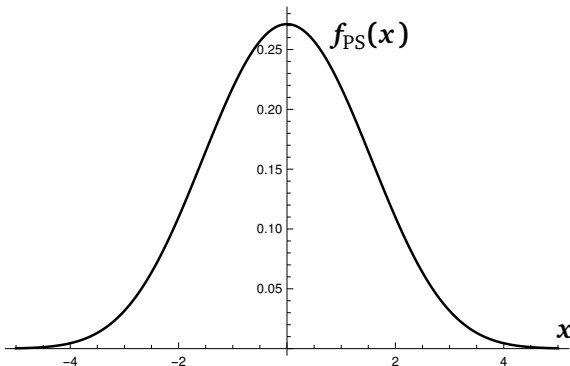

Figure 5: Prähofer-Spohn scaling function $f_{\mathrm{PS}}(x)$, see the text. While $f_{\mathrm{PS}}(x)$ looks like a Gaussian for small values of $x$, the logarithm of $f_{\mathrm{PS}}(x)$ actually has cubic tails.

tions, see figure 4 for plots of the three distributions.

Hybrid cases combining two of the initial conditions above (one on the left side $x < 0$ and the other on the right side $x > 0$) have also been studied, and lead to crossovers between the narrow wedge, flat and stationary initial conditions when varying $x$ between $-\infty$ and $+\infty$, see e.g. [243]. Some results are also known on the half-line [278, 279].

As mentioned already at the beginning of this section, the one-point statistics discussed above do not fully characterize the KPZ fixed point: correlations between different points are also needed. An important example in the driven particles setting (and also for Burgers' equation and non-linear fluctuating hydrodynamics) is the Prähofer-Spohn scaling function [242, 277, 280], equal to the covariance of the slope $\langle \partial_x \mathfrak{h}(0,0) \partial_x \mathfrak{h}(x,1) \rangle$ with stationary initial condition, and plotted in figure 5. Using stationarity [242], it reduces to $f_{\mathrm{PS}}(x) = \frac{1}{2} \partial_x^2 \langle \mathfrak{h}(x)^2 \rangle$, which involves a single time and can be computed from the $x$-dependent Baik-Rains distribution. The function $f_{\mathrm{PS}}$ can also be interpreted in the directed polymer setting as the probability density for the midpoint position of the random path [281].

Exact expressions for joint statistics of $\mathfrak{h}(x)$ at an arbitrary number of positions $x_1, \dots, x_n$ have also been obtained for specific initial conditions. For the narrow wedge case, $x^2/4 - \mathfrak{h}(x)$ has the same joint statistics as the Airy process [282, 283] (also called Airy$_2$ process in the context of random matrix theory, where it is obtained as a scaling limit of the largest eigenvalue of Dyson's matrix Brownian motion for GUE [284]), which is stationary in $x$. The Airy process is also the top line of the Airy line ensemble, a collection of infinitely many non-intersecting random paths obtained as a scaling limit of non-intersecting Brownian motions, with the nice resampling property that the statistics of each path between two points is simply a Brownian bridge conditioned not to intersect the paths above and below [285]. Multiple-point statistics of the Airy process can be expressed as Fredholm determinants with a matrix valued kernel involving again the Airy function. Other processes with explicit Fredholm determinant expressions for their joint statistics are known for flat [273, 286], stationary [287] and various hybrid initial conditions, see e.g. [243].

More generally, the linearity of the stochastic heat equation (4) combined with the fact that the narrow wedge initial condition is a Dirac delta for $Z_\lambda(x,0)$ suggests a variational formula for general initial condition, expressing $\mathfrak{h}(x)$ in terms of the Airy process [288]. An early instance of such a variational formula for the distribution of $\mathfrak{h}(x)$ with flat initial condition is the fact that the maximum of the Airy process minus a parabola has GOE Tracy-Widom distribution [289], see also [290–292] for other formulas of the same type. This was later generalized to arbitrary initial condition for the one-point distribution [293] and multiple point statistics [294], see also [295] where the same variational formula appears in the different context of particles constrained by a moving wall. Joint statistics of $\mathfrak{h}(x)$ for general initial

condition can again be expressed as a Fredholm determinant, with a kernel written in terms of the probability that a Brownian motion hits the initial condition $\mathfrak{h}_0(x)$ [294, 296, 297].

Joint statistics at multiple times $t_1, \ldots, t_n$ of $\mathfrak{h}(x, t)$ have also been investigated. Exact results were obtained for narrow wedge initial condition with $n = 2$ [298–300] and general $n$ [301, 302]. For $n = 2$, various limits have also been investigated when the two times are close or far from each other, either based on an exact solution for narrow wedge initial condition [303–306], or using a variational approach [307, 308]. Additionally, non-trivial invariance properties relating various joint statistics have also been discovered [309].

Surprising connections between KPZ fluctuations and (deterministic) non-linear integrable partial differential equations have been found recently, more specifically with the Kadomtsev–Petviashvili (KP) equation $3\partial_y^2 u = \partial_x(4\partial_t u - 6u\partial_x u - \partial_x^3 u)$, see e.g. [310]. In [311], it was shown that for general initial condition, $2\partial_x^2 \log \mathbb{P}(\mathfrak{h}(2y, 3t) > x)$ is a solution of the KP equation (in the language of integrable partial differential equations, the probability is then called a tau function for KP). For flat initial condition, the probability does not depend on the space variable, and the probability is instead a tau function for the Korteweg-De Vries (KdV) equation $4\partial_t u = 6u\partial_x u + \partial_x^3 u$, corresponding to a quite singular initial condition for $u(x, t)$. For multiple point statistics at a single time, a matrix valued KP equation appears [311]. A discrete version involving instead the two-dimensional Toda lattice was found for the polynuclear growth model [312]. Similar results have been obtained for joint statistics at multiple times for the KPZ fixed point [313].

### 2.3.2 KPZ equation

The full evolution in time of the KPZ height following the KPZ equation, which can also be seen as the crossover between the EW and KPZ fixed points, also leads to universal distributions. Exact results for one-point statistics are available for a few initial conditions.

For narrow wedge initial condition, the distribution of $\mathfrak{h}_\lambda(x, t)$ at a given position $x$ and time $t$ has been obtained both from ASEP [314, 315] and from the replica solution of the KPZ equation [229, 230], see [316] for numerical evaluations and plots. This distribution is expressed either as the integral of a Fredholm determinant or in terms of a non-local integro-differential generalization of the Painlevé II equation. It turns out that this distribution also appears for extremal positions of trapped fermions at finite temperature [222]. An interpretation in terms of random matrices was also found in [317].

Exact results for flat [318] and stationary [319, 320] initial conditions have also been obtained, as well as for various hybrid initial conditions, see e.g. [321]. The KPZ equation on the half-line $x \in \mathbb{R}^+$ with specific boundary condition at $x = 0$ has also been considered [116, 322]. As for the KPZ fixed point, the KP equation appears for one-point statistics of the KPZ equation [311], although so far only for a few cases including narrow wedge and stationary initial condition.

Joint statistics at multiple points and the same time have also been studied for the KPZ equation. For narrow wedge initial condition, exact expressions obtained from the replica method have been shown to converge to those for the Airy process when $\lambda \to \infty$, both for two-point [323–325] and an arbitrary number of points [326, 327]. Manageable expressions for joint statistics with arbitrary $\lambda$ are however still missing. General proofs that solutions of the KPZ equation converge to the KPZ fixed point are known [328–330].

The exact results discussed above concern typical fluctuations of the KPZ height. Rare events corresponding to a height much larger or smaller than typical have an exponentially small probability, and non-trivial, universal large deviation functions may appear. Large deviations have been studied for the KPZ equation both at late and short times. At late times, the

logarithm of the probability that $\mathfrak{h}_\lambda(x, t)$ equals $jt$ grows as $|j|^{3/2}$ for $j < 0$ [8], while for $j > 0$, one obtains a non-trivial large deviation function interpolating between $j^3$ for small $j$ and $j^{5/2}$ for large $j$, see [331–334] for the narrow wedge initial condition. While the exponents $3/2$ and $3$ match the tail behaviour of the Tracy-Widom distributions at the KPZ fixed point on the infinite line, the exponent $5/2$ also appears for large deviations at the KPZ fixed point in finite volume, see section 2.4.1.

At short time (or equivalently $\lambda \to 0$, i.e. close to the EW fixed point), on the other hand, the height at time $t$ typically grows as $t^{1/4}$ for fixed $\lambda$, and the probability that the height (shifted by a logarithmic term in $t$) is equal to a fixed value $j$ vanishes as $e^{-\Phi(\lambda j)/\sqrt{\lambda^4 t}}$, where the large deviation function $\Phi$ generally depends on the initial condition. Exact expressions for $\Phi$ have been obtained for various initial conditions from exact results for the full probability of the height for the KPZ equation [335, 336], see also [337] for sub-leading terms and [338] for an approach using the fact that a generating function of the height solves the KP equation. The large deviation function $\Phi$ has also been obtained from the path integral approach discussed in section 2.2.5. The tails of $\Phi$ were computed from the system of coupled heat equations [339–343], see also [344–346] for finite volume effects and [347] for a rigorous approach, and the classical integrability of that system has led to an exact solution for the full function $\Phi$ [235, 236].

Unexpectedly, the expression obtained for a Legendre transform of $\Phi$ with narrow wedge initial condition on the infinite line (respectively on the half-line $\mathbb{R}^+$, with an infinite slope for the height at the boundary $x = 0$) coincides up to some factors with the function (11) below describing stationary large deviations at the KPZ fixed point in finite volume with periodic boundaries (resp. (13) corresponding to stationary large deviations with open boundaries, and infinite slopes for the height at the boundaries), see section 4 of [337]. In the narrow wedge case, the connection extends to the leading correction (18) to large deviations [337], but not to the infinitely many further corrections of the probability of the height for the KPZ equation on the infinite line, which do not have a counterpart for stationary large deviations in finite volume since the Markov evolution has a discrete spectrum, see the remark below (15). It would be interesting to understand whether there could exist some observable related to the height of the KPZ equation on the infinite line whose large deviations have an expansion that terminates, like for stationary large deviations in finite volume (15). If so, the correspondence between the two regimes might appear more clearly on that observable.

We emphasize that the correspondence discussed above, represented by a dotted line in figure 1, relates two completely distinct regimes and does not follow by the usual rescaling of the KPZ equation. The correspondence is somewhat reminiscent of the modular invariance of the partition function for conformal field theories on a torus, where a very long torus and a very large torus are related by exchanging the two coordinates. This reasoning is however not expected to hold for KPZ fluctuations, for which the dynamical exponent $z = 3/2$ signals a strong anisotropy between the time and position variables.

## 2.4   Exact results in finite volume

In finite volume, the position $x$ belongs to an interval of finite size, taken equal to $[0, 1]$ in the following, and for any initial condition the statistics of the KPZ height at position $x$ depends both on time and on the coefficient $\lambda$ of the non-linearity (or time and the external force driving the particles for ASEP, or the aspect ratio of the random medium and the strength of the noise in the directed polymer setting), without any possibility to eliminate either parameter by rescaling unlike for the case on the infinite line.

---

[8]The identification of the left and right tails follows from our conventions in (1).

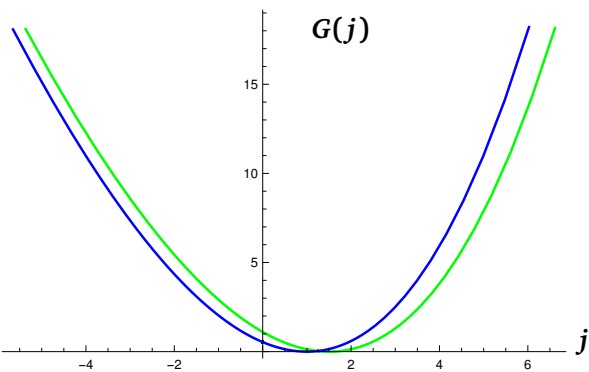

Figure 6: Stationary large deviation function $G(j)$ at the KPZ fixed point with periodic boundaries (blue) and open boundaries with slopes $\sigma_a = -\infty$, $\sigma_b = \infty$ (green). In both cases, the left tail grows as $G(j) \simeq \frac{4}{3}|j|^{3/2}$ and the right tail as $G(j) \simeq \frac{2\sqrt{3}}{5\pi}j^{5/2}$.

### 2.4.1  Stationary large deviations

The stationary state discussed in section 2.1.6 does not say anything about the global growth of the interface at late times, which requires to take into account the dynamics and not just the invariant measure. Typically (i.e. with probability $1$), the height simply grows for large $t$ as $h_\lambda(x,t) \simeq Jt$ where $J = \lim_{t\to\infty}\langle h_\lambda(x,t)\rangle/t$ is independent of the position $x$ and of the initial condition, and the fluctuations of the random variable $h_\lambda(x,t) - Jt$ are Gaussian in the long time limit.

Rare events with $h_\lambda(x,t)/t = j \neq J$ for long times are still independent of $x$ and of the initial condition, but correspond to non-Gaussian fluctuations. Their statistics, which is described in terms of a large deviation function $G_\lambda(j)$ as $\mathbb{P}(h_\lambda(x,t) = jt) \asymp e^{-tG_\lambda(j)}$, characterize stationary [9] KPZ fluctuations in finite volume. Equivalently, the Legendre transform $F_\lambda(s) = \max_j(js - G_\lambda(j))$ of $G_\lambda$ appears as

$$\langle e^{sh_\lambda(x,t)}\rangle \asymp e^{tF_\lambda(s)} \,. \tag{10}$$

For Markovian discrete models, $F_\lambda(s)$ is the eigenvalue with largest real part of a deformed generator associated to the height, see section 3.1.1. The cumulants of the height behave from (10) as $\langle h_\lambda(x,t)^k\rangle_c \simeq t\, c_k^{\mathrm{st}}(\lambda)$ at late times, and $F_\lambda(s)$ is the generating function of the stationary cumulants $c_k^{\mathrm{st}}(\lambda) = F_\lambda^{(k)}(0)$. At the KPZ fixed point $\lambda \to \infty$, (2) then implies $\langle e^{sh(x,t)}\rangle \asymp e^{tF(s)}$ with $F(s) = \lim_{\lambda\to\infty}\lambda^{-1}F_\lambda(s)$ the generating function of the stationary cumulants $c_k^{\mathrm{st}} = F^{(k)}(0)$.

At the KPZ fixed point with periodic boundary condition, the variance $c_2^{\mathrm{st}} = \sqrt{\pi}/2$ was first obtained in [348] for TASEP, see also [349] for a rigorous approach from the stochastic heat equation (4). An explicit parametric expression was eventually obtained for $F(s)$ [186, 226, 350–354] from various exactly solvable microscopic models. One has $F(s) = \chi(v)$ with $v$ solution of $\chi'(v) = s$ and $\chi$ defined as

$$\chi(v) = -\frac{\mathrm{Li}_{5/2}(-e^v)}{\sqrt{2\pi}} \,. \tag{11}$$

Here $\mathrm{Li}_s$ is the polylogarithm of index $s$, defined by $\mathrm{Li}_s(z) = \sum_{k=1}^\infty \frac{z^k}{k^s}$ for $|z| < 1$, and which can be extended to a holomorphic function in $\mathbb{C} \setminus [1, \infty)$ by analytic continuation, see e.g.

---

[9]Stationary fluctuations obtained at late times with arbitrary initial condition are completely distinct from the time-dependent fluctuations starting with stationary initial condition considered in section 2.4.4.

| | periodic | open $\sigma_a = -\infty$ $\sigma_b = \infty$ | open $\sigma_a = 0$ $\sigma_b = 0$ |
|---|---|---|---|
| $c_1^{\mathrm{st}}$ | $1$ | $3/2$ | $0$ |
| $c_2^{\mathrm{st}}$ | $\frac{\sqrt{\pi}}{2}$ | $\frac{3\sqrt{\pi}}{4\sqrt{2}}$ | $\frac{2}{\sqrt{\pi}}$ |
| $c_3^{\mathrm{st}}$ | $(\frac{3}{2} - \frac{8}{3\sqrt{3}})\pi$ | $(\frac{27}{8} - \frac{160}{27\sqrt{3}})\pi$ | $\frac{24}{\pi} - 8$ |
| $c_4^{\mathrm{st}}$ | $(\frac{15}{2} + \frac{9}{\sqrt{2}} - \frac{24}{\sqrt{3}})\pi^{3/2}$ | $(\frac{945}{32} + \frac{405}{8\sqrt{2}} - \frac{160}{\sqrt{6}})\pi^{3/2}$ | $\frac{192}{\sqrt{2\pi}} - \frac{288}{\sqrt{\pi}} + \frac{480}{\pi^{3/2}}$ |

Table 1: First stationary cumulants of the height $c_k^{\mathrm{st}} = F^{(k)}(0)$ at the KPZ fixed point with various boundary conditions.

[355]. This implies for the function $\chi$ the branch cuts $2\mathrm{i}\pi(j + 1/2) + [0, \infty)$, $j \in \mathbb{Z}$. In order to consider also complex values of $s$, we choose instead to rotate these branch cuts so that $\chi$ becomes analytic outside $\mathrm{i}(-\infty, -\pi] \cup \mathrm{i}[\pi, \infty)$ and coincides with (11) when $\mathrm{Re}\, v < 0$. The parametric expression above for $F(s)$ is then valid when $\mathrm{Re}\, s > 0$, and $\chi'(v) = s$ has a single solution. In particular, for all $s > 0$, $v \in \mathbb{R}$. Analytic continuation is needed for $s < 0$ [356].

Expanding near $s = 0^+$ (which corresponds to $v \to -\infty$ as $\mathrm{e}^v \simeq s\sqrt{2\pi}$) gives exact expressions for the stationary cumulants of the height $c_k^{\mathrm{st}}$, see table 1 for the first cumulants. The large deviation function $G(j)$ corresponding to $F(s)$ is plotted in figure 6. As with the various distributions on the infinite line, the tails are asymmetric.

The cumulants $c_k^{\mathrm{st}}$ are also known at the KPZ fixed point with open boundary conditions. The variance was first obtained in [357] as $c_2^{\mathrm{st}} = \frac{3\sqrt{\pi}}{4\sqrt{2}}$ for $\sigma_a = -\infty$, $\sigma_b = \infty$ and $c_2^{\mathrm{st}} = \frac{2}{\sqrt{\pi}}$ for $\sigma_a = \sigma_b = 0$. A parametric expression similar to the one with periodic boundaries was obtained for the generating function $F(s)$ with slopes $\sigma_a = -\infty$, $\sigma_b = \infty$ in [358–363]. The case of arbitrary finite slopes $\sigma_a \le 0 \le \sigma_b$ was extracted in [364] from the results of [358]: the generating function of the stationary cumulants is given by $F(s) = \chi_{\sigma_a, \sigma_b}(v)$ with $v$ solution of $\eta_{\sigma_a, \sigma_b}(v) = s$, where

$$\chi_{\sigma_a, \sigma_b}(v) = \frac{1}{6\pi} \int_{-\infty}^{\infty} \mathrm{d}y\, y^4 f_{\sigma_a, \sigma_b}(y; v) \quad \text{and} \quad \eta_{\sigma_a, \sigma_b}(v) = \frac{1}{2\pi} \int_{-\infty}^{\infty} \mathrm{d}y\, y^2 f_{\sigma_a, \sigma_b}(y; v)$$

(12)

with $f_{\sigma_a, \sigma_b}(y; v) = \left(1 + y^2 - \frac{\sigma_a^2}{y^2 + \sigma_a^2} - \frac{\sigma_b^2}{y^2 + \sigma_b^2}\right) / \left(y^2 + \frac{y^2 + \sigma_a^2}{1 + \sigma_a^2} \frac{y^2 + \sigma_b^2}{1 + \sigma_b^2} \mathrm{e}^{y^2 - v}\right)$. In the limit of infinite slopes, this can be rewritten more simply as $F(s) = s + \chi_\infty(v)/2$ (the extra term $s$ could be removed by an additional shift by $\tau$ of the height $h(x, \tau)$, which would amount to changing the way the KPZ equation is regularized) with $v$ solution of $\chi'_\infty(v) = s$ and

$$\chi_\infty(v) = -\frac{1}{4\pi} \int_{-\infty}^{\infty} \mathrm{d}y\, \mathrm{Li}_2(-y^2\, \mathrm{e}^{v - y^2}).$$

(13)

The corresponding large deviation function $G(j)$ is plotted in figure 6 for $\sigma_a = -\infty$, $\sigma_b = \infty$, and the first cumulants given in table 1.

It was observed in [359] that when one of the slopes is infinite and the other one is equal to zero, the function $\chi$ found for periodic boundary condition and given in (11) is unexpectedly recovered for the system with open boundaries, as $\chi_{0,\infty}(v) = \chi(v)/(4\sqrt{2})$ and $\eta_{0,\infty}(v) = \chi'(v)/(2\sqrt{2})$. In terms of TASEP, this corresponds at the level of the Markov generator to an inclusion property of the spectrum of open TASEP with $L$ sites and specific boundary rates within the spectrum of periodic TASEP with $L + 1$ particles and $2L + 2$ sites.

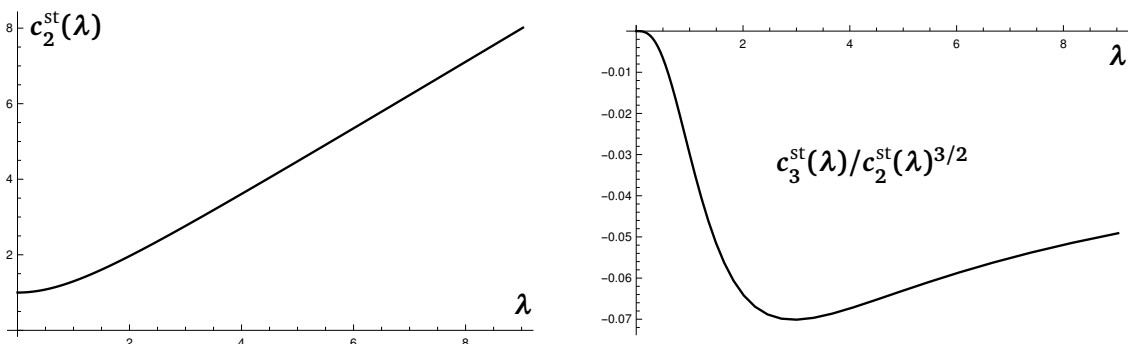

Figure 7: Second stationary cumulant $c_2^{\text{st}}(\lambda)$ (left) and skewness $c_3^{\text{st}}(\lambda)/c_2^{\text{st}}(\lambda)^{3/2}$ (right) plotted as a function of $\lambda$ for KPZ fluctuations with periodic boundary condition.

There is so far no satisfactory physical explanation for this puzzling observation, which is however reminiscent of various instances where open TASEP is interpreted as a periodic system with twice as many sites and some additional constraints, see e.g. [365,366], and also [367] for a general construction of the same nature for quantum integrable models with open boundary conditions.

Parametric expressions for cumulant generating functions have been found in other contexts, see for instance [368] where the parameter is a constant of integration obtained after integrating the saddle point equation in the path integral approach. For KPZ fluctuations in finite volume, the parameter $v$ solution of $\chi'(v) = s$ appears as a convenient variable for analytic continuation and is thus a particularly simple parametrization of the underlying Riemann surface, see section 2.4.6. A more physical interpretation is however missing.

Away from the KPZ fixed point, the first few stationary cumulants of the height $c_k^{\text{st}}(\lambda)$ for periodic boundaries were obtained explicitly [174,226,369–372] as a function of the parameter $\lambda$. One has $c_1^{\text{st}}(\lambda) = \lambda$ (a consequence of our choice of regularization for the KPZ equation), $c_2^{\text{st}}(\lambda) = 2\lambda \int_0^\infty du \, \frac{u^2 e^{-u^2}}{\tanh(\lambda u)}$ (which is closely related to the variance of the winding number of the random path around the cylinder in the directed polymer setting [373], see also [374]) and $c_3^{\text{st}}(\lambda) = -\frac{2\pi\lambda}{3\sqrt{3}} + 6\lambda \int_0^\infty du\,dv \, \frac{(u^2+v^2)e^{-u^2-v^2}-(u^2+uv+v^2)e^{-u^2-uv-v^2}}{\tanh(\lambda u)\tanh(\lambda v)}$, see figure 7. While no explicit expression is available so far for the function $F_\lambda(s)$, an iterative procedure [226] and exact combinatorial expressions involving trees [375] give in principle all the stationary cumulants $c_k^{\text{st}}(\lambda)$. In the case with open boundaries, it should be possible to extract the first cumulants for general boundary slopes $\sigma_a$, $\sigma_b$ from the exact result in [361] for ASEP.

In the limit $\lambda \to 0$ to the EW fixed point, the first stationary cumulants for periodic boundaries behave as $c_2^{\text{st}}(\lambda) \simeq 1 + \lambda^2/3$ and $c_3^{\text{st}}(\lambda) \simeq -\lambda^3/15$. More generally, matching with results from the path integral approach [376] and for ASEP with very weak asymmetry $1-q \sim 1/L$ [377] leads to $c_k^{\text{st}}(\lambda) \simeq \delta_{k,2} + B_{2k-2}(2\lambda)^k/(2(k-1)!)$ with $B_n$ the Bernoulli numbers. This implies that $F_\lambda(\frac{s}{2\lambda}) - \frac{s^2}{8\lambda^2}$ converges when $\lambda \to 0$ to

$$F_{\text{EW}}(s) = \frac{1}{2} \sum_{k=1}^\infty \frac{B_{2k-2}}{k!(k-1)!} s^k \,. \tag{14}$$

This function has radius of convergence $\pi^2$ and can be analytically continued to all $s \geq -\pi^2$, with the asymptotics $F_{\text{EW}}(s) \simeq \frac{2s^{3/2}}{3\pi}$ when $s \to +\infty$. The singularity at $s = -\pi^2$ signals a transition to a phase-separated regime, visible in the path integral approach as the point where the constant optimal density field becomes unstable and is replaced by a travelling wave

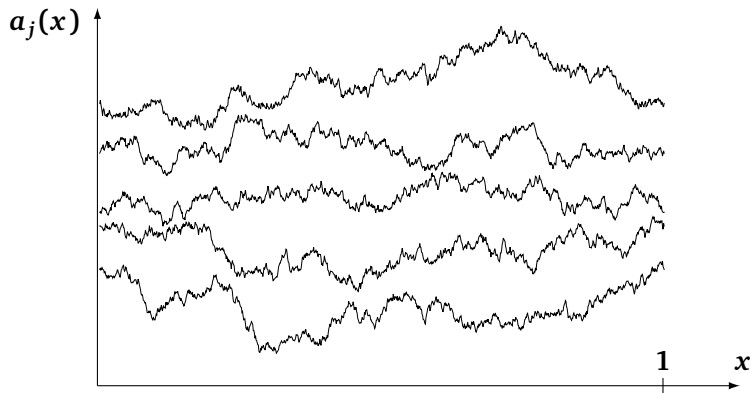

Figure 8: Typical realization of five non-intersecting Brownian bridges $a_j^s(x)$ with increments between endpoints exponentially distributed with parameter $s = 2$ as in equation (17).

[175,376,378]. The cumulant generating function (14) is universal, and has been obtained in various other models of diffusive particles [379–381], both with periodic and open boundary conditions, see also [382,383] in the context of quantum spin chains.

### 2.4.2 Approach to stationarity and statistics of Brownian bridges

We are interested in this section to the approach to stationarity of KPZ fluctuations in finite volume. We restrict to the KPZ fixed point with periodic boundaries, which is the only case where exact results are available. At long but finite times, one has

$$\langle e^{s h(x,t)} \rangle \simeq \theta(s) e^{t F(s)} \tag{15}$$

up to exponentially small terms corresponding to the contributions of higher eigenstates of the deformed Markov generator, see the next section. Here, the prefactor $\theta(s)$ depends on the initial condition $h_\lambda(x,0) = h_0(x)$ unlike $F(s)$. The first cumulants of the height at late times are thus given by

$$\langle h(x,t)^k \rangle_c \simeq t F^{(k)}(0) + \partial_s^k \log \theta(s)_{|s \to 0} , \tag{16}$$

up to exponentially small corrections in time.

It was shown in [384] based on a rewriting in terms of lattice paths of the iterated matrix product formalism introduced in [358] for TASEP with open boundaries, that in the expansion $\theta(s) = \theta_{\mathrm{bb}}^{(n)}(s) + \mathcal{O}(s^{n+1})$ up to arbitrary order $n \in \mathbb{N}$, one has the Brownian representation $\theta_{\mathrm{bb}}^{(n)}(s) = \langle e^{-s \max(b_1 - h_0) - s \sum_{j=2}^{n} \max(b_j - b_{j-1})} \rangle Z_n / Z_{2n}$ and $Z_m = \langle e^{-s \sum_{j=1}^{m} \max(b_j - b_{j-1})} \rangle$. The $b_j(x)$ with $j \in \mathbb{N}$ and $0 \leq x \leq 1$ are independent standard Brownian bridges, and a global shift of the initial height $h_0$ is chosen so that $h_0(0) = 0$. For $s > 0$, one has the equivalent representation in terms of non-intersecting Brownian bridges, see figure 8

$$\theta_{\mathrm{bb}}^{(n)}(s) = \frac{\mathbb{P}(a_{-1}^s < h_0 \,|\, a_{-n}^s < \ldots < a_{-1}^s)}{\mathbb{P}(a_{-1}^s < a_0^s \,|\, a_{-n}^s < \ldots < a_{-1}^s \text{ and } a_0^s < \ldots < a_n^s)} , \tag{17}$$

where the $a_j^s(x) - a_j^s(0)$, $0 \leq x \leq 1$ are independent standard Brownian bridges, the distances $a_{j+1}^s - a_j^s$ between consecutive endpoints $a_j^s = a_j^s(0) = a_j^s(1)$ are independent exponentially distributed random variables of parameter $s$, and $a_0^s = 0$.

From (15), one has in particular $\langle h(x,t) \rangle \simeq t + \frac{\sqrt{\pi}}{2} - \langle \max(b - h_0) \rangle$, up to exponentially small corrections in time, where the average on the right is over the Brownian bridge $b$. For an initial height with small amplitude, one has in particular the particularly simple expansion

$\langle\max(b - h_0)\rangle \simeq \frac{\sqrt{2\pi}}{4} - \mathfrak{c}_0 - \sqrt{2\pi}\sum_{k=-\infty}^{\infty}\mathfrak{c}_k\mathfrak{c}_{-k}(-1)^k k\pi J_1(k\pi)$ up to second order [384] in the Fourier coefficients $\mathfrak{c}_k$ of $h_0$, and with $J_1$ the Bessel function of the first kind. Higher orders are not known explicitly, nor are the terms of the expansion of higher cumulants of the height for an initial height with small amplitude.

From the Bethe ansatz solution for the full time evolution discussed in the next sections, exact formulas can be written for $\theta(s)$ in (15) with simple initial condition $h_0$. For the stationary, flat and narrow wedge initial conditions discussed in section 2.1.6, one has for all $s$ with $\mathrm{Re}\, s > 0$

$$
\begin{array}{rcl}
\theta_{\mathrm{stat}}(s) & = & \dfrac{\sqrt{2\pi}\, s^2\, \exp(\int_{-\infty}^{v} \mathrm{d}u\, \chi''(u)^2)}{\mathrm{e}^v\, \chi''(v)} \\[3mm]
\theta_{\mathrm{flat}}(s) & = & \dfrac{s\, \exp(\frac{1}{2}\int_{-\infty}^{v} \mathrm{d}u\, \chi''(u)^2)}{\mathrm{e}^{v/4}(1 + \mathrm{e}^{-v})^{1/4}\, \chi''(v)} \\[3mm]
\theta_{\mathrm{nw}}(s) & = & \dfrac{s\, \exp(\int_{-\infty}^{v} \mathrm{d}u\, \chi''(u)^2)}{\chi''(v)}
\end{array}
\tag{18}
$$

with $v$ the solution of $\chi'(v) = s$, and the choice of branch specified below (11) for $\chi$. Exact expressions for the leading correction in (16) to the cumulants can be extracted from (18).

Comparing (18) with (17) at large $n$ suggests expressions for the probability that non-intersecting Brownian bridges at finite density stay below an independent Brownian bridge ($\theta_{\mathrm{stat}}$) or a constant value ($\theta_{\mathrm{flat}}$), while $\theta_{\mathrm{nw}}$ corresponds to (17) with its numerator set equal to one. It would be nice to have a proof of these expressions directly from the Brownian bridges, without going through the relation to KPZ.

It would also be interesting to understand whether non-intersecting paths as in (17) also appear for the KPZ fixed point with open boundaries, with Brownian bridges perhaps replaced by the processes from [117, 119, 120] describing the stationary state. A possible approach would involve interpreting the iterated matrix product representation for TASEP in [358] in terms of lattice paths, as was done in [384] for the periodic case. Extensions to ASEP under the weakly asymmetric scaling (7), leading to KPZ fluctuations with finite $\lambda$, would also be nice, both for periodic and open boundaries.

Additionally, the appearance of non-intersecting Brownian bridges in (17) is reminiscent of a similar situation on the infinite line, where the Airy process is obtained as a scaling limit of the top line of a collection of a large number of non-intersecting Brownian paths. The main distinction appears to be that the density of Brownian bridges in (17) remains finite. It would be interesting to understand whether the finite time dynamics of KPZ fluctuations in finite volume discussed in the next sections, which interpolates between the KPZ fixed point on the infinite line and stationary fluctuations, can also be formulated in terms of non-intersecting paths.

### 2.4.3 Spectral gaps at the KPZ fixed point with periodic boundaries

The full finite time dynamics of KPZ fluctuations in finite volume involves corrections to the generating function (15) corresponding to the infinitely many higher eigenstates of the deformed Markov generator, see section 3.1. Again, we focus mainly on the KPZ fixed point with periodic boundaries, where exact results are available. Translation invariance of the dynamics implies that the generation function of $h(x, t)$ can be written as

$$
\langle \mathrm{e}^{sh(x,t)} \rangle = \sum_n \theta_n(s)\, \mathrm{e}^{2\mathrm{i}\pi p_n x + t e_n(s)}\,,
\tag{19}
$$

with eigenvalues $e_n(s)$, momenta $p_n \in \mathbb{Z}$, and $\theta_n(s)$ involving the overlap between the left eigenvector and the initial condition. The contribution of the stationary eigenstate $n = 0$ has

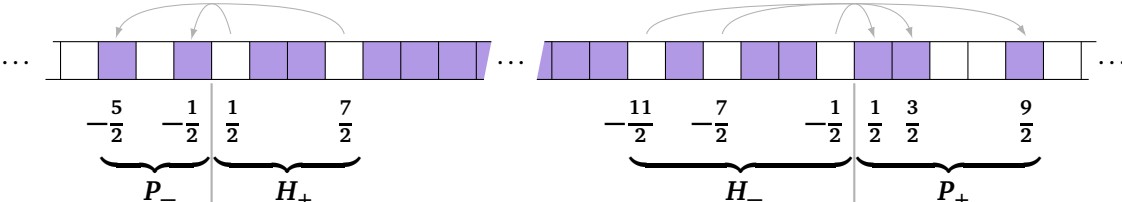

Figure 9: Particle-hole excitations labelling the eigenstates in (19), characterized by two finite sets of half-integers $P = P_+ \cup P_-$ (particles created outside of the Fermi sea) and $H = H_+ \cup H_-$ (holes in the Fermi sea) with $P_\pm, H_\pm \in \pm(\mathbb{N} + 1/2)$. The excitations on both sides of the Fermi sea are independent, and the cardinals of the set must verify $|P_+| = |H_-|$ and $|P_-| = |H_+|$.

eigenvalue $e_0(s) = F(s)$, momentum $p_0 = 0$ and $\theta_0(s) = \theta(s)$ in the notations of the previous section.

For $s \in \mathbb{R}$, the spectral gaps $\epsilon_n(s) = \mathrm{Re}(e_0(s) - e_n(s))$, $n \neq 0$ are all strictly positive and lead to exponentially small corrections at late time for the generating function $\langle e^{sh(x,t)} \rangle$, with relaxation times $\tau_n(s) = 1/\epsilon_n(s)$. The leading correction at long enough times comes from the eigenstate $n = 1$ with largest $\tau_n(s)$.

Special initial conditions with $\theta_1(s) = 0$ lead to a faster approach to stationarity, a phenomenon akin to the strong Mpemba effect discussed in [385], which does happen here at arbitrary $s$ for flat initial condition [386]. At $s = 0$, which governs fluctuations of the slope $\sigma = \partial_x h$, one has $e_0(0) = 0$, and the smallest gap is equal in our units to $\epsilon_1(0) \approx 13.0184$ [387], while the one relevant for flat initial condition is $\epsilon_2(0) \approx 32.0353$ [386, 388]. Fluctuations of the slope then reach their stationary state almost 2.5 times faster when starting from flat initial condition compared to even a small perturbation of the stationary state.

Exact expressions were obtained for the infinitely many spectral gaps $e_n(s)$ and the corresponding momenta $p_n$ in (19) (and also for the coefficients $\theta_n(s)$ for special initial conditions, see the next section) from large system size asymptotics of exact Bethe ansatz formulas for TASEP and related models, see section 3.2. The smallest spectral gap $e_1$ was first obtained in [387], see also [191, 389, 390], and all the higher gaps and corresponding momenta in [388, 391, 392].

Each eigenstate $n$ in (19) has a convenient representation in terms of particle-hole excitations at the edges of a Fermi sea, see figure 9. The Fermi sea represents the stationary state, and two sets of half-integers $P$ and $H$ corresponding to momenta of particles and holes characterize the eigenstate. Setting $n = (P, H)$, the summation in (19) is over all finite sets $P, H \subset \mathbb{Z} + 1/2$ such that $P$ and $-H$ have the same number of positive elements and the same number of negative elements, i.e. the excitations on each side of the Fermi sea do not mix. The number of eigenstates grows exponentially fast with the size of the sets $P$ and $H$, as $\exp(2\pi\sqrt{q/3})$ with $q = \sum_{a \in P} |a| + \sum_{a \in H} |a|$ [392]. The momentum $p_n$ in (19) is the total momentum of the corresponding particle-hole excitations, equal to $p_n = \sum_{a \in P} a - \sum_{a \in H} a$.

Restricting to $\mathrm{Re}\, s > s_n$ (for some $s_n$ that verifies $s_0 = 0$ and $s_n < 0$ for all other $n$), all the eigenvalues have the same parametric representation as $F(s)$, $e_n(s) = \chi_{P,H}(v)$ with $v$ solution of $\chi'_{P,H}(v) = s$. The function $\chi_{P,H}$ is equal to $\chi(v)$ from (11) plus extra terms contributed to by the particle-hole excitations, as

$$\chi_{P,H}(v) = \chi(v) + \Big( \sum_{a \in P} + \sum_{a \in H} \Big) \frac{(4\mathrm{i}\pi a)^{3/2}}{3} \Big( 1 - \frac{v}{2\mathrm{i}\pi a} \Big)^{3/2}, \tag{20}$$

and is again considered as analytic outside the branch cut $\mathrm{i}(-\infty, -\pi] \cup \mathrm{i}[\pi, \infty)$.

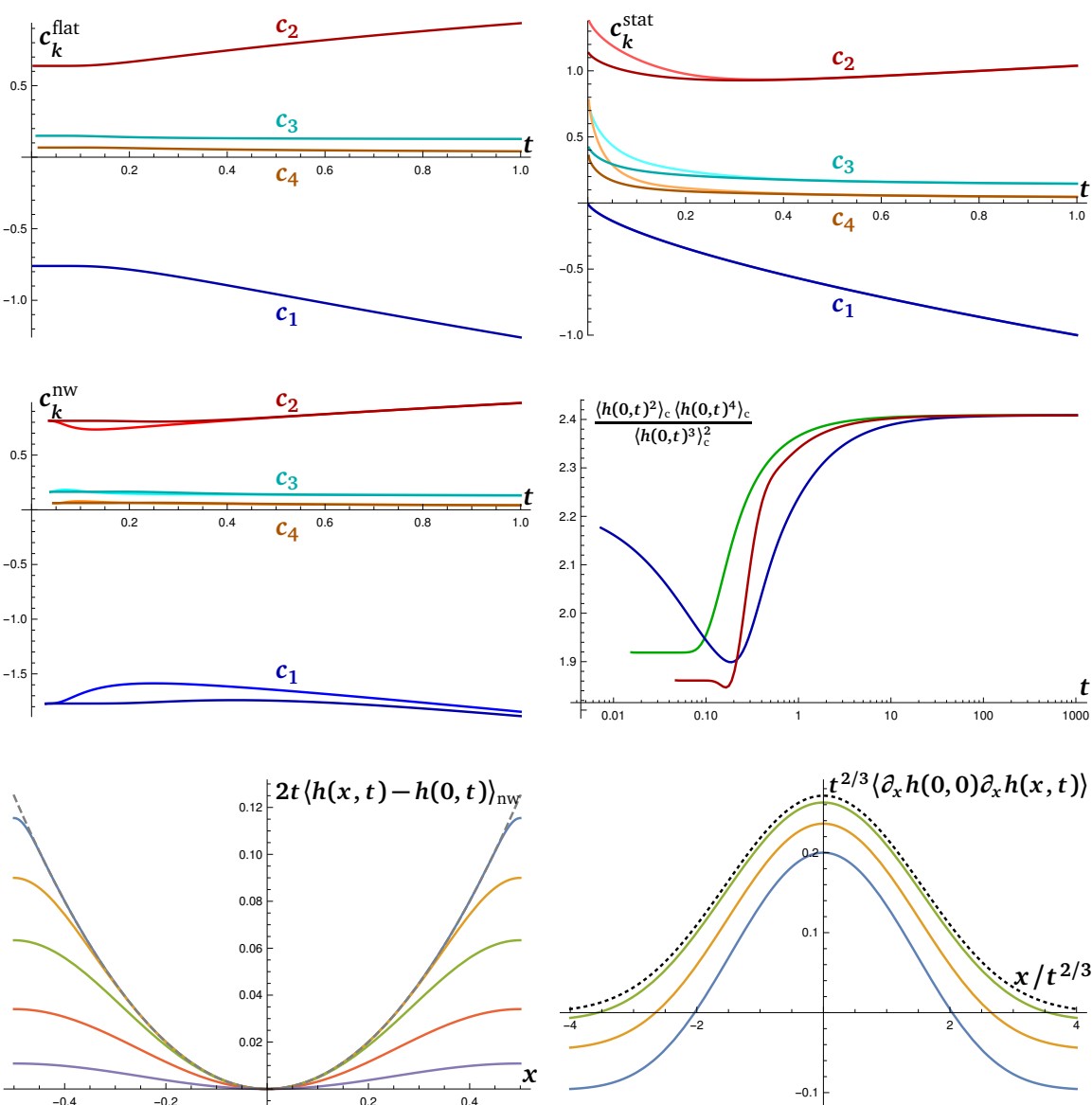

Figure 10: Plots of universal scaling functions at the KPZ fixed point with periodic boundary condition. The cumulants of the height $c_k = (-1)^k \langle h(x,t)^k \rangle_c / t^{k/3}$ at position $x$ are plotted as a function of time, for flat (top left), stationary (top right; $x = 0$ (darker) and $x = t^{2/3}$ (lighter)) and narrow wedge (middle left; $x = 0$ (darker) and $x = 2/5$ (lighter)) initial condition. When $t \to 0$, the cumulants respectively converge to those of the GOE [7] Tracy-Widom, Baik-Rains and GUE Tracy-Widom distributions of figure 4, and grow as $t$ times the stationary cumulants from section 2.4.1 for large $t$. A ratio of cumulants considered in [356] for stationary large deviations and which converges to finite non-zero values both at short and long times is plotted (middle right; log scale for the time) at $x = 0$ for flat (green), stationary (blue) and narrow wedge (red) initial condition. At bottom left, the average height difference $\langle h(x,t) - h(0,t) \rangle$ for narrow wedge initial condition, approximately equal to the parabola $x^2/4t$ at short time, is plotted as a function of $x$ at various times, from $t = 0.03$ (top) to $t = 0.35$ (bottom). At bottom right, the covariance of the slope with stationary initial condition is plotted as a function of the rescaled distance for various values of $t$, from $t = 0.001$ (top) to $t = 0.03$ (bottom), and the Prähofer-Spohn scaling function of figure 5 (dotted curve) is recovered at short time.

### 2.4.4   Exact expression for the generating function of the height

Expressions for the coefficients $\theta_n(s)$ in (19) were obtained from large system size asymptotics of exact Bethe ansatz formulas for various overlaps of eigenvectors of TASEP [393], see section 3.2, first for an evolution conditioned on simple initial and final states [394, 395], and eventually for the free evolution without conditioning on the final state, for stationary, flat and narrow wedge initial condition [396]. For these three initial conditions, $\theta_n(s)$ generalizes $\theta(s)$ in (18) as [10]

$$
\begin{aligned}
\theta_{P,H}^{\text{stat}}(s) &= \frac{\sqrt{2\pi}\, s^2\, D_{P,H}(v)^2}{e^v\, \chi_{P,H}''(v)} \\[2mm]
\theta_{P,H}^{\text{flat}}(s) &= 1_{\{P=H\}}\frac{s\, D_{P,H}(v)}{e^{v/4}(1+e^{-v})^{1/4}\,\chi_{P,H}''(v)} \\[2mm]
\theta_{P,H}^{\text{nw}}(s) &= \frac{s\, D_{P,H}(v)^2}{\chi_{P,H}''(v)}\,,
\end{aligned}
\tag{21}
$$

where $v$ is the solution of $\chi_{P,H}'(v) = s$. The factor $D_{P,H}(v)$ is defined in terms of a regularized integral as $D_{P,H}(v) = \frac{(\pi/2)^{m^2}}{(2i\pi)^m}V_P V_H \exp\big(\lim_{\Lambda\to\infty} -m^2\log\Lambda + \frac{1}{2}\int_{-\Lambda}^{v}\mathrm{d}u\,\chi_{P,H}''(u)^2\big)$, where $m$ is the common cardinal of $P$ and $H$ and $V_S = \prod_{a>b\in S}(a-b)$. In the sector $P = H$, the numerical factor in front of the exponential in the definition of $D_{P,H}(v)$ can be understood by analytic continuation from (18) [397]. In terms of the underlying Riemann surface $\mathcal{R}_{\text{KPZ}}$ discussed in section 2.4.6, which specifies how analytic continuations in the parameter $s$ leads to permutations of the eigenstates $n = (P, H)$, this means that the sector $P = H$ corresponds to the connected component of $\mathcal{R}_{\text{KPZ}}$ containing the stationary state. In particular, the restriction $P = H$ for flat initial condition eliminates the contribution of the eigenstates from all other connected components, including the one corresponding to the smallest spectral gap, and is responsible for the strong Mpemba effect discussed in section 2.4.3.

Convergence to the stationary large deviations from section 2.4.1 at late times is clear from the expressions above. Convergence at short time to the KPZ fixed point on the infinite line is harder. So far, it was only proved that Fredholm determinant expressions (see below) for the one-point distribution of $-t^{-1/3}h(x,t)$ with narrow wedge initial condition (at $x = 0$ only, although the result is expected to hold for arbitrary $x$ away from the shock $x = 1/2$) and flat initial conditions (whose statistics is independent of the position $x$) respectively converge at short time to GUE and GOE [7] Tracy-Widom distributions [241]. For stationary initial condition, where $h_0(x)$ is a Brownian bridge, the distribution of the height is expected to converge at short time to the $x$-dependent Baik-Rains distribution under a rescaling of $x$ by $t^{2/3}$, and to the distribution of the Brownian bridge at finite position $x$, $0 < x < 1$, i.e. a Gaussian distribution with mean zero and variance $x(1-x)$. Interestingly, non-trivial large deviations may appear instead at short time for an evolution conditioned both on the initial and the final state, in particular $P_t(u) \asymp \exp\big((2t)^{-2/3}\int\frac{\mathrm{d}w}{2i\pi}\,w\,\log(1-e^{w^3/3+2^{2/3}uw})\big)$ where the integral is between two specific directions at infinity for the probability distribution function $P_t(u)$ of $h(x,t)$ with flat initial and final states [395], see also [398] where the existence of large deviations at short times with exponent $t^{-2/3}$ was argued on the basis of local stationarity.

The first cumulants $\langle h(x,t)^k\rangle_c = \partial_s^k \log\langle e^{sh(x,t)}\rangle_{|s\to 0}$ of the height are plotted as a function of time in figure 10 for flat, stationary and narrow wedge initial condition from the exact result above for the eigenvalues $e_n(s)$ and the coefficients $\theta_n(s)$. They interpolate between the cumulants of the KPZ height on the infinite line at short time (after a rescaling of the height

---

[10]The coefficients $D_{P,H}(v)$ in (21) differ slightly from those in [396], but the corresponding expressions for the $\theta_{P,H}(s)$ are equivalent.

by $t^{1/3}$), and $\langle h(x,t)^k \rangle_c \simeq t c_k^{\text{st}} + \partial_s^k \log \theta(s)_{|s \to 0}$ at late times with $c_k^{\text{st}} = F^{(k)}(0)$ the station-ary cumulants. From (19), the cumulants $\langle h(x,t)^k \rangle_c$ have an essential singularity at $t = \infty$. Interestingly, they also have an essential singularity at $t = 0$, with a correction to the Tracy-Widom cumulants decaying as $e^{-(t/\tau)^{-2/3}}$ [241], where $\tau$ is a characteristic time controlling the emergence of finite volume effects distinct from the characteristic time of section 2.4.3 for the relaxation to the stationary state. This is consistent with the very fast convergence at short time of the cumulants for flat and narrow wedge initial condition in figure 10. It would be nice to understand more precisely the exponentially small corrections at short time, and in particu-lar whether the generating function $\langle e^{sh(x,t)} \rangle$ has an alternative representation similar to the late time expansion (19), with $e^{te_n(s)}$ replaced by $e^{t^{-2/3}a_n(s)}$. It is not clear however that the coefficients in front of the exponentials, akin to the $\theta_n(s)$ in (19), would still be independent of time.

The average slope $\langle \partial_x h(x,t) \rangle$ can be computed from the generating function (19) as $\lim_{s \to 0} \partial_x \langle e^{sh(x,t)} \rangle$. While it is always equal to zero for flat and stationary initial condition, it does converge at short time to the expected parabola $\langle \partial_x h(x,t) \rangle \simeq \frac{x^2}{4t}$ for narrow wedge initial condition, see figure 10. General higher cumulants of $\partial_x h$ can not be extracted directly from (19). An exact expression of the form (19) with a single sum over eigenstates $n$ evaluated at $s = 0$ should however exist for them, with appropriate insertions of operators correspond-ing to density fluctuations in the context of driven particles. Such an expression would in particular give access to the Family-Vicsek scaling function (see section 2.1.2) characterizing the growth and saturation of the width of the interface. More generally, it would be particu-larly interesting to obtain a reasonably simple characterization of the time dependent process $x \mapsto h(x,t) - h(0,t)$ which is expected to interpolate between e.g. the Airy process at short time and a Brownian bridge at long time, and elucidate the relation with the non-intersecting Brownian bridges with exponentially distributed distances between the endpoints discussed in section 2.4.2.

As on the infinite line, the covariance $\langle \partial_x h(0,0) \partial_x h(x,t) \rangle$ in the stationary state does not require correlations in time thanks to the identity $\langle \partial_x h(0,0) \partial_x h(x,t) \rangle = \frac{1}{2} \partial_x^2 \langle h(x,t)^2 \rangle$ [242], and can thus be computed from the generating function (19) as $\lim_{s \to 0} \frac{1}{s^2} \partial_x^2 \langle e^{sh(x,t)} \rangle$, which amounts to replacing $s^2$ by $(2i\pi p_n)^2$ in $\theta_n^{\text{stat}}(s)$. As expected, the covariance appears to converge at short time under rescaling of $x$ by $t^{2/3}$ to the Prähofer-Spohn scaling function appearing on the infinite line, see figure 10, although a direct proof from the exact expression in finite volume is still missing.

### 2.4.5 Probability distribution and Riemann surfaces

Fourier transform of the generating function $\langle e^{sh(x,t)} \rangle$ with respect to the variable $s$ combined with the expansion (19) gives an expression for the probability distribution of the height at a single position $x$ and time $t$, see figure 11 for plots. For the three usual initial conditions, the change of variable $s \to v$ with $\chi'_{P,H}(v) = s$ leads for the Fourier transform to an integral for $v \in c + i\mathbb{R}$ with $c > 0$. The sum over $n = (P,H)$ can then be performed explicitly in terms of a Fredholm determinant with discrete kernel after adding a contour integral to enforce the constraints on $P$ and $H$ [396].

An alternative approach to KPZ fluctuations in finite volume with periodic boundary con-dition relies on an integral formula [399–402] for the full probability of the TASEP height function at time $t$ starting from a given initial condition, in the spirit of similar formulas that were obtained for TASEP and ASEP on the infinite line [403–406]. The fact that the probabil-ities satisfy the master equation can be proved directly, making this approach fully rigorous. Expressions similar to the ones in [396] but with subtle differences were obtained using such integral formulas in [401, 407], see also [408] for a treatment of general initial condition

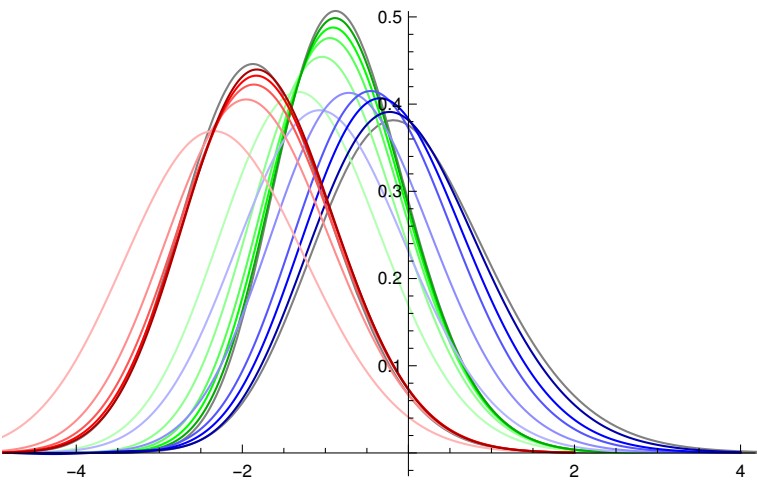

Figure 11: Probability distributions of $h(0, t)/t^{1/3}$ plotted for flat (green), stationary (blue) and narrow wedge (red) initial condition for small values of $t$, along with the corresponding Tracy-Widom and Baik-Rains distributions reached at short time (grey curves).

involving undetermined functions. Both sets of expressions for flat, stationary and narrow wedge initial conditions were eventually found to be equal in [397] by understanding them as contour integrals on a Riemann surface $\mathcal{R}_{\text{KPZ}}$ of infinite genus: the differences between them were merely a consequence from a different choice of contour ($v \in c + i\mathbb{R}$ with either $c > 0$ or $c < 0$) associated with a distinct partition of $\mathcal{R}_{\text{KPZ}}$ into sheets.

Extensions to joint statistics at multiple times were obtained in [302, 409, 410] from the integral formulas for the probability of TASEP. Although joint statistics at multiple points and the same time can in principle be extracted from those by taking the limit where all times are equal, the resulting expression still involves in essence a summation over as many eigenstates as the number of points, which should not be needed since the generating function of the height at multiple points requires a single expansion over eigenstates in that case. A simpler expression for joint statistics at multiple points is however still missing. The expressions for joint statistics at multiple times were interpreted in [397] in terms of multiple contour integrals on $\mathcal{R}_{\text{KPZ}}$, and were derived more directly later on from an interpretation of the Bethe ansatz formalism for TASEP on a finite lattice in terms of a compact Riemann surface [411], see section 3.2.2.

Like on the infinite line, connections to non-linear integrable partial differential equations have been discovered for the KPZ fixed point in finite volume. For flat initial condition, it was noted in [397] that in the identity $\mathbb{P}(h(y, t) > x) = \int_{-i\pi}^{i\pi} dv \, \tau(x, t; v)$ from [401], the Fredholm determinant with Cauchy kernel $\tau(x, 3t; v)$ is identical to the known $\tau$-function associated with a solution of the KdV equation (with the same normalizations as at the end of section 2.3.1) featuring infinitely many solitons with complex velocities $2v + 4i\pi a$, $a \in \mathbb{Z} + 1/2$. Soliton tau functions for the KP equation have been found instead for general initial condition [241]. Matrix valued KP equations appear for joint statistics at multiple times [313]. A soliton interpretation of KPZ fluctuations had been suggested earlier [412] from a path integral approach.

### 2.4.6 Analytic continuations

Using analytic continuation to compute higher eigenvalues of a quantum Hamiltonian is an old idea, see e.g. [413] for anharmonic oscillators or [414, 415] in a Bethe ansatz integrable

setting. The strategy was used recently for the spectral gap at the KPZ fixed point with open boundaries, see section 2.4.7.

Considering a matrix $H(s)$ depending analytically on a parameter $s$ (the deformed Markov generator for $\langle e^{sh(x,t)} \rangle$ in the KPZ setting, see section 3.1), analytic continuations of the spectrum with respect to $s$ constructs a Riemann surface $\mathcal{R}$ such that the set of all eigenstates of $M(s)$ for all complex values of $s$ are in one to one correspondence with the points of $\mathcal{R}$, except for a discrete set of exceptional points [416] where several eigenstates coincide. The non-trivial topology of the spectrum under analytic continuation has been particularly studied in non-Hermitian settings [417–422], where the existence of exceptional points for physically accessible values of the parameter $s$ leads for instance to symmetry breaking induced by the reduction of the eigenspace dimension.

At the KPZ fixed point with periodic boundaries, the Riemann surface $\mathcal{R}_{\text{KPZ}}$ is already apparent at the level of stationary large deviations, from the square root branch point singularities of the function $\chi$ defined in (11). Indeed, Jonquière's relation between $\text{Li}_s$ and the Hurwitz zeta function $\zeta(s,z) = \sum_{n=0}^{\infty}(n+z)^{-s}$ (defined for $\text{Re } s > 1$ and extended to a holomorphic function on $\mathbb{C} \setminus \{1\}$) implies $\chi(\nu) = \frac{8\pi^{3/2}}{3}\big(e^{i\pi/4}\zeta(-3/2, \frac{1}{2} + \frac{\nu}{2i\pi}) + e^{-i\pi/4}\zeta(-3/2, \frac{1}{2} - \frac{\nu}{2i\pi})\big)$, which is valid for all complex values of $\nu$ outside $i(-\infty, -\pi] \cup i[\pi, \infty)$, and has thus the correct branch cuts required below (11). It follows that $\chi(\nu)$ can be written alternatively as $\chi(\nu) = \lim_{M\to\infty}\big(P_M(\nu) - \sum_{a=-M+1/2}^{M-1/2}\frac{(4i\pi a)^{3/2}}{3}(1 - \frac{\nu}{2i\pi a})^{3/2}\big)$, where the sum is over half-integers $a \in \mathbb{Z} + 1/2$, and $P_M(\nu)$ is a specific polynomial in $\nu$ and $\sqrt{M}$ subtracting divergent terms.

Analytic continuation in the variable $\nu$ for $\chi(\nu)$ thus simply amounts to flipping the sign of some of the terms in the infinite sum representation above, and the function $\chi$ and all its derivatives live on the Riemann surface obtained by gluing together the infinitely many sheets corresponding to all possible choices of signs. This Riemann surface is in fact only one of the infinitely many connected component of $\mathcal{R}_{\text{KPZ}}$, the other ones corresponding to removing some of the square roots altogether. This is visible on the expression (20) for the function $\chi_{P,H}$ in terms of which the spectral gaps are written: $a \in P \cap H$ corresponds to flipping the sign of $(1 - \frac{\nu}{2i\pi a})^{3/2}$, while $a \in P$, $a \notin H$ (or $a \notin P$, $a \in H$) removes this term from the sum. For flat initial condition, since only the connected component $P = H$ contributes, the exact expressions for the $e_n(s)$ and $\theta_n(s)$ can be fully understood from analytic continuations of the corresponding stationary values $F(s)$ and $\theta_{\text{flat}}(s)$.

Analytic continuation suggests that the Riemann surface $\mathcal{R}_{\text{KPZ}}$ (or at least its connected component $P = H$) should generally describe higher eigenstates for systems where the ground state eigenvalue is expressed in terms of a polylogarithm with half-integer index. In the textbook example of the one-dimensional ideal Fermi gas, this manifests itself in the identity [423] $Z = \prod_{j\in\mathbb{Z}}\big(1 + e^{\mu - \frac{1}{2}(\frac{2\pi j}{L})^2}\big) = e^{L\chi'(\mu)}\prod_{a\in\mathbb{Z}+1/2}\big(1 - e^{-L\sqrt{4i\pi a - 2\mu}}\big)^2$ for the grand-canonical partition function with chemical potential $\mu$ in a box of finite length $L$. Both expressions have indeed the same zeroes. The first expression is the definition of the partition function, from which the small $L$ behaviour can be extracted. The second expression, equal to $Z = \sum_{P,H\subset\mathbb{Z}+1/2}(-1)^{|P|+|H|}e^{L\chi'_{P,H}(\mu)}$ with a sum over finite sets $P$ and $H$, gives $Z \simeq e^{L\chi'(\mu)}$ at large $L$, from which the known expression for the free energy is recovered. In comparison with (19), there is no prefactor $\theta_n$ in front of the exponentials here since $Z$ is the trace of the exponential of the Hamiltonian, and thus does not involve eigenvectors.

A similar phenomenon could happen in various contexts where some observable is given by a polylogarithm with half-integer index, for instance the partition function of lattice paths on a torus [424], large deviations for the largest real eigenvalue for random matrices in the real Ginibre ensemble [425–427], large deviations for the current [239, 428] and the position of a tagged particle [429–432] in single file diffusion, large deviations for the return proba-

bility after a quench for the XXZ spin chain [433], and as already mentioned at the end of section 2.3.2, large deviations at the EW fixed point on the infinite line. In the case of large deviations, the Riemann surface $\mathcal{R}_{\mathrm{KPZ}}$ might in particular describe typical fluctuations of the observable, like at the KPZ fixed point in finite volume.

Additionally, we note that sums of infinitely many square root singularities, leading to a similar structure for analytic continuations as for the function $\chi$ and called remnant functions in [434], have been obtained in other contexts, for instance the two-dimensional Ising field theory in finite volume [435, 436], and also the EW fixed point in finite volume, see section 2.4.8 below.

### 2.4.7   Spectral gaps at the KPZ fixed point with open boundaries

The eigenstate expansion (19) for the generating function $\langle e^{sh(x,t)} \rangle$ is in principle valid for arbitrary boundary condition at the KPZ fixed point. While the coefficients $\theta_n(s)$ are only available so far for periodic boundaries (with simple initial conditions, as discussed in the previous section), exact expressions for the eigenvalues $e_n(s)$ have been obtained by analytic continuation of the cumulant generating function $F(s)$ from section 2.4.1.

For arbitrary slopes with $\sigma_a \leq 0 \leq \sigma_b$, analytic continuation in the variable $s$ can be performed easily in the variable $v$ such that $s = \eta_{\sigma_a,\sigma_b}(v)$, with $\eta_{\sigma_a,\sigma_b}(v)$ defined in (12), by deforming the integration path for $y$ in (12) so that the poles of the integrand $f_{\sigma_a,\sigma_b}(y;v)$ never cross the integration path. Equivalently, analytic continuation amounts to adding contributions of the residues at those poles to $\eta_{\sigma_a,\sigma_b}(v)$ and $\chi_{\sigma_a,\sigma_b}(v)$. The poles $y_j(v)$, $j \in \mathbb{Z}$ of $f_{\sigma_a,\sigma_b}(y;v)$ are located at $y_j = \pm i(W_j^{\sigma_a,\sigma_b}(e^{-v}))^{1/2}$ where the $W_j$ are branches of a modified Lambert function solution of $W_j^{\sigma_a,\sigma_b}(z) + \log \frac{(\sigma_a^2+1)(\sigma_b^2+1) W_j^{\sigma_a,\sigma_b}(z)}{(\sigma_a^2 - W_j^{\sigma_a,\sigma_b}(z))(\sigma_b^2 - W_j^{\sigma_a,\sigma_b}(z))} = \log z + 2i\pi j$, which reduces for infinite slopes to the standard Lambert function, see e.g. [437].

The Lambert functions themselves must be properly analytically continued when moving $v$, which makes the situation a bit more complicated than in the periodic case, where the analogue of $W_j(e^{-v})$ is simply $-v + 2i\pi(j + 1/2)$. Ultimately, the procedure gives a Riemann surface $\mathcal{R}_{\mathrm{KPZ}}^{\sigma_a,\sigma_b}$ with a different connectivity compared to the one in the previous section. When $\sigma_b = \infty$, a natural labelling of the sheets of $\mathcal{R}_{\mathrm{KPZ}}^{\sigma_a,\infty}$ with respect to the variable $v$ was found in terms of identical particle-hole excitations at both sides of a Fermi sea [364, 438], corresponding in figure 9 to sets $P = H$.

Additionally, since stationary large deviations with boundary slopes $\sigma_a = 0$, $\sigma_b = \infty$ involve the function $\chi$ in (11) corresponding to the KPZ fixed point with periodic boundaries, see section 2.4.1, the Riemann surface $\mathcal{R}_{\mathrm{KPZ}}^{\sigma_a,\infty}$ (which has a single connected component for finite $\sigma_a$) interpolates between the Riemann surface $\mathcal{R}_{\mathrm{KPZ}}$ with periodic boundaries (or more precisely its connected component containing the stationary state since only particle-hole excitations with sets $P = H$ appear for $\mathcal{R}_{\mathrm{KPZ}}^{\sigma_a,\infty}$) and $\mathcal{R}_{\mathrm{KPZ}}^{-\infty,\infty}$ corresponding to infinite boundary slopes, which has infinitely many connected components.

### 2.4.8   Crossover between the EW and KPZ fixed points

The eigenstate expansion of the generating function $\langle e^{sh_\lambda(x,t)} \rangle$ on the crossover between the EW and KPZ fixed points in finite volume is expected to have the same form (19) for arbitrary $\lambda$, with corresponding eigenvalues $e_n(s;\lambda)$ and coefficients $\theta_n(s;\lambda)$, about which very little is known so far beyond the eigenvalue $e_0(s;\lambda) = F_\lambda(s)$ discussed toward the end of section 2.4.1. Indeed, the corresponding Riemann surface $\mathcal{R}_\lambda$ appears to have a much more complicated branching structure than at the KPZ fixed point, see section 3.3.3 for a discussion of Bethe ansatz approaches. We discuss below the limits to the EW and KPZ fixed points for the case with periodic boundaries.

In the limit $\lambda \to \infty$, the eigenvalues $e_n(s; \lambda)$ reduce to those at the KPZ fixed point $e_n(s)$ as $\lambda^{-1} e_n(s; \lambda) \to e_n(s)$. For the smallest spectral gap with $s = 0$, corrections of order $\lambda^{-2}$ and $\lambda^{-4}$ were computed in [388]. A systematic expansion $\lambda \to \infty$ for the probability of the height $h_\lambda(x, t)$ could in principle be extracted from an expansion for ASEP near $q = 0$ [439].

The EW equation (1) with $\lambda = 0$ is linear in the height $h_0(x, t)$. Its Fourier solution, see e.g. [440,441], leads to $\langle e^{s h_0(x,t)} \rangle = \exp(s \mathcal{H}(x, t) + \frac{ts^2}{2} + \frac{s^2}{24} - s^2 \sum_{k=1}^{\infty} \frac{e^{-(2\pi k)^2 t}}{(2\pi k)^2})$ with $\mathcal{H}(x, t)$ the solution of the EW equation without noise. This implies that all $e_n(s; \lambda) - s^2/2$ converge to integer multiples of $2\pi^2$ when $\lambda \to 0$ with fixed $s$. Additionally, the expression (14) for the stationary large deviation function $F_{\text{EW}}(s) = \lim_{\lambda \to 0}(e_0(\frac{s}{2\lambda}; \lambda) - \frac{s^2}{8\lambda^2})$ can alternatively be written [376] as $F_{\text{EW}}(s) = \lim_{M \to \infty} \left( P_M(s) - \sum_{m=-M}^{M} \sqrt{m^2 \pi^2 (m^2 \pi^2 + s)} \right)$ with $P_M$ a polynomial, which has the same kind of analytic structure with infinitely many square root branch points as at KPZ fixed point with periodic boundaries, although here in the variable $s$ directly and not through a parametric representation involving the solution of $\chi'(v) = s$. This suggests that higher eigenvalues could simply be obtained by analytic continuation of $F_{\text{EW}}(s)$. Flipping the signs of some square roots would however increase the eigenvalue rather than decrease it, which can not happen since $e_0$ is the eigenvalue with largest real part. Limited numerics for ASEP suggest rather that $\lim_{\lambda \to 0}(e_n(\frac{s}{2\lambda}; \lambda) - \frac{s^2}{8\lambda^2})$ is equal to $2 F_{\text{EW}}(s) - F_{\text{EW}}^{\text{a.c.}}(s)$, where analytic continuation is only performed on the second term, at least for the first few spectral gaps. Obtaining the universal finite time dynamics in the vicinity of the EW fixed point with periodic boundaries would additionally require the small $\lambda$ behaviour of the coefficients $\theta_n(s; \lambda)$, which is still missing, but could presumably be obtained from the path integral approach.

Even less is known for the solution of the KPZ equation at finite $\lambda$ with open boundaries. We note however that an inclusion property for the spectrum of open ASEP within the spectrum of periodic ASEP for special boundary rates [362] implies that the Riemann surface $\mathcal{R}_\lambda^{\sigma_a, \sigma_b}$ with boundary slopes $\sigma_a$ and $\sigma_b$ must coincide with $\mathcal{R}_\lambda$ for periodic boundaries when either $\sigma_a = 0$ and $\sigma_b = \lambda$ or $\sigma_a = -\lambda$ and $\sigma_b = 0$. This generalizes a similar statement at the end of the previous section for the KPZ fixed point.

# 3    Riemann surface approach

We detail in this section how Riemann surfaces are the natural framework for the statistics at a finite time of current-like observables in Markov processes. The approach is explained in section 3.1 in the general context of integer counting processes, following [442], before turning in section 3.2 to height fluctuations for TASEP with periodic boundaries, expressed in terms of the Riemann surface introduced in [411]. Possible extensions to open boundary conditions, systems with several conserved quantities, or ASEP with weak asymmetry are finally discussed in section 3.3.

## 3.1    Integer counting processes

In this section, we explain the Riemann surface formalism for current fluctuations of Markov processes, which is then applied to height fluctuations for TASEP in section 3.2.

### 3.1.1    Generating function

We consider in the whole section 3.1 a general Markov process in continuous time, which is assumed to be ergodic (see below), and has a finite set of states $\Omega$ with cardinal $|\Omega|$. Since the process is memoryless, the dynamics is fully characterized by the transition rates $w_{\mathcal{C}' \leftarrow \mathcal{C}}$ from any state $\mathcal{C} \in \Omega$ to any other state $\mathcal{C}' \in \Omega$.

The infinitesimal evolution in time of the probability $P_t(\mathcal{C})$ that the system is in the state $\mathcal{C} \in \Omega$ at time $t$ is the sum of a gain term corresponding to transitions from any other state $\mathcal{C}'$ to $\mathcal{C}$ and a loss term corresponding to transitions from $\mathcal{C}$ to any other state $\mathcal{C}'$. This leads to the master equation $\frac{\mathrm{d}}{\mathrm{d}t} P_t(\mathcal{C}) = \sum_{\mathcal{C}' \neq \mathcal{C}} \left( w_{\mathcal{C} \leftarrow \mathcal{C}'} P_t(\mathcal{C}') - w_{\mathcal{C}' \leftarrow \mathcal{C}} P_t(\mathcal{C}) \right)$ where the sum is over all states $\mathcal{C}' \in \Omega$ distinct from $\mathcal{C}$. Introducing the probability vector $|P_t\rangle = \sum_{\mathcal{C} \in \Omega} P_t(\mathcal{C})|\mathcal{C}\rangle$ belonging to the vector space generated by the state vectors $|\mathcal{C}\rangle$, $\mathcal{C} \in \Omega$, the master equation rewrites $\frac{\mathrm{d}}{\mathrm{d}t}|P_t\rangle = M|P_t\rangle$, where the Markov matrix $M$ has non-diagonal terms $\langle \mathcal{C}'|M|\mathcal{C}\rangle = w_{\mathcal{C}' \leftarrow \mathcal{C}}$ and diagonal terms $\langle \mathcal{C}|M|\mathcal{C}\rangle = -\sum_{\mathcal{C}' \neq \mathcal{C}} w_{\mathcal{C}' \leftarrow \mathcal{C}}$. The probability vector at time $t$ is thus given in terms of the initial state $|P_0\rangle$ by $|P_t\rangle = \mathrm{e}^{tM}|P_0\rangle$.

Conservation of probability, ensured by $\sum_{\mathcal{C} \in \Omega} \langle \mathcal{C}|M = (0, \ldots, 0)$, implies that $\sum_{\mathcal{C} \in \Omega} \langle \mathcal{C}|$ is a left eigenvector of $M$ with eigenvalue $0$. We assume in the following that the process is ergodic, i.e. any two states can be reached from each other by successive allowed transitions. Then the eigenvalue $0$ is non-degenerate, and the only right eigenvector with eigenvalue $0$ is the unique stationary state $|P_{\mathrm{st}}\rangle$ of the process, reached from any initial state $|P_0\rangle$ in the long time limit.

In order to study non-equilibrium features of Markov processes, it is useful to consider quantities accumulated over time, whose value at a given time $t$ can not be expressed only from the probabilities $P_t(\mathcal{C})$. This is for example the case of time-integrated currents for selected transitions between states. Such observables $Q_t$, incrementing by quantities $\delta Q_{\mathcal{C}' \leftarrow \mathcal{C}}$ at the transitions of the underlying Markov process, belong to the class of integer counting processes [443, 444]. Their statistics can be obtained from a deformed version of the master equation, $\frac{\mathrm{d}}{\mathrm{d}t} F_t(\mathcal{C}) = \sum_{\mathcal{C}' \neq \mathcal{C}} \left( g^{\delta Q_{\mathcal{C} \leftarrow \mathcal{C}'}} w_{\mathcal{C} \leftarrow \mathcal{C}'} F_t(\mathcal{C}') - w_{\mathcal{C}' \leftarrow \mathcal{C}} F_t(\mathcal{C}) \right)$, where $F_t(\mathcal{C}) = \sum_{Q \in \mathbb{Z}} g^Q P_t(\mathcal{C}, Q)$ and $P_t(\mathcal{C}, Q)$ is the joint probability of finding the system at time $t$ in the state $\mathcal{C}$ with $Q_t = Q$. Introducing the corresponding deformation $M(g)$ of the Markov matrix, one has the generating function

$$\left\langle g^{Q_t} \right\rangle = \sum_{\mathcal{C} \in \Omega} \langle \mathcal{C}|\mathrm{e}^{tM(g)}|P_0\rangle \tag{22}$$

for the evolution starting with initial probabilities $P_0(\mathcal{C}) = \langle \mathcal{C}|P_0\rangle$. The matrix $M(g)$ is not a Markov matrix when $g \neq 1$ (it does not conserve the total probability), but is closely related by Doob's transform to the dynamics of the original Markov process conditioned on the late time value of $Q_t/t$ [445–447].

In the long time limit, one has $\log\left\langle g^{Q_t} \right\rangle \simeq t\mu_0(g)$ with $\mu_0(g)$ the eigenvalue of $M(g)$ with largest real part. Stationary large deviations of $Q_t$ are then encoded in the generating function of the cumulants, $\mu_0(\mathrm{e}^\gamma) \simeq J\gamma + \frac{D\gamma^2}{2} + \frac{C_3\gamma^3}{6}$ up to third order in $\gamma$, with $J = \lim_{t \to \infty} \frac{\langle Q_t \rangle}{t}$ controlling the average growth of $Q_t$, $D = \lim_{t \to \infty} \frac{\langle Q_t^2 \rangle - \langle Q_t \rangle^2}{t}$ the typical amplitude of fluctuations and $C_3 = \lim_{t \to \infty} \frac{\langle Q_t^3 \rangle - 3\langle Q_t \rangle \langle Q_t^2 \rangle + 2\langle Q_t \rangle^3}{t}$ the degree of asymmetry and non-Gaussianity in the distribution of $Q_t$.

We are interested in the following in the statistics of $Q_t$ at arbitrary finite time $t$, which depends from (22) on all the eigenstates of $M(g)$, and in particular on the corresponding eigenvalues $\mu_n(g)$ and eigenvectors $\langle \psi_n(g)|, |\psi_n(g)\rangle$ (which are not transposed of each other if $M(g)$ is not symmetric). Those are in general multivalued functions of $g \in \mathbb{C}$, each branch having discontinuities on various cuts. Analytic continuation of the eigenstates when varying $g$ across the cuts then allows to consider the eigenstates as simply valued functions on the Riemann surface made by gluing together copies $\mathbb{C}_n$ (also called sheets) of $\mathbb{C}$ along the cuts. This point of view will be particularly fruitful when considering the probability of the observable $Q_t$.

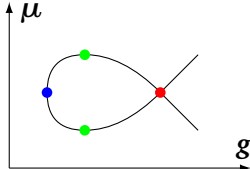 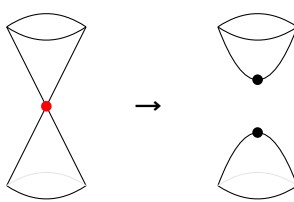

Figure 12: Special points on the spectral curve. On the left, a section of the spectral curve corresponding to e.g. real values of the parameters $g$ and $\mu$ is represented. The red point where the curve intersects itself is a singular point. The blue and green regular points on the curve, with tangents parallel to the axes, are ramified respectively for the parameter $g$ (exceptional point where $M(g)$ is not diagonalizable) and for the parameter $\mu$. On the right, example of desingularization of a complex algebraic curve, which splits the apex of the double cone (nodal singular point, in red) into two regular points on the resulting Riemann surface.

### 3.1.2 Riemann surfaces

Before considering the probability of the observable $Q_t$ defined in the previous section, we summarize in this section a few well known facts about Riemann surfaces, meromorphic functions on them, and how they describe eigenstates of a parameter-dependent matrix $M(g)$, see e.g. [448–450] for more detailed introductions.

Since all the entries of the matrix $M(g)$ defined in the previous section are monomials in the variable $g$ (with an exponent that may be negative), the characteristic equation

$$\det(M(g) - \mu \mathrm{Id}) = 0 \,, \qquad (23)$$

whose solutions $\mu(g)$ are the eigenvalues of $M(g)$, is a polynomial equation in both $g$ and $\mu$ and thus a complex algebraic curve, called the spectral curve of the process. The spectral curve may have (at most finitely many) singular points [11] around which it is not a surface but for instance a double cone (nodal point), see figure 12. A desingularization procedure removing the singular points, if any, leads to a compact surface $\mathcal{R}$, which may have several connected components. The surface $\mathcal{R}$ is endowed with a complex structure describing how local expressions of $\mu$ as a function of $g$ are related in various neighbourhoods, and $\mathcal{R}$ is then identified with a compact Riemann surface.

Any eigenstate of $M(g)$ for any complex value of $g$ corresponds to a point $p = [g, n] \in \mathcal{R}$, with $n$ labelling the eigenstates, and $\mu$ and $g$ are understood as meromorphic functions on $\mathcal{R}$ (i.e. functions whose only singularities are poles). Additionally, the eigenvectors $\langle \psi(p)|$ and $|\psi(p)\rangle$ can be normalized in such a way that all their entries are rational functions of $\mu(p)$ and $g(p)$, and thus also meromorphic on $\mathcal{R}$, since the left and right eigenvalue equations $\langle \psi(p)|M(g(p)) = \mu(p)\langle \psi(p)|$ and $M(g(p))|\psi(p)\rangle = \mu(p)|\psi(p)\rangle$ are linear in the eigenvectors.

Meromorphic functions on a compact Riemann surface $\mathcal{R}$ are highly constrained. For example, each meromorphic function $f$ has a fixed degree $d \in \mathbb{N}^*$, equal for any $z \in \widehat{\mathbb{C}}$ (the Riemann sphere $\mathbb{C} \cup \{\infty\}$) to the number of antecedents $p \in \mathcal{R}$, such that $f(p) = z$, counted with multiplicity. This implies in particular that $f$ has the same number of poles and zeroes

---

[11] Singular points on the spectral curve must not be confused with ramification points on the corresponding Riemann surface, see figure 12. Singular points are merely accidental degeneracies of the spectrum of $M(g)$, absent for generic $M(g)$, and correspond to the intrinsic property that the spectral curve in not locally a surface around the singular point. Ramification points *with respect to some parametrization* of the Riemann surface are caused by the presence of a branch point with respect to that parametrization. For the special case of the parametrization by $g$, they are called exceptional points, and correspond to values of $g$ where $M(g)$ is not diagonalizable.

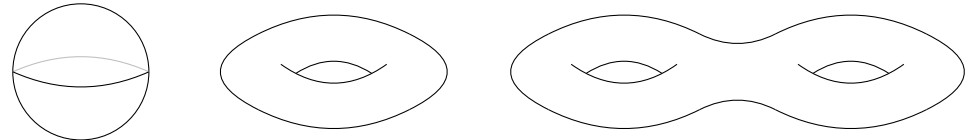

Figure 13: Compact surfaces of genus zero (isomorphic to the Riemann sphere $\widehat{\mathbb{C}}$), one (torus) and two.

(counted with multiplicity), which proves a useful consistency check when trying to locate all the zeroes of a function.

For meromorphic differentials, on the other hand, the number of zeroes minus the number of poles is equal to $2\mathfrak{g} - 2K$ with $K$ the number of connected components of $\mathcal{R}$ and $\mathfrak{g}$ its total (geometric) genus, see figure 13. Considering a meromorphic function $f$, this is equivalent for the differential $\mathbf{d}f$ to the Riemann-Hurwitz formula

$$\mathfrak{g} = -d + K + \frac{1}{2}\sum_{p \in \mathcal{R}}(e_p - 1), \tag{24}$$

where $d$ is the degree of $f$ and $e_p$ the ramification index of $p$ for $f$ (i.e. the winding number around $f(p)$ of the image by $f$ of a small loop around $p$, equal to $1$ except at a finite number of ramification points for $f$). For the special case of the function $g$, ramification points $p \in \mathcal{R}$ correspond for the matrix $M(g)$ to exceptional points [416] $g(p) = g_*$ where several eigenstates coincide, and $M(g_*)$ is not diagonalizable but rather has a Jordan block structure.

If the algebraic curve has no singular point [11], the genus is also equal to the number of points with integer coordinates in the interior of the convex hull (called the Newton polygon of the algebraic curve) of the set of indices $(j, k)$ where $g^j \mu^k$ has a non-zero coefficient in the polynomial $\det(M(g) - \mu\mathbf{Id})$. The presence of singular points however lowers the genus compared to the value given by the Newton polygon.

The Newton polygon allows to compute the genus easily for deformed Markov matrices with generic transition rates, whose algebraic curve does not have singular points, see for instance [442] for non-integrable variants of periodic TASEP and ASEP with generic state dependent transition rates $w_{\mathcal{C}' \leftarrow \mathcal{C}}$. Integrable models on the other hand have a very large number of singular points of high order, and the Newton polygon approach is not usable in practice. The Bethe ansatz representation of the Riemann surface however makes it possible to use the Riemann-Hurwitz formula (24) instead, see section 3.2.2 for TASEP.

### 3.1.3 Probability of the counting process

By definition of the average, the generating function of the integer counting process $Q_t$ can be written in terms of the probability of $Q_t$ as $\left\langle g^{Q_t} \right\rangle = \sum_{U \in \mathbb{Z}} \mathbb{P}(Q_t = U) g^U$. The probability can be extracted as a residue, $\mathbb{P}(Q_t = U) = \oint_\gamma \frac{\mathbf{d}g}{g^{U+1}} \left\langle g^{Q_t} \right\rangle$, with $\gamma$ a simple loop with positive orientation around $0$. The integrand is analytic everywhere except at $g = 0$ and $g = \infty$, which are both essential singularities in general. The expansion over the eigenstates of $M(g)$ of the generating function (22) then gives

$$\mathbb{P}(Q_t = U) = \oint_\gamma \frac{\mathbf{d}g}{g^{U+1}} \sum_n \frac{\sum_{\mathcal{C}} \langle \mathcal{C}|\psi_n(g)\rangle \langle \psi_n(g)|P_0\rangle \, \mathrm{e}^{t\mu_n(g)}}{\langle \psi_n(g)|\psi_n(g)\rangle} \, . \tag{25}$$

In the expression above, the eigenvalue and eigenvectors of $M(g)$ with index $n$ are equal to the corresponding meromorphic functions on $\mathcal{R}$ evaluated at the point $[g, n] \in \mathcal{R}$. Taking $\gamma$ as a small loop around $0$, the integral on $g \in \gamma$ can then be understood as the integral of a

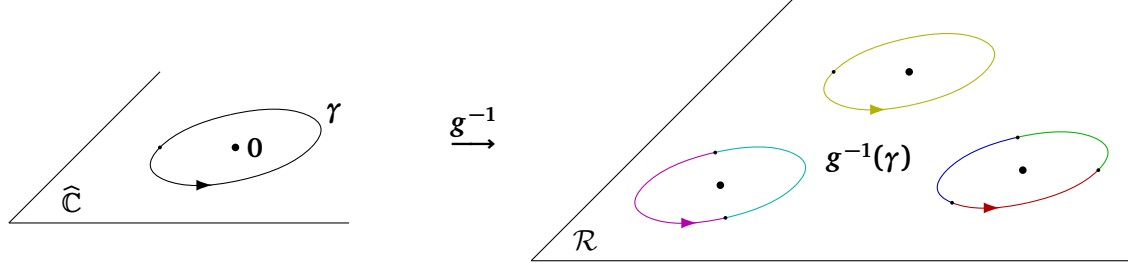

Figure 14: Lift $g^{-1}(\gamma) \subset \mathcal{R}$ of a small loop $\gamma \subset \widehat{\mathbb{C}}$ around $\mathbf{0}$. The various coloured curves on the right represent the lifts of $\gamma$ on $\mathcal{R}$. Taken together, they reconstruct loops on $\mathcal{R}$ around the points $p \in g^{-1}(\mathbf{0})$.

meromorphic differential on $\mathcal{R}$ over the path obtained by lifting $\gamma$ to $\mathcal{R}$. The lift $g^{-1}(\gamma) \subset \mathcal{R}$, which is an union of small loops around the points of $g^{-1}(\mathbf{0})$, see figure 14, can be deformed into a simple closed curve $\Gamma \subset \mathcal{R}$, such that the elements of $g^{-1}(\mathbf{0})$ are inside the curve and the elements of $g^{-1}(\infty)$ outside (with respect to positive orientation). This finally leads to

$$\mathbb{P}(Q_t = U) = \oint_\Gamma \frac{\mathrm{d}g(p)}{g(p)^{U+1}} \mathcal{N}(p) \, \mathrm{e}^{t\mu(p)} \,, \tag{26}$$

where $p$ is the current point on $\Gamma$. The function $\mathcal{N}(p) = \frac{\sum_{\mathcal{C}} \langle \mathcal{C} | \psi(p) \rangle \langle \psi(p) | P_0 \rangle}{\langle \psi(p) | \psi(p) \rangle}$, independent of the choice of normalization for the eigenvectors, is meromorphic on $\mathcal{R}$, with poles at the exceptional points $p$ where several eigenstates of $M(g(p))$ coincide. These poles however cancel in the meromorphic differential $\mathcal{N}(p)\mathrm{d}g(p)$, which has only poles in $g^{-1}(\mathbf{0}) \cup g^{-1}(\infty)$. The factor $\mathrm{e}^{t\mu(p)}$ is responsible for essential singularities at points in $g^{-1}(\mathbf{0}) \cup g^{-1}(\infty)$. Extension to joint statistics of $Q_t$ at multiple times is straightforward, and leads to a similar expression involving multiple integrals on $\mathcal{R}$ and overlaps $\langle \psi(p) | \psi(q) \rangle$ for $p, q \in \mathcal{R}$.

The function $\mathcal{N}(p)$ above is fully characterized by the locations and orders of its poles and zeroes and the normalization $\mathcal{N}(o) = 1$, with $o \in \mathcal{R}$ corresponding to the stationary state of the non-deformed Markov matrix, assumed to be unique, and such that $g(o) = 1$ and $\mu(o) = 0$. In the generic case where $\mathcal{R}$ has a single connected component, one can then write

$$\mathcal{N}(p) = \mathrm{e}^{\int_o^p \omega} \,, \tag{27}$$

where the meromorphic differential $\omega = \mathrm{d}\log\mathcal{N}$ only has simple poles with integer residues, and periods (i.e. integrals over closed curves on $\mathcal{R}$) integer multiples of $2\mathrm{i}\pi$. The locations and residues of the poles of $\omega$, which are the orders of the corresponding zeroes and poles of $\mathcal{N}$, fully characterize $\omega$.

Finding the poles of the differential $\omega$ corresponding to a given initial condition is in general a difficult problem. A partial result is available for models with generic transition rates and stationary initial condition. First, any pole $p$ of $\mathcal{N}$ corresponds to an exceptional point where $m \geq 2$ eigenstates of $M(g(p))$ coincide, the order of the pole being equal to $m - 1$. The Riemann-Hurwitz formula (24) for the function $g$ then implies that the function $\mathcal{N}$ has $2\mathfrak{g} + 2|\Omega| - 2$ poles on $\mathcal{R}$, counted with multiplicity, and thus also $2\mathfrak{g} + 2|\Omega| - 2$ zeroes. Among those, the $|\Omega| - 1$ points $p \in g^{-1}(1) \setminus \{o\}$ are double zeroes of $\mathcal{N}$ for stationary initial condition since both $\sum_{\mathcal{C}} \langle \mathcal{C} | \propto \langle \psi(o) |$ and $| P_0 \rangle \propto | \psi(o) \rangle$ are eigenvectors of $M(1)$. This leaves $2\mathfrak{g}$ unknown, extra zeroes for $\mathcal{N}$. The locations of these non-trivial zeroes can be found for simple counting processes $Q_t$. For instance, when $Q_t$ counts the number of times a specific transition $C_1 \to C_2$ takes place, or also when $Q_t$ counts the number of transitions from (or to) a specific state $C_0 \in \Omega$, it can be shown that $\mathcal{R}$ is simply the Riemann sphere $\widehat{\mathbb{C}}$, of genus zero,

and $\mathcal{N}$ does not have extra zeroes [442]. A slightly more complicated example is when $Q_t$ counts the current between to specific states $C_1$ and $C_2$. Then, $\mathcal{R}$ is a hyperelliptic Riemann surface (corresponding to an algebraic curve of the form $w^2 = P(\mu)$ with $P$ a polynomial) of genus $|\Omega| - 1$, and the non-trivial zeroes of $\mathcal{N}$ can be found explicitly [451].

For more complicated integer counting process of physical interest, an explicit characterization of the Riemann surface $\mathcal{R}$ appears to be out of reach. An exception is integrable models, where the Bethe equations fixing the allowed momenta in finite volume lead to a reasonably manageable description of $\mathcal{R}$. This is especially true for TASEP, whose Bethe equations have a mean-field structure characterizing $\mathcal{R}$ explicitly in terms of a covering map $B : \mathcal{R} \to \widehat{\mathbb{C}}$ with three branch points. Explicit expressions are in particular known for $\omega$ with simple initial states, see section 3.2.3.

## 3.2 TASEP with periodic boundaries

We consider in this section TASEP on a one-dimensional lattice of $L$ sites with periodic boundary condition, with a fixed number $N$ of particles and $|\Omega| = \binom{L}{N}$ distinct states, see section 2.2.1. The integer counting process $Q_t$ equal to the time-integrated current from site $L$ to site $1$, i.e. the number of times a particle has moved from site $L$ to site $1$ up to time $t$, is associated with a deformed generator $M(g)$.

### 3.2.1 Bethe ansatz

TASEP is known to be integrable, in the sense of quantum integrability introduced initially for quantum spin chains, and also called stochastic integrability (or integrable probability) in the context of Markov processes. Bethe ansatz [452] postulates in particular that each eigenstate of $M(g)$ is characterized by momenta $k_1, \ldots, k_N$ (not necessarily real numbers) depending on $g$.

In its coordinate form, see e.g. [141,453] for TASEP, Bethe ansatz starts with the assumption that eigenvectors are given by linear combinations of plane waves whose momenta $k_j$ are permuted, akin to elastic scattering in one dimension where conservation of energy and momentum only allows exchanges of momenta. More precisely, the component with particles at positions $x_1 < \ldots < x_N$ of a (right) eigenvector of $M(g)$ is expressed quite simply as $\langle x_1, \ldots, x_N | \psi \rangle = \sum_\sigma A_\sigma \prod_{j=1}^N e^{ik_j x_{\sigma(j)}/L}$, where the sum is over all $N!$ permutations $\sigma$ of the $N$ particles. The eigenvalue equation fully determines the coefficients $A_\sigma$ in terms of the $k_j$. Periodicity $x_j = x_j + L$ finally gives $N$ constraints on the $k_j$, called the Bethe equations, which quantize the momenta. The corresponding left eigenvector has a similar form, with the same momenta $k_j$, but with plane waves moving in the opposite direction.

A more elaborate approach builds on the fact that operators verifying the Yang-Baxter equation can be built from the local action of $M(g)$ on pairs of neighbouring sites, see e.g. [144] for TASEP. From this, an infinite hierarchy of commuting "Hamiltonians" $H_j$ containing $M(g)$ and acting locally on the lattice [454] can be constructed, as well as commuting creation operators $B(k)$ increasing the number of particles by one. The algebraic Bethe ansatz then builds the common eigenvectors of the $H_j$ by acting with the creation operators on the vector $|\emptyset\rangle$ representing an empty system, as $|\psi\rangle = B(k_1)\ldots B(k_N)|\emptyset\rangle$. The eigenvectors built in this approach are the same as the ones constructed from the coordinate Bethe ansatz above, with the same Bethe equations for the $k_j$.

In terms of the Bethe roots $y_j$ with $e^{ik_j/L} = g^{1/L}(1 - y_j)$, the Bethe equations for the deformed generator $M(g)$ of TASEP are

$$g(1 - y_j)^L + (-1)^N \prod_{k=1}^N \frac{y_j}{y_k} = 0 \,. \tag{28}$$

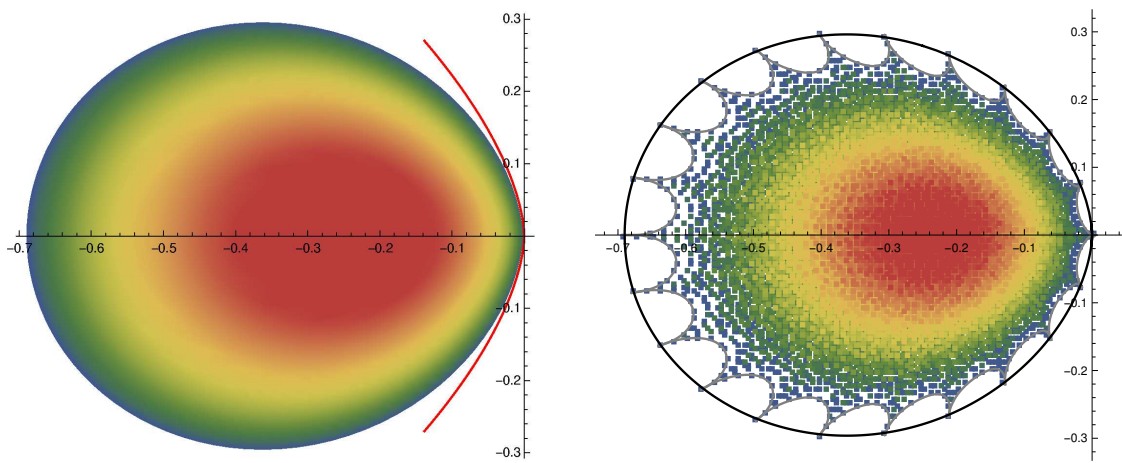

Figure 15: Density of eigenvalues $e = \mu/L$ for the Markov matrix $M = M(1)$ of TASEP at half-filling $N = L/2$, in the limit $L \to \infty$ (left) and for $L = 22$ (right). The red curve tangent to the envelope of the spectrum is $\operatorname{Re} e = -\frac{3^{7/6}(2\pi)^{2/3}}{10}|\operatorname{Im} e|^{5/3}$. At finite $L$, a microscopic structure of $L$ peaks appears for $e$ at a distance of order $1/L$ of the envelope of the spectrum. Only the rightmost peak contributes to the KPZ regime $\log g \sim L^{-1/2}$, unlike in the conformal regime of large current, see section 3.2.5. Considering instead the spectrum of $M(e^{i\theta})$ and increasing $\theta$ from $0$ to $2\pi$ induces a cyclic permutation of the peaks, in an overall rotation of the eigenvalues around the region of maximal density.

Compared to those of ASEP, see (42) below, the Bethe equations of TASEP have a mean-field structure: they decouple when considered as functions of $B = g \prod_{k=1}^{N} y_k$ rather than $g$. This crucial observation is in particular used in section 3.2.2 below to describe in a simple way the compact Riemann surface $\mathcal{R}$ associated with $M(g)$.

The Bethe equations (28) have exactly $|\Omega| = \binom{L}{N}$ solutions corresponding to actual eigenstates of $M(g)$. For large $|g|$, where all $y_j \to 1$ and $1 - y_j \simeq \omega_j g^{-1/L}$ with $\omega_j^L = (-1)^{N-1}$, the physical solutions of (28) simply correspond to all possible choices of $N$ distinct $\omega_j$ among the $L$-th roots of $(-1)^{N-1}$. The extension to finite values of $g$ follows from the interpretation of $g$ as a meromorphic function on $\mathcal{R}$, whose degree $d$ is necessarily fixed. Since the behaviour at large $g$ implies $d = |\Omega|$, the Bethe equations then have exactly $|\Omega|$ distinct solutions (away from the exceptional points $g_*$ where $M(g_*)$ is not diagonalizable and eigenstates coincide).

A given solution of the Bethe equations is associated to an eigenstate with eigenvalue

$$\mu = \sum_{j=1}^{N} \frac{y_j}{1 - y_j} \tag{29}$$

for $M(g)$. The corresponding eigenvectors on the left and on the right are obtained directly by coordinate Bethe ansatz as determinants, $\langle x_1, \ldots, x_N|\psi\rangle = \det(y_j^{-k}(1 - y_j)^{x_k})$ and $\langle\psi|x_1, \ldots, x_N\rangle = \det(y_j^k(1 - y_j)^{-x_k})$. The normalization $\langle\psi|\psi\rangle$, given rather generally for spin chains by a determinant [455, 456] when the $y_j$ are solution of the Bethe equations, reduces for TASEP to $\langle\psi|\psi\rangle = \left(\frac{L}{N}\sum_{j=1}^{N}\frac{y_j}{N+(L-N)y_j}\right)\prod_{j=1}^{N}(L - N + N/y_j)$ [386, 457]. Another general determinant formula [458] for the scalar product of two Bethe vectors with distinct sets of Bethe roots (and only one of them solution of the Bethe equations) also led to exact formulas for various overlaps between fixed vectors representing simple initial conditions and eigenstates [386, 396, 457, 459, 460].

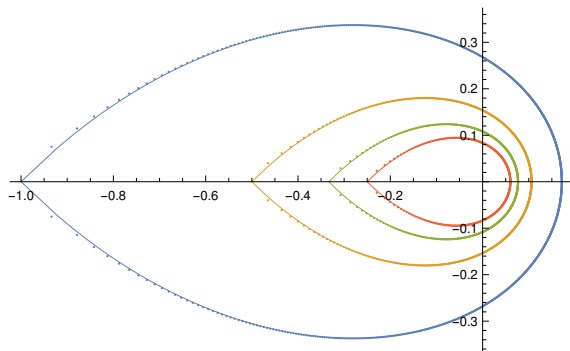

Figure 16: Curve $\mathcal{Y}_\rho$ on which the Bethe roots $y_j$ solution of the Bethe equations (28) accumulate for the eigenstates of TASEP contributing to KPZ fluctuations. The curve is plotted for density of particles $\rho = 1/2, 1/3, 1/4$ and $1/5$, starting from the outside. The small dots are corresponding Bethe roots for the stationary eigenstate of $M(g)$ with $N = \rho L$ particles on $L = 2400$ sites and $\log g = s/\sqrt{\rho(1-\rho)L}$ with $s = 1/2$.

In the limit where their number $N \to \infty$, Bethe roots generally tend to accumulate on curves in the complex plane, which do not depend on the precise eigenstate but merely on the asymptotic value of the ratio $\mu/N$. These curves can be determined from the solution of integral equations obtained by taking the logarithm of the Bethe equations and replacing sums over Bethe roots by integrals on the corresponding curve. For TASEP, (28) implies $f_L(y_j) = \frac{2i\pi n_j - \log g}{L}$ with $f_L(y) = \log \frac{1-y}{y^\rho} + \frac{1}{L}\sum_{k=1}^N \log y_k$ and $\rho = N/L$. The numbers $n_j$, which are distinct integers or half-integers depending on the parity of $N$, characterize the corresponding eigenstate. At large $L, N$ with fixed $\rho$, scaled eigenvalues $\mu/L$ given by (29) accumulate in a bounded region of the complex plane, with spikes at the boundaries (see also [461] for a similar phenomenon in some classes of random matrices), and the density of eigenvalues in the bulk of the spectrum can be extracted by a saddle point on the coarse-grained density profile of the $n_j$ [159], see figure 15. The eigenstates contributing to the KPZ regime $\log g \sim L^{-1/2}$ are characterized by $n_j = j - \frac{N+1}{2}$, except for a finite number of $n_j$ which are interpreted as particle-hole excitations over a Fermi sea in section 3.2.4 below. At leading order in $L$, the Bethe roots then accumulate on the curve $\mathcal{Y}_\rho = \{y \in \mathbb{C}, f(y) = 2i\pi u, -\rho/2 \leq u \leq \rho/2\}$, where the function $f$ is solution of $f(y) = \log \frac{1-y}{y^\rho} + \int_{\mathcal{Y}_\rho} \frac{dz}{2i\pi} f'(z) \log z$. The appropriate solution is $f(y) = \log \frac{1-y}{y^\rho} + \rho \log \rho + (1-\rho)\log(1-\rho)$, and both endpoints of the corresponding curve $\mathcal{Y}_\rho$ meet at the location $y = -\frac{\rho}{1-\rho}$, see figure 16.

Extracting the KPZ scaling limit of e.g. the generating function $\langle g^{H_{i,t}} \rangle$ of the TASEP height function $H_{i,t}$ at site $i$ and time $t$ requires, after expanding over eigenstates, the asymptotics of sums and products involving Bethe roots. Considering $y_j$ as a smooth function of $j/N$ as above, such asymptotics should in principle be computable from the Euler-Maclaurin formula $\sum_{j=1}^N \varphi\left(\frac{j+z}{N}\right) \simeq N \int_0^1 du\, \varphi(u) + \mathcal{R}_N(1,z) - \mathcal{R}_N(0,z)$, with $\mathcal{R}_N(u,z) = \sum_{k=0}^\infty \frac{B_{k+1}(z+1)\varphi^k(u)}{(k+1)!N^k}$ understood as a formal series in $N$ and $B_{k+1}$ the Bernoulli polynomials, see e.g. [462]. Since the curve $\mathcal{Y}_\rho$ is closed, the integral in the Euler-Maclaurin formula is conveniently independent of the precise equation of the curve, and in particular of $\rho$, which is expected from the universality of KPZ fluctuations. This is in contrast with the conformal regime of TASEP, where the corresponding curve is not closed and asymptotics exhibit an explicit dependency in $\rho$, see section 3.2.5.

Singularities caused by the anomalous scaling of the Bethe roots at the singular point $-\frac{\rho}{1-\rho}$ of $\mathcal{Y}_\rho$, see figure 16, require however a special treatment. Indeed, the terms $\mathcal{R}_M$ in the

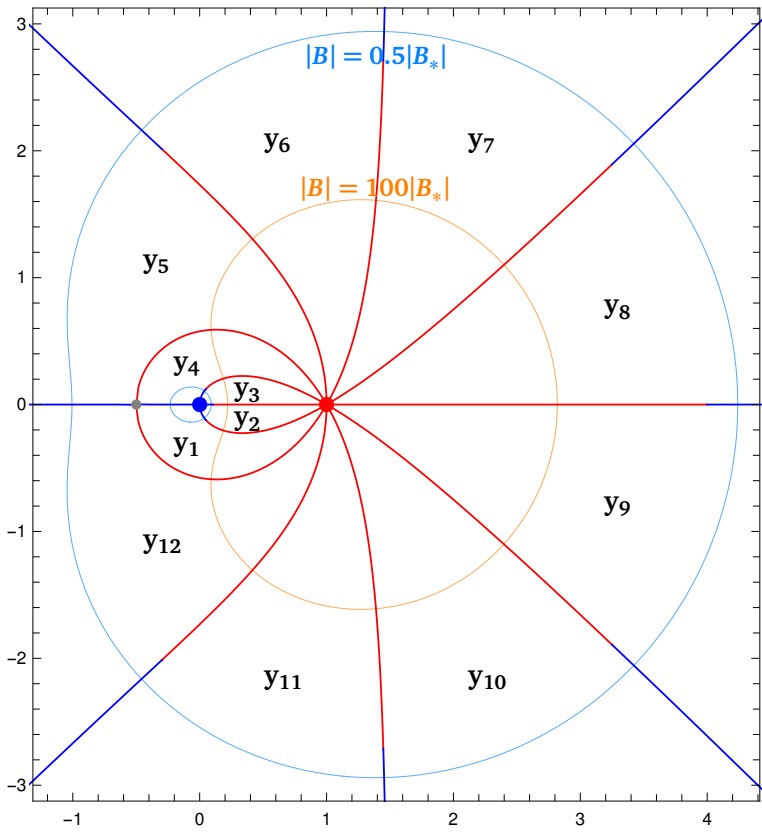

Figure 17: Domains $\mathbf{y}_j(\mathbb{C} \setminus \mathbb{R}^-)$ for the Bethe root functions $\mathbf{y}_j$ of TASEP with $N = 4$ particles on $L = 12$ sites. The thicker, red and blue curves are respectively the images of the cuts $(-\infty, B_*)$ and $(B_*, 0)$. The blue, red and smaller grey dots are respectively the images of the branch points $\mathbf{0}$, $\boldsymbol{\infty}$ and $B_*$ (the image of the branch point $\mathbf{0}$ by $\mathbf{y}_j$ is either the blue dot if $1 \leq j \leq N$ or the point at infinity if $N + 1 \leq j \leq L$). Analytic continuation along the thinner orange and light blue curves, which correspond to all possible values of the Bethe roots for some fixed values of $|B|$, realize respectively the action of the generators $\boldsymbol{a}$ and $\boldsymbol{b}$ of figure 18 as cyclic permutations of the indices of the Bethe roots.

Euler-Maclaurin formula must be replaced by contributions specific to the type of singularity. Square root singularities contribute in particular Hurwitz zeta functions with half-integer index, related to half-integer polylogarithms, and are responsible for the appearance of the function $\chi$ from (11) in various expressions. Extensions of the Euler-Maclaurin formula to double sums with various singularities on the edges are also needed for the Vandermonde determinant $\prod_{j=1}^{N} \prod_{k=j+1}^{N} (y_j - y_k)$, which appears for some overlaps. Such asymptotic computations, while doable, are excruciatingly tedious, see e.g. [393]. In the following, we exploit instead an interpretation of symmetric functions of Bethe roots as meromorphic functions on the Riemann surface $\mathcal{R}$ associated with the deformed generator $M(g)$. This allows to restrict asymptotic calculations to the meromorphic differential $\omega$ in (27), which has a simple expression for initial conditions corresponding at large $L$ to the stationary, flat and narrow wedge initial conditions for the KPZ height, see (34), (33) below.

### 3.2.2 Riemann surface for TASEP

As explained in section 3.1, each eigenstate of the deformed generators $M(g)$, $g \in \mathbb{C}$ can be viewed as a point on a Riemann surface $\mathcal{R}$, which is naturally parametrized locally (away from

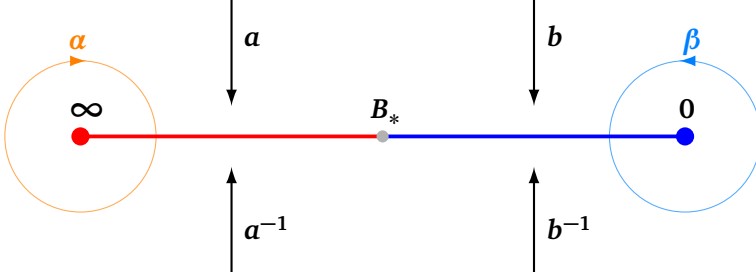

Figure 18: Generators $a$ and $b$ for analytic continuations of the Bethe root functions $y_j(B)$ across their branch cuts, and corresponding loops $\alpha$, $\beta$ in the parameter $B$. The point $B_*$ is only a branch point for the Bethe root function $y_j$ when $j \in \{1, N, N+1, L\}$. For other values of $j$, the generators $a$ and $b$ have the same action on the index $j$.

exceptional points) by the complex number $g$. In the Bethe ansatz approach with corresponding Bethe roots $y_j$ solution of the Bethe equations (28), it is more convenient to parametrize $\mathcal{R}$ in terms of the variable

$$B = g \prod_{k=1}^{N} y_k \,, \tag{30}$$

which is shown below to be a covering map from $\mathcal{R}$ to $\widehat{\mathbb{C}}$ with only three branch points, associated with a completely explicit labelling scheme for the corresponding partition of $\mathcal{R}$ into sheets.

In terms of the variable $B$, the Bethe equations decouple, and each Bethe root $y_j$ is solution of the polynomial equation

$$B(1 - y_j)^L + (-1)^N y_j^N = 0 \tag{31}$$

in the two variables $y_j$ and $B$. This allows to define $L$ distinct Bethe root functions $y_j(B)$ which are analytic for $B \in \mathbb{C} \setminus \mathbb{R}^-$ and have the three branch points $0$, $B_* = -N^N(L-N)^{L-N}/L^L$ and $\infty$.

For any value of $B \notin \{0, B_*, \infty\}$, each eigenstate corresponds to a choice of $N$ distinct Bethe roots among $\{y_j(B), j = 1, \ldots, L\}$. The Riemann surface $\mathcal{R}$ is thus made of $\binom{L}{N}$ sheets $\mathbb{C}_J$ corresponding to sets $J \subset [\![1, L]\!]$ with $|J| = N$ elements, and the points of the sheet $\mathbb{C}_J$ may be written as $p = [B, J]$. The sheets are glued together along the cuts $(-\infty, B_*)$ and $(B_*, 0)$. For the convenient labelling of the functions $y_k(B)$ in figure 17, the analytic continuation operator $a$ (respectively $b$) acting on the sets $J$ and represented in figure 18 generates the cyclic permutation $1 \to 2 \to \ldots \to L \to 1$ (resp. the product of cyclic permutations $1 \to 2 \to \ldots \to N \to 1$ and $N+1 \to N+2 \to \ldots \to L \to N+1$) of the elements of $J$, see figures 19 and 20 for graphical representation of how the sheets are connected in two examples.

The topology of the Riemann surface $\mathcal{R}$ follows from the group generated by $a$ and $b$, whose properties depend crucially on the common divisors of $L$ and $N$. In particular, $\mathcal{R}$ has a single connected component if and only if $L$ and $N$ are co-prime. The genus $\mathfrak{g}$ of $\mathcal{R}$ can be computed using the Riemann-Hurwitz formula (24) from the knowledge of the ramification indices of the covering map $[B, J] \mapsto B$ from $\mathcal{R}$ to $\widehat{\mathbb{C}}$. The genus grows exponentially fast with the system size $L$ at fixed density of particles $\rho = N/L$, and one has $\mathfrak{g} \simeq \frac{\rho(1-\rho)}{2} |\Omega|$ when $L$ and $N$ are co-prime, with $|\Omega| = \binom{L}{N}$ the total number of states. Incidentally, this value of the genus is much smaller than for non-integrable models with generic state dependent transition rates, where $\mathfrak{g} \simeq \frac{1}{2L} |\Omega|^2$, indicating the existence of a large number of singular points for the algebraic curve $\det(M(g) - \mu \mathrm{Id}) = 0$ in the integrable case [442]. The singular points are in

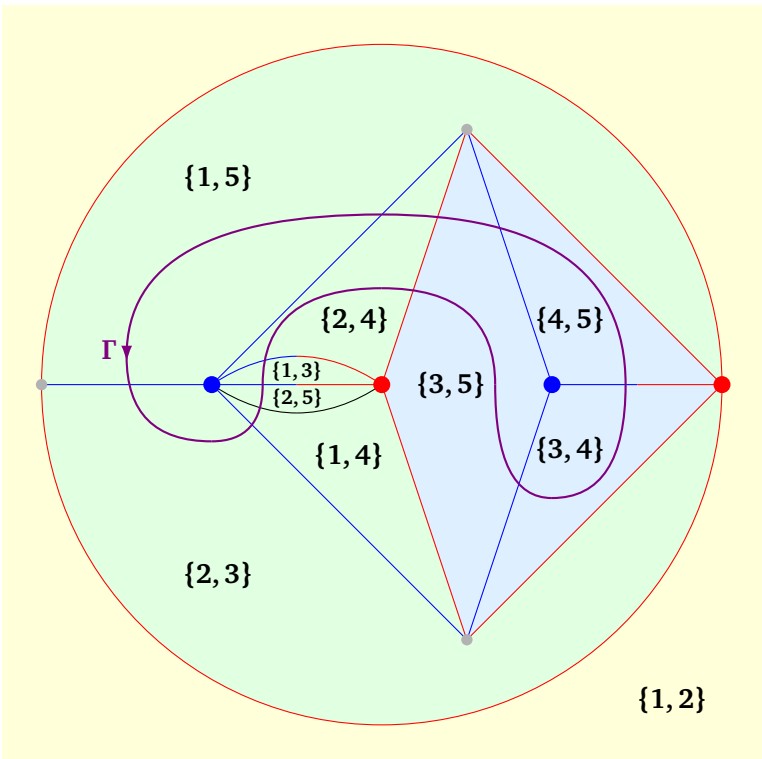

Figure 19: Schematic representation (with angles not preserved) of the connectivity of the sheets $\mathbb{C}_J$ of the Riemann surface $\mathcal{R}$ for TASEP with $N = 2$ particles on $L = 5$ sites. The sheets are labelled by the sets $J$, and their colour depends on the number of elements of $J \cap [\![1, N]\!]$. The blue, red and smaller grey dots are respectively the ramification points of the form $[0, J]$, $[\infty, J]$ and $[B_*, J]$ for the covering map $B$. The blue (respectively red) dots are also the points of $\mathcal{R}$ with $g = 0$ (resp. $g = \infty$). The purple contour $\Gamma = \beta^6 \alpha^{-1} \beta^3 \alpha \{2, 4\}$ is suitable for equation (32). Adding the point at infinity on the diagram, the Riemann surface $\mathcal{R}$ is isomorphic to the Riemann sphere $\widehat{\mathbb{C}}$. This is in agreement with the Riemann-Hurwitz formula (24) for $B$, which outputs the genus $\mathfrak{g} = 0$ since $\mathcal{R}$ has a single connected component, $B$ has degree $d = \binom{L}{N} = 10$, the three grey dots have ramification index $2$, and the blue and red dots ramification indices are respectively $6$, $5$, $3$, $5$ from left to right.

particular responsible for the existence of several connected components for $\mathcal{R}$ when $L$ and $N$ are not co-prime, and for the spectral degeneracies observed in [463, 464] at $g = 1$.

The existence of a covering map $B$ from $\mathcal{R}$ to $\widehat{\mathbb{C}}$ with only three branch points ($0, B_*, \infty$, or more conventionally $0, 1, \infty$ if one considers instead $B/B_*$) implies that $\mathcal{R}$ is a Belyi surface (and $B$ a corresponding Belyi function), see e.g. [465]. From Belyi's theorem, a Belyi surface may be represented as a non-singular algebraic curve with coefficients that are algebraic numbers (for $\mathcal{R}$, this algebraic curve is not $\det(M(g) - \mu\mathrm{Id}) = 0$ since that one is singular, but another one representing the same Riemann surface) and Belyi surfaces are thus dense in the moduli space of all compact Riemann surfaces. Belyi surfaces have a graphical characterization in terms of "dessins d'enfant", which are essentially the ribbon graphs (also called fat graphs, which are graphs endowed with a prescribed cyclic order for the edges connected to each vertex) made by the red (or blue) edges in figures 19 and 20. Additionally, Belyi surfaces $\mathcal{R}$ have the nice property that their conformal embedding into the Poincaré disk with constant negative curvature (for genus $\mathfrak{g} \geq 2$, after cutting $\mathcal{R}$ along $2\mathfrak{g}$ closed curves intersecting at the same point) can be realized explicitly in terms of geodesic triangles [466–468].

### 3.2.3   Exact formula for the probability of the height

Following section 3.1, we can now write down an expression for the probability of the current of TASEP with periodic boundaries in terms of contour integrals on the Riemann surface $\mathcal{R}$, respectively (32) and (36) below for the fluctuations at one-point and for the joint statistics at multiple times. Alternative expressions not using Riemann surfaces, but proved rigorously unlike (32) and (36), can be found in [241, 401, 407–409].

We switch to height variables instead of currents in the following in order to keep track in a convenient way of positions on the lattice. We define the height function $H_{i,t}$ at site $i$ and time $t$ as in section 2.2.1, see also figure 2. For deterministic initial condition, the height is equal at time $t = 0$ to $H_{i,0} = \sum_{k=1}^{i}(N/L - n_k)$ with $n_k = 1$ (respectively $n_k = 0$) if site $k$ is occupied (respectively empty), and random initial conditions lead to an extra summation with appropriate weights over the allowed states. The height $H_{i,t}$ then increases by $1$ each time a particle moves from site $i$ to $i + 1$. The height increment $H_{i,t} - H_{i,0}$ is thus equal to the time-integrated current $Q_{i,t}$ from site $i$ to site $i+1$, which can be deduced from the current $Q_t = Q_{L,t}$ by translation invariance after suitably shifting the initial condition. In terms of the deformed generators $M_i(g)$ for the counting processes $Q_{i,t}$, translation invariance results from similarity transformations relating $M_i(g)$ to $M_{\mathrm{tot}}(g^{1/L})$, with $M_{\mathrm{tot}}(g)$ the deformed generator for the total time-integrated current anywhere in the system. All the counting processes $Q_{i,t}$ are thus associated with the same Riemann surface $\mathcal{R}$, which is the one constructed by Bethe ansatz in the previous section. Additionally, correlations between several sites can also be studied by considering a deformed generator $M(g_1, \ldots, g_L)$ with $g_i$ conjugate to the current between sites $i$ and $i + 1$, which is related to $M_{\mathrm{tot}}(g_1^{1/L} \times \ldots \times g_L^{1/L})$ by a similarity transformation and thus also leads to the same Riemann surface $\mathcal{R}$.

From the discussion in section 3.1.3, the probability of the height can be computed as an integral on a simple closed curve $\Gamma$ on $\mathcal{R}$ such that the points $p \in \mathcal{R}$ with $g(p) = 0$ (respectively $g(p) = \infty$) are inside (resp. outside) the curve. Since a good parametrization of $\mathcal{R}$ is in terms of $B$ rather than $g$, it is convenient to have $\mathrm{d}B/B$ as prefactor of the exponential (27) rather than $\mathrm{d}g/g$ as in (26). This creates an extra zero at $p = o$, which can be removed by considering the cumulative distribution of the height instead. Taking $H \in H_{i,0} + \mathbb{N}$, one finally finds [411]

$$\mathbb{P}(H_{i,t} \geq H) = \oint_{\Gamma} \frac{\mathrm{d}B}{2\mathrm{i}\pi B} \, e^{\int_o^p \left(t \, \mathrm{d}\mu - H \frac{\mathrm{d}g}{g} + \omega\right)}, \tag{32}$$

under the condition that $L$ and $N$ are co-prime so that $\mathcal{R}$ has a single connected component.

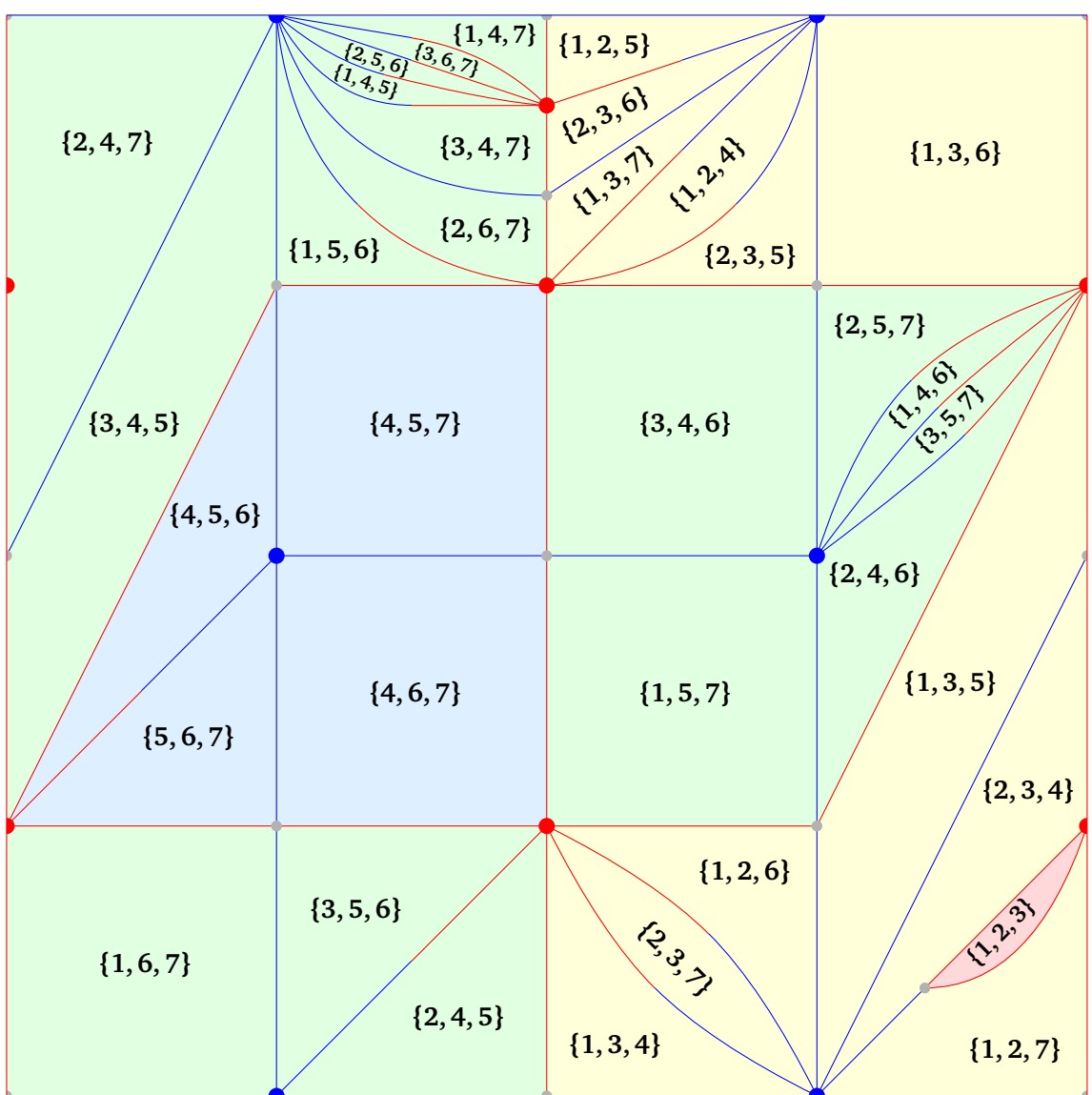

Figure 20: Schematic representation of the connectivity of the sheets $\mathbb{C}_J$ of the Riemann surface $\mathcal{R}$ for TASEP with $N = 3$ particles on $L = 7$ sites, with the same conventions as in figure 19. Opposite sides are identified and $\mathcal{R}$ is thus a torus, of genus $\mathfrak{g} = 1$. This is in agreement with the Riemann-Hurwitz formula (24) for $B$, with $K = 1$, $d = \binom{L}{N} = 35$, and ramification indices **2** for the ten grey dots, **7** for the five red dots and **4**, **6**, **12**, **12** for the blue dots.

The points of $\mathcal{R}$ with $B = 0$ (respectively $B = \infty$) lie inside (resp. outside) $\Gamma$ with respect to positive orientation, see figure 21. The stationary point $o = [0, [\![1, N]\!]] \in \mathcal{R}$ is inside $\Gamma$, and $p = [B, J]$ is the current point on $\Gamma$. The dependency on the position $i$ in (32) is hidden in the fact that $H_{i,0} - Ni/L$ is an integer and not $H_{i,0}$. In terms of the meromorphic function $\mathcal{N}$ from section 3.1.3, one has $\omega = \mathrm{d}\log(\frac{\kappa g}{g-1}\mathcal{N})$ here, with

$$\kappa = \frac{\mathrm{d}\log g}{\mathrm{d}\log B} = \frac{L}{N}\sum_j \frac{y_j}{N + (L-N)y_j}\,, \tag{33}$$

which differs slightly from the definition of $\omega$ used in (27).

The meromorphic differential $\omega$ in (32) depends on the initial condition. For stationary initial condition (with height $H_{i,0}$ built from a random configuration chosen according to the stationary state of TASEP, i.e. uniformly as a consequence of pairwise balance [469]), flat initial condition (even sites occupied, with $L = 2N$; only the connected component of $\mathcal{R}$ containing $o$ contributes) and domain wall initial condition (dw for short, with sites $L - N + 1$ through $L$ occupied), one has in particular the very explicit expressions

$$\begin{aligned}
\omega_{\mathrm{stat}} &= \Big(\frac{N(L-N)}{L}\kappa^2 + \frac{\kappa}{1-g^{-1}} - 1\Big)\frac{\mathrm{d}B}{B} \\
\omega_{\mathrm{flat}} &= \Big(\frac{L}{8}\kappa^2 + \frac{\kappa}{2} - \frac{1/4}{1 + 2^{-L}B^{-1}}\Big)\frac{\mathrm{d}B}{B} \\
\omega_{\mathrm{dw}} &= \frac{N(L-N)}{L}\kappa^2\,\frac{\mathrm{d}B}{B}\,.
\end{aligned} \tag{34}$$

For arbitrary deterministic initial condition with particles at positions $1 \leq x_1^{(0)} < \ldots < x_N^{(0)} \leq L$, Bethe ansatz gives more generally $\omega = \omega_{\mathrm{dw}} + \omega_0$ with

$$\omega_0 = \mathrm{d}\log\Big(\frac{\det(y_j^{k-1}(1-y_j)^{L-x_k^{(0)}})}{\prod_{j=1}^N\prod_{k=j+1}^N(y_k - y_j)}\Big) \tag{35}$$

Random initial conditions involve a summation over the positions inside the logarithm in (35). A characterization of the $|\Omega|$-dimensional space of meromorphic differentials $\omega$ corresponding to valid initial states $|P_0\rangle$, with in particular precise constraints on the number of poles, their locations and residues, might be helpful for the KPZ scaling limit in order to go beyond the initial conditions (34).

The expression (32) can be extended to joint statistics of $H_{i,t}$ at $n$ times $t_1 < \ldots < t_n$ and positions $i_1, \ldots, i_n$. Taking again $L$ and $N$ co-prime and $H_\ell \in H_{i_\ell,0} + \mathbb{N}$, the probability $\mathbb{P}(H_{i_1,t_1} \geq H_1, \ldots, H_{i_n,t_n} \geq H_n)$ is equal to [411]

$$\oint_{\Gamma_1}\frac{\mathrm{d}B_1}{2\mathrm{i}\pi B_1}\cdots\oint_{\Gamma_n}\frac{\mathrm{d}B_1}{2\mathrm{i}\pi B_n}\,\frac{\prod_{\ell=1}^n e^{\int_o^{p_\ell}\big((t_\ell - t_{\ell-1})\mathrm{d}\mu - (H_{i_\ell} - H_{i_{\ell-1}})\frac{\mathrm{d}g}{g} + \frac{N(L-N)}{L}\kappa^2\frac{\mathrm{d}B}{B} + \delta_{\ell,1}\omega_0\big)}}{\prod_{\ell=1}^{n-1}\Big(\big(1 - \frac{B_{\ell+1}}{B_\ell}\big)e^{\frac{N(L-N)}{L}\int_{\gamma_\ell}\frac{\mathrm{d}u}{u}\kappa(q_\ell)\kappa(q_{\ell+1})}\Big)}\,, \tag{36}$$

with $t_0 = H_0 = 0$, and $\omega_0$ as in (35). The contours $\Gamma_\ell$ are nested as in figure 21 and the point $p_\ell = [B_\ell, J_\ell]$ is the current point on $\Gamma_\ell$. The path $\gamma_\ell$ from $0$ to $1$ is such that $(q_\ell, q_{\ell+1}) = ([uB_\ell, \cdot], [uB_{\ell+1}, \cdot])$ goes from $(o, o)$ to $(p_\ell, p_{\ell+1})$ and thus belongs to the fibre product $\mathcal{R}\times_B\mathcal{R}$, which is a connected space here. Interestingly, the integral over $\gamma_\ell$ leads to the cancellation of the pole at $B_\ell = B_{\ell+1}$ if $J_\ell \neq J_{\ell+1}$, leaving only poles at $p_\ell = p_{\ell+1}$.

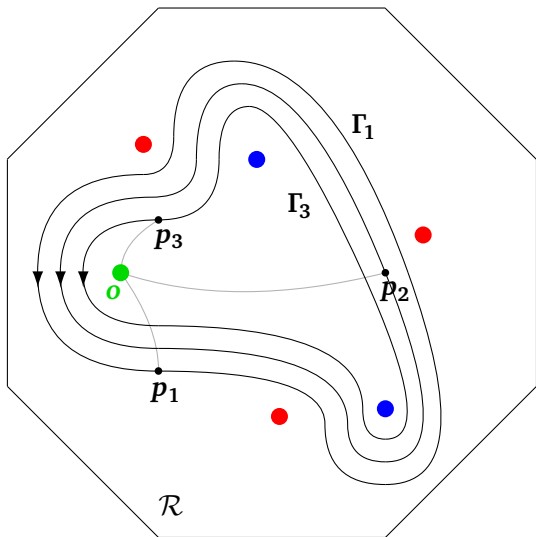

Figure 21: Simple contours $\Gamma_1 = \Gamma$ as in (32) and $\Gamma_1$, $\Gamma_2$, $\Gamma_3$ as in (36) for the (joint) probability of the TASEP height. The curves belong to the Riemann surface $\mathcal{R}$ of TASEP, represented here as a planar domain with $4\mathfrak{g}$ boundaries obtained after cutting $\mathcal{R}$ along $2\mathfrak{g}$ curves intersecting at the same point, with $\mathfrak{g}$ the genus of $\mathcal{R}$. The green dot is the stationary point $o \in \mathcal{R}$, while the blue and red dots are respectively the points $p \in \mathcal{R} \setminus \{o\}$ with $B(p) = 0$ (equivalent to $g(p) = 0$ when $p \neq o$) and $B(p) = \infty$ (equivalent to $g(p) = \infty$). The grey curve between $o$ and the current point $p_\ell \in \gamma_\ell$ represents the path for the integral inside the exponential in the numerator of (32), (36).

### 3.2.4 KPZ scaling limit

We would finally like to take the KPZ scaling limit, as defined in section 2.2.1, of the probability of the TASEP height $H_{i,t}$ in (32), (36). For the stationary, flat and domain wall initial conditions in (34), this leads respectively to the distribution of the height $h(x, \tau)$ at the KPZ fixed point with stationary, flat and narrow wedge initial conditions, see section 2.1.6.

In the expression (32) for $\mathbb{P}(H_{i,t} = H)$, asymptotics require choosing a contour $\Gamma$ such that $B \sim B_*$ when $L \to \infty$, and similarly in (36) for the joint statistics. In order to achieve this, we either take $|B| < |B_*|$ and split $\Gamma$ into simple loops with positive orientation around the points $B^{-1}(0)$ or take $|B| > |B_*|$ and split $\Gamma$ into simple loops with negative orientation around the points $B^{-1}(\infty)$. These loops span all the sheets $\mathbb{C}_J$ of $\mathcal{R}$ for the variable $B$. Among them, the only sheets $\mathbb{C}_J$ contributing to (32) in the KPZ scaling limit are the ones that are "close" to the sheet $J_0 = [\![ 1, N ]\!]$ containing $o$, i.e. such that for large $L$, $N$, the set $J$ can be built from $J_0$ by removing a finite number of elements of $J_0$ within a finite distance of $1$ and placing them at a finite distance of $1$ (modulo $L$), and similarly for elements of $J_0$ close to $N$. Such sets $J$ have the form of independent particle-hole excitations at both sides of the Fermi sea representing the set $J_0$, and are characterized by two finite sets of half-integers $P$ and $H$, see figure 9, representing respectively momenta of particles outside of the Fermi sea and holes inside the Fermi sea. The quantity $p_{P,H} = \sum_{a \in P} a - \sum_{a \in H} a$ corresponds in particular to the total momentum of the particle-hole excitations.

Writing $B/B_* = -e^\nu$ and $p = [B, J] \in \mathcal{R}$ with $J$ corresponding to a particle-hole excitation specified by sets $P$ and $H$, the Euler-Maclaurin formula with half-integer power singularities

at both ends eventually gives the asymptotics [392]

$$
\begin{aligned}
\kappa(p) &\simeq \frac{\chi''_{P,H}(v)}{\sqrt{\rho(1-\rho)L}} \\
\log g(p) &\simeq \frac{\chi'_{P,H}(v)}{\sqrt{\rho(1-\rho)L}} \\
\mu(p) - \rho(1-\rho)\log g(p) &\simeq -2\mathrm{i}\pi(1-2\rho)p_{P,H} + \sqrt{\rho(1-\rho)}\frac{\chi_{P,H}(v)}{L^{3/2}} \,,
\end{aligned}
\tag{37}
$$

respectively for the function $\kappa$ defined in (33), the eigenvalue (29) of the deformed generator $M(g)$, and the function $g$. The function $\chi_{P,H}$, defined in (20), is analytic outside the branch cut $\mathrm{i}(-\infty, -\pi] \cup \mathrm{i}[\pi, \infty)$.

Inserting the asymptotics (37) into (32) gives the probability of the KPZ height $h(x,t)$, related to $H_{i,t}$ by (6). In particular, the terms $\rho(1-\rho)\log g(p)$ and $-2\mathrm{i}\pi(1-2\rho)p_{P,H}$ in the asymptotics of the eigenvalue $\mu(p)$ are respectively related to the term $\rho(1-\rho)t$ in (6) and the fact that KPZ fluctuations are observed in a moving reference frame with velocity $1-2\rho$. At the level of the generating function of the height, the asymptotics for $g$ gives in particular the parametric expression $e_n(s) = \chi_{P,H}(v)$ with $\chi'_{P,H}(v) = s$ for the eigenvalues $e_n(s)$ in (19).

The complete asymptotics for the probability of the height (32) or (36) requires some extra care, due to the fact that the compact Riemann surface $\mathcal{R}$ splits under KPZ scaling into infinitely many connected components $\mathcal{R}^{\Delta}_{\mathrm{KPZ}}$ (collecting all the sheets indexed by sets $P$ and $H$ with the same symmetric difference $\Delta = P \ominus H = (P \cup H) \setminus (P \cap H)$), and the probability of the height is then expressed as a sum over finite subsets $\Delta \subset \mathbb{Z} + 1/2$ of the contribution of the connected components (except for flat initial condition, where only the connected component $\mathcal{R}^{\emptyset}_{\mathrm{KPZ}}$ containing the stationary point contributes). In particular, the path of integration in (32) between the point $o$ and a point $p \in \Gamma$ does not exist if $p \notin \mathcal{R}^{\emptyset}_{\mathrm{KPZ}}$. For the one-point statistics, the limit of the integral from $o$ to a point that belongs to another connected component when $L \to \infty$ is responsible for an extra coefficient in the sum over the connected components of $\mathcal{R}_{\mathrm{KPZ}}$, equal to $(\mathrm{i}/4)^{|\Delta|} \sum_{A \subset \Delta, |A| = |\Delta \setminus A|} \mathrm{e}^{2\mathrm{i}\pi x(\sum_{a \in A} a - \sum_{a \in \Delta \setminus A} a)} V_A^2 V_{\Delta \setminus A}^2$ with $V_S = \prod_{a > b \in S}(a-b)$. This coefficient contains in particular all the dependency in the position $x$ of the probability of $h(x,t)$ for stationary and narrow wedge initial condition.

### 3.2.5 KPZ scaling vs conformal invariance

The KPZ regime of TASEP corresponds to typical fluctuations of $H_{i,t}$ with amplitude of order $L^{1/2}$ as in (6), characterized by the generating function $\langle g^{Q_t} \rangle$ with $\log g \sim L^{-1/2}$. Scalings $|\log g| \gg L^{-1/2}$ then correspond to rare events for $H_{i,t}$, whose properties depend on the sign of $\log g$. For finite $\log g$, this signals the presence of a dynamical phase transition at $\log g = 0$, for which the KPZ scale acts as a crossover when zooming to the region $\log g \sim L^{-1/2}$. The case $\log g < 0$, which corresponds to conditioning the dynamics to smaller than typical values of the current, leads to a phase-separated regime with regions of low and high density of particles [175] where anti-shocks become stable unlike in the non-conditioned dynamics [470].

On the other hand, for $\log g > 0$, an effective long range repulsive interaction keeps the particles as far away as each other as possible in order to maximize the current, which leads to a hyperuniform state with reduced density fluctuations [471]. In particular, in the limit $g \to +\infty$, the Bethe equations reduce to that of free fermions. The corresponding dynamics conditioned on large current is then known exactly [472,473], see also [362,366] for the case of open boundaries, and the effective potential is logarithmic at short distance. The dynamical exponent for the conditioned dynamics is equal to $z = 1$, corresponding to ballistic propagation where space $x$ and time $t$ play essentially the same role, in contrast with the KPZ regime whose

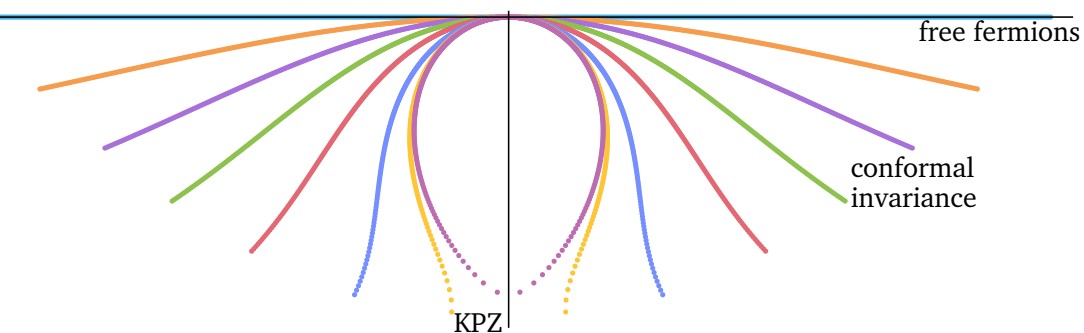

Figure 22: Representation in the complex plane of the momenta $k_j$ for the stationary eigenstate of TASEP with a large system size, for several values of $g > 1$ spanning the crossover between the conformal and KPZ regimes. The curves were shifted vertically for clarity. Deep within the conformal regime, the momenta are real and equally spaced, forming a "string". In the KPZ regime, the momenta are complex valued, and the spacing between neighbouring momenta has the anomalous scaling $L^{-1/2}$ instead of $L^{-1}$ at the edge.

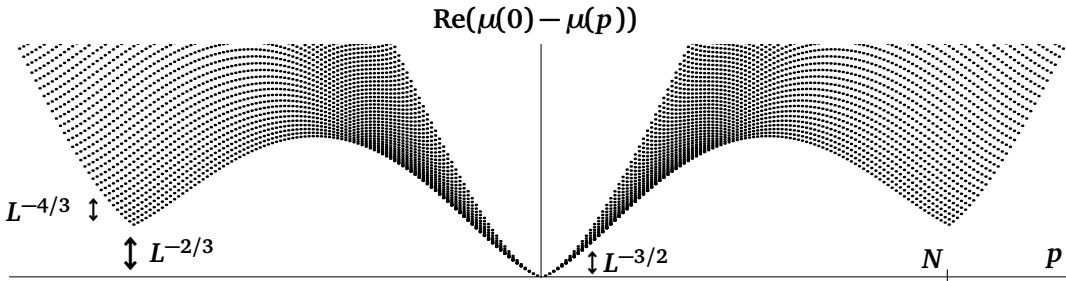

Figure 23: One particle-hole spectrum of TASEP in the KPZ regime $\log g \sim L^{-1/2}$. The one particle-hole spectrum corresponds to the eigenstates of $M(g)$ indexed by sets of the form $J = (\llbracket 1, N \rrbracket \setminus \{j\}) \cup \{j + p\}$ in the Bethe ansatz picture, and representing a single particle-hole excitation of total momentum $p$ moving the integer $j$ from the Fermi sea to $j + p$ outside of the Fermi sea. The eigenvalues corresponding to several particle-hole excitations fill the spectrum above and lead to secondary peaks for $p$ integer multiples of $N$, corresponding to the peaks on the right of the full spectrum in figure 15.

dynamical exponent $z = 3/2$ signals an anisotropy in the $(x, t)$ variables. It has been shown that this regime exhibits conformal invariance with central charge $c = 1$ [436, 474].

The crossover between the KPZ and conformally invariant regimes is visible on the shape of the curve on which the Bethe roots accumulate for the eigenstates with smallest spectral gaps, see figure 22. For $\log g \to \infty$, the $y_j$ accumulate on an arc of circle with centre $1$, corresponding for the momenta $k_j$, $e^{ik_j/L} = g^{1/L}(1 - y_j)$, to a string with all $k_j$ real and equally spaced. In the KPZ regime, on the other hand, the string becomes completely distorted into a closed curve with a singular point. Away from this singular point, the distance between consecutive $y_j$ scales as $L^{-1}$. At the singular point, these distances scale instead as $L^{-1/2}$. This mismatch between the bulk and edge scalings requires the use of singular summation formulas for the calculation of asymptotics of sums needed for eigenvalues and overlaps of eigenvectors, see at the end of section 3.2.1.

The spectrum of $M(g)$ can in fact be computed for large but finite $\log g$, in the regime $\log g = \gamma L, \gamma > 0$. With density $\rho = N/L$, the largest eigenvalue has the parametric expression [356] $\mu_0(\gamma) \simeq -\rho(1 - \rho)LH(B)$ with $H(B) = \sum_{m=-1}^{\infty} \binom{\rho m+1}{m+1} \frac{\mathrm{sinc}(m\pi\rho)}{\rho m+1} B^m$ and $B = -\mathrm{e}^{-b}$

solution of $b - \gamma = F(B)$ with $F(B) = \sum_{m=1}^{\infty} \binom{\rho m+1}{m+1} \frac{(m+1)\operatorname{sinc}(m\pi\rho)}{m(\rho m+1)} B^m$. Higher excited states are again given in terms of particle-hole excitations at the edges of a Fermi sea as in figure 9, except that the cardinals of the sets $P$ and $H$ are only required to satisfy $|P| = |H|$ (conservation of the total number of particles and holes), with arbitrary $\Pi = |P_+| - |H_-| = |H_+| - |P_-| \in \mathbb{Z}$. Excitations from one side to the other side of the Fermi sea are thus now allowed, unlike in the KPZ regime where only the eigenstates with $\Pi = 0$ contribute. Straightforward calculations lead to

$$-\frac{\mu(\gamma + \frac{2i\pi\Pi}{L})}{\rho(1-\rho)} \simeq LH(B) + \frac{1}{L}\left(\frac{\pi^2}{6} - \pi p + 2\pi^2\Pi^2 + \dots\right)\frac{\partial_\gamma^2 H(B)}{(1+\partial_\gamma F(B))^2}, \qquad (38)$$

with $B$ related to $\gamma$ as above and $p = p_{\mathrm{R}} - p_{\mathrm{L}}$ the total momentum of particle-hole excitations, with $p_{\mathrm{R}} = 2\pi(\sum_{a\in P_+} a - \sum_{a\in H_-} a)$, $p_{\mathrm{L}} = 2\pi(\sum_{a\in P_-} a - \sum_{a\in H_+} a)$ the momenta on each side of the Fermi sea. The purely imaginary extra term $\dots$, proportional to $p_{\mathrm{R}} + p_{\mathrm{L}}$, involves analogues of $H(B)$ and $F(B)$ with $\sin(m\pi\rho)$ replaced by $\cos(m\pi\rho)$ and vanishes at half-filling $\rho = 1/2$. The factor $(\partial_\gamma^2 H(B))/(1 + \partial_\gamma F(B))^2$ in (38) is essentially the Fermi velocity in the Luttinger liquid picture, see below, and the central charge $c = 1$ can be read from the term $\pi^2/6$.

The $1/L$ correction to the eigenvalues in (38) is quadratic in the momentum transfer $\Pi$ between both sides of the Fermi sea, and linear in the momenta $p_{\mathrm{R}}$ and $p_{\mathrm{L}}$. This is consistent with the Luttinger liquid paradigm (in a non-Hermitian setting here) from condensed matter theory, whose correlations exhibit conformal invariance, see e.g. [475], and also [476] in relation with interface growth. In particular, the same scaling $1/L$ is found for spectral gaps $\mathrm{Re}(\mu_0 - \mu)$ corresponding to particle-hole excitations close to the edges of the Fermi sea, but also for excitations with $\Pi = 0$ corresponding to exchanges between both sides of the Fermi sea (called Umklapp excitations). In the KPZ regime, spectral gaps scale differently, see figure 23, as $L^{-3/2}$ for excitations close to the edges of the Fermi sea, which are the only ones contributing to height fluctuations on the KPZ time scale $t \sim L^{3/2}$, and as $\Pi^{5/3}L^{-2/3}$ [159] (instead of $\Pi^2/L$ in the conformal regime) for Umklapp excitations, which are indeed suppressed on the KPZ time scale.

Working out the correspondence to the Luttinger liquid directly at the level of operators and not just for the spectrum as in (38) would be interesting in order to understand more precisely the implications of conformal invariance for the large deviations of the current of TASEP on the Euler scale $t \sim L$.

## 3.3 Possible extensions

Topological concepts have been used increasingly in the past decades in physics. The approach described above to KPZ fluctuations in finite volume and fluctuations of current-like observables for Markov processes, where the topology of the phase space associated with the observable is encoded in a Riemann surface, demonstrates the usefulness of topological ideas in this context, by allowing for rather elementary expressions for the statistics of fluctuations in terms of simple contour integrals on the phase space. Additionally, compared to more local approaches where the global structure of the phase space is obscured, thinking globally about the whole Riemann surface allows to use powerful tools from algebraic geometry to characterize the relevant space of functions on the phase space. We discuss in this section a few topics for which such global approaches seem particularly well suited.

As explained in section 3.1, fluctuations of current-like observables for general integer counting processes with a finite state of space can be conveniently understood in terms of a compact Riemann surface $\mathcal{R}$. Consequences of this elementary observation are still largely unexplored. In particular, a better understanding of the relationship between the physical features of the process (reversibility, ergodicity, fluctuation theorems, ...) and the complex

structure characterizing the Riemann surface would be welcome. The simple example where a single current of an underlying Markov process is monitored makes a useful toy model [451], for which $\mathcal{R}$ turns out to be hyperelliptic, and the meromorphic function $\mathcal{N}$ in (26) is fully characterized by the known location of its poles and zeroes if the system is prepared initially in its stationary state. A full characterization of the space of functions $\mathcal{N}$ corresponding to proper initial conditions is however still missing. For more general counting processes, on the other hand, to what extent explicit formulas can be expected is still unclear. Additionally, it would be interesting to explore whether standard methods for compact Riemann surfaces, in particular uniformization to a surface of constant curvature (and the associated closed geodesics), could have a meaningful interpretation in terms of counting processes, especially when thinking of disordered systems where the transition rates of the underlying Markov process are themselves random variables on which one would like to perform a further average.

Coming back to KPZ fluctuations in finite volume, a major open problem for the KPZ fixed point with periodic boundaries is understanding the precise relationship between initial conditions and meromorphic differentials $\omega$ on the non-compact Riemann surface $\mathcal{R}_{\text{KPZ}}$. Already at the level of TASEP with a finite system size $L$ and a finite number of particles $N$, where the corresponding Riemann surface $\mathcal{R}$ coming from Bethe ansatz is compact, it would be nice to determine what are the constraints for the poles of $\omega$ leading to the proper $|\Omega|$-dimensional space corresponding to initial conditions, and understand in particular how the dynamics acts on such meromorphic differentials. Another outstanding issue for the KPZ fixed point with periodic boundaries is joint statistics at a single time but multiple points in space. Although taking the limit of the known expressions for joint statistics at multiple times when all the times become equal gives in principle the desired result, this leads to an expression with multiple contour integrals on $\mathcal{R}_{\text{KPZ}}$, while a single one should be necessary according to the eigenstate expansion of the generating function. Simpler expressions for joint statistics at multiple points would in particular give a better understanding of the random process $x \mapsto h(x, t)$ and its convergence to a Brownian bridge at late times.

Extension to the KPZ fixed point with open boundaries, where the slope of the KPZ height is fixed on both ends of the system, appears to be quite natural in view of its relevance for the physical setting of driven particles in contact with reservoirs, see section 3.3.1 below. As explained throughout section 2.4, much is known already about the stationary measure, stationary large deviations and the relaxation rate to stationarity. An important missing piece is time-dependent height fluctuations, which requires not only the eigenvalues but also the eigenvectors of the generator. Unexpected recent advances on the Bethe ansatz side for TASEP with open boundaries suggest the existence of a Riemann surface defined from analytic continuation of sets of solutions of a polynomial equation, just like the one for TASEP with periodic boundaries. Numerics suggest that the similarity between periodic and open TASEP extends to some extent to the meromorphic differentials $\omega$, at least for simple initial conditions and infinite boundary slopes. Combined with an interpretation of eigenvectors as meromorphic functions on the Riemann surface, these numerical insights are expected to lead in a near future to an explicit coordinate version of Bethe ansatz for TASEP with open boundaries, which would then be a perfect starting point for a full characterization of height fluctuations for open TASEP, and eventually for KPZ fluctuations with open boundaries after asymptotic computations.

Another natural extension of the Riemann surface approach is to systems with multiple conservation laws, see section 3.3.2 below, where some eigenmodes generically exhibit KPZ fluctuations in accordance with non-linear fluctuating hydrodynamics. A first open problem is that there is still no characterization of large scale fluctuations in finite volume of the corresponding height increments in terms of Brownian-like random processes in interaction like in section 2.1.6 for a single height field with periodic or open boundaries, despite the fact that the invariant measure of some exactly solvable models in this class is known. Another out-

standing issue is stationary large deviations of the height fields, which are still largely missing. Finally, like with open boundaries, a major open problem beyond stationary fluctuations for a system with finite volume is the whole relaxation process, describing the time evolution of the eigenmodes from an initial regime where they propagate without interacting with each other, to their decay at late times when they begin to interact as they become delocalized throughout the system. Using an integrable discretization in the form of an exclusion process with multiple species, a nested version of Bethe ansatz allows to diagonalize the generator, but analytic continuation naturally appears to require higher dimensional complex manifolds instead of Riemann surfaces.

Finally, the renormalization group flow between the EW and KPZ fixed points in finite volume, characterized by the $\lambda$-dependent height field $h_\lambda(x, t)$ solution of the KPZ equation (1), is discussed in section 3.3.3 below. Unlike on the infinite line, see section 2.3.2, and for stationary large deviations, see section 2.4.1, no exact result is available so far for the time-dependent fluctuations of $h_\lambda(x, t)$. Extracting those from Bethe ansatz, either from the weakly asymmetric version of ASEP (WASEP) discussed at the end of section 2.2.1 or from the delta-Bose gas within the replica approach, see section 2.2.4, appears to be a difficult problem, and little progress has been achieved so far even with periodic boundaries. From the point of view of the Riemann surface approach, the main issue appears to be the lack of a good parametrization in terms of which the corresponding Riemann surface $\mathcal{R}^\lambda_{\text{KPZ}}$ has a manageable branching structure, in terms of which the eigenstate expansion can be written explicitly. Further studies from the perspective of analytic continuation of the solutions to the functional equations arising from Bethe ansatz could be helpful in order to understand better how the complex structure of $\mathcal{R}^\lambda_{\text{KPZ}}$ depends on $\lambda$. Additionally, elucidating the precise relationship between WASEP and the delta-Bose gas, whose functional equation describe in a simple way respectively the large and small $\lambda$ regimes, might lead to some progress.

### 3.3.1 Open boundaries

In this section, we consider TASEP with open boundaries defined as in section 2.2.2, see in particular figure 3, with boundary hopping rates $\alpha$, $\beta$ related to the boundary slopes $\sigma_a$, $\sigma_b$ of the KPZ height as in (8).

Unlike the model with periodic boundaries, stationary probabilities of open TASEP are not uniform. The stationary state has however an exact matrix product representation [181, 477]. In terms of site occupation numbers $n_i = 1$ (respectively $n_i = 0$) if site $i$ is occupied (resp. empty), the stationary probabilities are equal to $P_{\text{st}}(n_1, \ldots, n_L) = Z_L^{-1} \langle W | X_{n_1} \ldots X_{n_L} | V \rangle$, where the infinite dimensional matrices $D = X_1$ and $E = X_0$ verify the quadratic algebra $DE = D + E$ and the boundary identities $\langle W | E = \alpha^{-1} \langle W |, D | V \rangle = \beta^{-1} | V \rangle$. Various combinatorial interpretations of the stationary probabilities are known [365, 478–484]. For ASEP, when particles hop in both directions, the matrix product representation has instead the bulk algebra $DE - qED = D + E$, which can be interpreted in terms of lattice paths [485], and is related to families of $q$-orthogonal polynomials [486–489]. At large scales, typical fluctuations in the stationary state have been computed from the matrix product representation, and lead to Brownian-like processes, see section 2.1.6.

TASEP with open boundaries is integrable. A first version of Bethe equations giving the spectrum of its Markov operator was obtained in [490], using earlier results for the XXZ spin chain with special non-diagonal boundaries [491–493], see also [445, 494, 495] for a construction of the corresponding eigenvectors. These Bethe equations require special relations for non-diagonal $2 \times 2$ boundary operators, which are automatically satisfied for TASEP with $g = 1$. This led to exact expressions in the low and high density phases for the spectral gap [490, 496], see also [497, 498] for extensions to ASEP and [499] for the large deviation function of the current. In the maximal current phase, however, the intricate nature of these

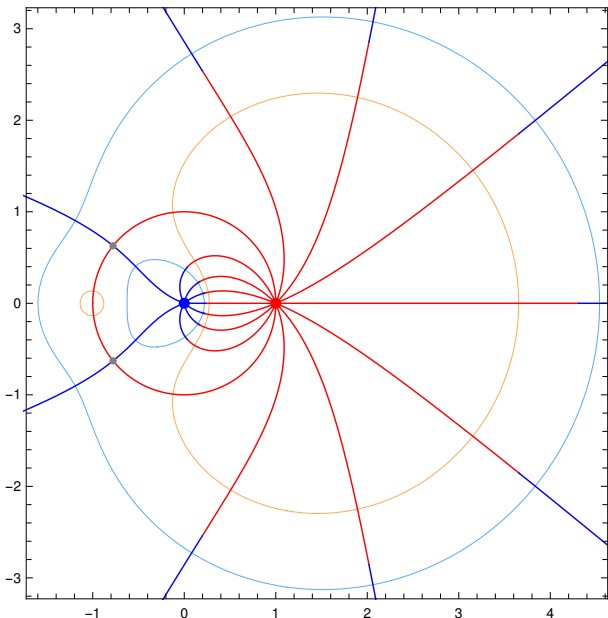

Figure 24: Domains for Bethe roots of open TASEP with $L = 7$ sites and $\alpha = \beta = 1$, computed from (39). The same conventions are used as in figure 17 for the periodic case.

Bethe equations had prevented asymptotic calculations. Then, another version of Bethe ansatz for TASEP valid for any value of the parameter $g$ was obtained in [500] from a modified version of algebraic Bethe ansatz [501, 502], where eigenstates are built by acting on a reference state with exactly $L$ creation operators, instead of an arbitrary number $N$ in the periodic case. The associated Bethe equations, which resemble the ones obtained previously in [490], were however still too intricate to make progress in the maximal current phase at large $L$.

Yet another version of TASEP Bethe equations (without Bethe ansatz, strictly speaking, since the corresponding Bethe vectors are not known) was discovered in [363] by comparison with the one from [500] at the level of Baxter's equation (see section 3.3.3 below) for small system size $L$. Surprisingly, these new Bethe equations have the same mean-field structure as the ones for periodic TASEP, compare in particular figure 24 with figure 17. The strategy detailed in section 3.2 for building the Riemann surface of periodic TASEP from Bethe ansatz can then be used again, with (31) replaced by

$$B(1 + y_j)^2(1 - y_j)^{2L+2} + (-1)^{L+1}(y_j + a)(y_j + b)(a y_j + 1)(b y_j + 1) y_j^L = 0 , \qquad (39)$$

with $a = \alpha^{-1} - 1$, $b = \beta^{-1} - 1$. Each eigenstate corresponds to a choice of $L + 2$ Bethe roots among $2L + 4$ solutions, with some additional constraints preserved by analytic continuation, and the corresponding eigenvalue of $M(g)$ (counting e.g. the current entering the system at site 1) with $g = \alpha^{-1}\beta^{-1}\prod_{j=1}^{L+2}(1 - y_j)^{-1}$ is equal to $\mu = 1 - \frac{\alpha+\beta}{2} + \frac{1}{2}\sum_{j=1}^{L+2}\frac{y_j}{1-y_j}$. A peculiar feature compared to the periodic case is that for a generic value of $B$, the number of suitable choices of Bethe roots is equal to twice the size $|\Omega| = 2^L$ of $M(g)$.

When $\alpha$ or $\beta$ is equal to $1/2$, a factor $(1 + y_j)^2$ cancels from both sides of (39). If the other boundary rate is equal to $1$, then (39) matches with (31) for a periodic system with $L + 1$ particles on $2L + 2$ sites. Additionally, one and only one of the chosen Bethe roots converges to $-1$, while the remaining $L + 1$ are equal to Bethe roots for the periodic case. Matching the eigenvalue and the parameter $g$ in both cases implies that the spectrum of $M(g)$ for the open case is a subset of the spectrum of $\frac{1}{2}M_{\text{per}}(g^2)$, with $M_{\text{per}}(g)$ the deformed generator for the

periodic case. This generalizes to the whole spectrum of open TASEP with finite $L$ the observation below (13) about stationary large deviations of the KPZ height when one boundary slope vanishes and the other one is infinite. It is however not clear what this observation implies for height statistics, in the absence of a relation between the corresponding eigenvectors (which do not have the same size).

So far, the Bethe equations (39) have been exploited in [363] to recover the stationary large deviations of the height at the KPZ fixed point with infinite boundary slopes discussed in section 2.4.1, and obtained initially from an iterated version of the matrix product representation [181] for the stationary state, both for TASEP [358,359] and ASEP [360,361]. The Bethe equations (39) were also used in [438] to compute exactly the spectral gaps for $\alpha = \beta = 1$, and a perfect match was found with the result from analytic continuation of stationary large deviations.

After the gaps, the next logical step would be to compute the full dynamics of the height statistics at the KPZ fixed point in finite volume with open boundaries. An interesting observable would in particular be the statistics of the height difference $h(1, t) - h(0, t)$, related to the total number of particles on the lattice, which is not constant unlike for periodic boundaries, and is known to exhibit interesting features [503]. A difficulty for computing the dynamics of such observables is however that the eigenvectors corresponding to the Bethe equations (39) are still missing. Working directly at the level of the Riemann surface might help, however. Indeed, preliminary numerical results [504] for TASEP with $\alpha = \beta = 1$ indicate that the meromorphic differential $\omega$ in (32) for stationary initial condition is still given by (34) except that $N(L-N)/L$ is replaced by $L + 2$ and $\kappa = \frac{d \log g}{d \log B}$ is computed from (39). Expressions similar to $\omega_{\text{dw}}$ are also conjectured from numerics for initial conditions with all sites empty or filled:

$$
\begin{aligned}
\omega_{\text{stat}} &= \left( (L+2)\kappa^2 + \frac{\kappa}{1 - g^{-1}} - 1 \right) \frac{dB}{B} \\
\omega_{\text{empty}} &= \left( (L+2)\kappa^2 - \kappa \right) \frac{dB}{B} \\
\omega_{\text{filled}} &= \left( (L+2)\kappa^2 - \frac{L+2}{2}\kappa \right) \frac{dB}{B} \; .
\end{aligned}
\tag{40}
$$

The common term $(L+2)\kappa^2$ leads in particular to a factor $\prod_{j<k} \sqrt{\frac{y_j - y_k}{1 - y_j y_k}}$ in the integrand of (32) after performing explicitly the integral in the exponential. We expect that the conjectures (40) will help finding the correct ansatz for the eigenvectors.

### 3.3.2 Several conserved quantities

Integrable extensions of TASEP featuring several types (also called species or classes) of particles exist, with a hierarchy such that particles of a lower type can overtake particles of a higher type but not the reverse, and type dependent hopping rates for the exchange of particles between two neighbouring sites. Integrable multispecies versions of ASEP with particles hopping in both directions with different rates also exist. Constraints on the hopping rates are generally needed for such models to be integrable, see e.g. [505–508].

Multispecies exclusion processes have several locally conserved quantities, the number of particles of each type, and their description at large scales features coupled equations for the corresponding fields. Non-linear fluctuating hydrodynamics [12,13] predicts various universality classes for the fluctuations of normal modes, including diffusive and KPZ fluctuations, and more generally a whole family [509] with dynamical exponent $z$ given by a ratio of two consecutive Fibonacci numbers. At the level of microscopic models, coupled Burgers' equations have been proved in some cases [510–514].

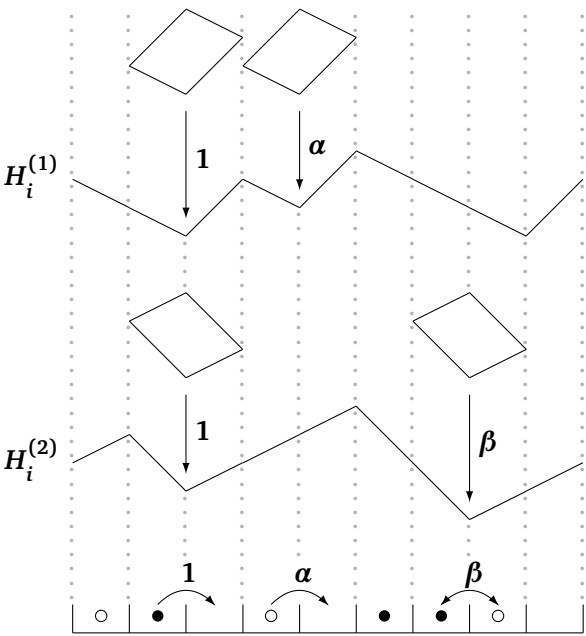

Figure 25: Height functions $H_i^{(1)}$ and $H_i^{(2)}$ for TASEP with two types of particles corresponding to the configuration of the system shown at the bottom. Particles of first type are represented by filled circles and particles of second type by empty circles.

Invariant measures for multispecies TASEP have been studied on the infinite line [515–518]. In finite volume, a matrix product representation of the stationary state similar to the one for open ASEP (with a trace instead of boundary vectors) is known for the two species case with periodic boundaries [519]. When all the hopping rates are equal, the infinite matrices $D$, $A$, $E$ corresponding respectively to particles of first, second type and empty sites verify the quadratic algebra $DE = D + E$, $DA = A$, $AE = E$, see also [517] for a combinatorial interpretation of the stationary probabilities. Tensor products of $D$, $A$ and $E$ are required to represent as a matrix product the stationary probabilities with a higher number of species [520–523], see also [518, 524–526] for a related multiline iterative construction, [527–529] for various combinatorial interpretations, [530–532] for relations with orthogonal polynomials, [533] for an alternative representation in terms of transfer operators satisfying a higher dimensional generalization of the Yang-Baxter equation, [534–539] for extensions to open boundaries and [540–544] for type-dependent hopping rates. Phase diagrams have been obtained for finite densities of particles of each type [535, 545–548], and various correlation functions [519, 549–551] have been computed in the stationary state. A complete characterization of stationary fluctuations in terms of Brownian-like paths, as in section 2.1.6 for the single species case with periodic and open boundaries, appears to be missing.

For concreteness, we focus in the following on the model with two types of particles, where each site is either empty or occupied by a particle of first or second type. Particles of first and second type can move forward on an empty site, and a particle of first type followed by a particle of second type can exchange their positions (in this order only, as if particles of the first type see particles of second type as empty sites). All the hopping rates are usually take equal (in which case the particles of second type are rather called second class), but Bethe ansatz integrability [506, 552] also holds for the more general hopping rates shown in figure 25. TASEP with two species of particles is equivalent to the AHR model [553], where particles of second type are interpreted as empty sites and empty sites as particles moving in the opposite direction.

An important point, see e.g. [519], is that the presence of even a single particle of second type has macroscopic effects on height fluctuations for the particles of first type. Additionally, at the level of hydrodynamics, a second class particle follows the characteristics of Burgers' deterministic equation [554,555], and in particular sticks to shocks, a property which has been used to study their microscopic structure [519,556–559]. Conversely, starting at an anti-shock, the particle of second class selects at random one of the infinitely many characteristics issued from the anti-shock and follows it asymptotically with constant random velocity [560–566].

On the infinite line, the deterministic hydrodynamic evolution of the density fields for particles of first and second type has been studied, see e.g. [545, 567–569]. Exact integral formulas were obtained for time-dependent microscopic probabilities [570–574], and GUE Tracy-Widom distributions were found for current fluctuations [575,576], in agreement with the predictions [577] from non-linear fluctuating hydrodynamics for the KPZ modes. Tracy-Widom distributions also describe the fluctuations of a single particle of second type [559,578] and stopping times for multispecies TASEP on the interval with all particles of distinct types [579,580].

In finite volume, the evolution of the system is encoded into the generator $M(g_1, g_2)$ with $g_1$ and $g_2$ respectively conjugate to the height functions $H_i^{(1)}$ and $H_i^{(2)}$ corresponding to the identification of particles of second type respectively with particles of first type and holes, see figure 25. The Bethe ansatz integrability of the model with a single particle of second type and periodic boundaries was obtained in [552]. Generalization to an arbitrary number $N_1$ and $N_2$ of particles of first and second type [506,581] requires the nested Bethe ansatz framework, which is an iterative construction of the eigenvectors of $M(g_1, g_2)$ featuring two different kinds of Bethe roots, $y_j$ and $w_i$, which are solution to two coupled equations. For equal hopping rates and a specific choice of nesting order, the Bethe equations read

$$
\begin{aligned}
B\,(1 - y_j)^L + (-1)^{N_1 + N_2}\, y_j^{N_2} \prod_{\ell=1}^{N_1} (y_j - w_\ell) = 0 \qquad & j = 1, \ldots, N_1 + N_2 \\
C \prod_{k=1}^{N_1 + N_2} (w_i - y_k) + (-1)^{N_2}\, w_i^{N_1} = 0 \qquad\qquad & i = 1, \ldots, N_1
\end{aligned}
\quad , \qquad (41)
$$

where $B = g_2 \prod_{k=1}^{N_1 + N_2} y_k$ and $C = g_1 \prod_{\ell=1}^{N_1} w_\ell / \prod_{k=1}^{N_1 + N_2} y_k$. The iterative nature of nested Bethe ansatz manifests itself in the fact that the first equation with the $w_i = 0$ is the same as for TASEP with a single species, compare with (31), while the second equation corresponds to TASEP with $N_1$ particles on $N_1 + N_2$ sites and the $y_k$ play the role of site inhomogeneities preserving the integrability of the model (but not the positivity of hopping rates).

An inclusion property for the spectrum when varying the number of species [582] has allowed to confirm a numerical prediction [583] that the spectral gap behaves according to KPZ scaling, with dynamical exponent $z = 3/2$. The average value of the current of each type of particle in the stationary state is also known [506] from Bethe ansatz, but higher cumulants of the currents are still missing, except for the case of a single particle of second type [552,584].

No exact result is available so far for the full dynamics in finite volume under KPZ scaling $t \sim L^{3/2}$. From the Bethe equations (41) of the model with two types of particles, extending the Riemann surface approach of section 3.2 to the joint statistics of the height functions $H_i^{(1)}$ and $H_i^{(2)}$ appears to involve naturally a two dimensional complex manifold, parametrized by the two complex numbers $B$ and $C$, which is a much more complicated object than a Riemann surface. The restriction to the statistics of a single quantity gives an additional relation between $B$ and $C$, for instance $g_1 g_2 = 1$ for the current of the particles of second type $H_i^{(2)} - H_i^{(1)}$, so that the relevant manifold is again a Riemann surface, but this comes at the price of more interdependence between the two sets of Bethe roots.

### 3.3.3 Crossover EW-KPZ

The $\lambda$-dependent crossover between the EW and KPZ fixed points in finite volume can be studied from ASEP with hopping rates $1$ and $q$ under weakly asymmetric scaling $1 - q \sim L^{-1/2}$, see section 2.2.1, or using the replica solution of the KPZ equation and its connection to an integrable Bose gas, see section 2.2.4. We restrict here to the simplest case with periodic boundary condition, and focus mainly on ASEP with $N$ particles on $L$ sites. The replica approach is mentioned at the end of the section.

The current of particles between two neighbouring for ASEP is defined as $Q_t = Q_+ - Q_-$ (it is closely related to the activity $A_t = Q_+ + Q_-$ [585]), with $Q_+$ (respectively $Q_-$) the number of particles that have moved forward (resp. backward) between the two sites. Like for TASEP, the associated deformed generator $M(g)$ is integrable. The eigenstates are again given from Bethe ansatz as linear combinations of plane waves, with momenta $k_j$ related as $e^{ik_j/L} = g^{1/L} \frac{1-y_j}{1-qy_j}$ to Bethe roots $y_j$ solution of the Bethe equations

$$g \left( \frac{1-y_j}{1-qy_j} \right)^L + (-1)^N \prod_{k=1}^{N} \frac{y_j - qy_k}{y_k - qy_j} = 0 \,. \tag{42}$$

As for TASEP, while Bethe roots are generically distinct and finite, some care is needed for specific values of $g$ with coinciding or divergent Bethe roots, see e.g. [586]. Attempts to prove completeness of the Bethe equations (i.e. that any suitable solution does correspond to an eigenstate of $M(g)$) were made in [587–589], see also [590] for a rigorous approach to ASEP with periodic boundaries that bypasses the issue by writing the time-dependent probabilities of microstates in terms of multiple contour integrals.

Symmetric rational functions of Bethe roots can again be understood as meromorphic functions on a Riemann surface $\mathcal{R}_q$ corresponding to $M(g)$, whose local behaviour follows directly from the Bethe equations. Unlike for TASEP, the absence of a nice parametrization of $\mathcal{R}_q$ makes it difficult to study its global properties. Numerics suggest that $\mathcal{R}_q$ still has a single connected component if and only if $L$ and $N$ are co-prime, and a conjecture for the location of the poles of the normalization of Bethe states leads through the Riemann-Hurwitz formula (24) to a conjecture for the genus in that case [442]. Additionally, the Gallavotti-Cohen symmetry [591, 592], which relates the probabilities that the current $Q_t$ is equal to $Q$ and $-Q$ at the level of stationary large deviations, corresponds to the symmetry $g \leftrightarrow q^L/g$ of the Bethe equations, and is thus identified as an automorphism of $\mathcal{R}_q$.

The eigenvalue of $M(g)$ corresponding to a solution of the Bethe equations (42) is equal to $\mu = (1-q) \sum_{j=1}^{N} (\frac{1}{1-y_j} - \frac{1}{1-qy_j})$. Explicit expressions in terms of the $y_j$ are also known for the left and right eigenvectors, their scalar product and various overlaps, see e.g. [593]. As for TASEP, simpler finite size expressions are available for stationary, flat and domain wall initial conditions. Another good candidate for exact calculations is the random initial condition where a configuration with particles at positions $x_j$ has a weight proportional to $q^{-\sum_{j=1}^{N}(j+x_j)}$, which converges to domain wall initial condition in the limit $q \to 0$, and for which the Gallavotti-Cohen symmetry for the current holds at any finite time [442].

According to the definition (6) of the KPZ fixed point from ASEP, it is expected that large $L$ asymptotics for ASEP with finite $q \neq 1$ of overlaps of Bethe vectors contributing to KPZ fluctuations in finite volume should be the same as for TASEP. In the case of the spectral gaps, this was derived in [388, 391] by showing that the Bethe roots of ASEP actually accumulate on the same curve $\mathcal{Y}_\rho$ from figure 16 as those for TASEP, and by computing the required subleading terms. Such a derivation is however missing for the various overlaps needed to recover the KPZ fixed point in finite volume at large $L$, which have complicated expressions involving large determinants. However, the fact that symmetric functions of Bethe roots generally have

clean expansions in powers of $L^{-1/2}$ allows to extract high precision numerical values for the large $L$ asymptotic rather easily using Richardson extrapolation [594–596]. A perfect match was found in [593] with the expected results for the KPZ fixed point discussed in section 2.4.4.

An alternative way to look at the Bethe equations of ASEP is to consider the polynomial $Q(y) = \prod_{j=1}^{N}(y - y_j)$, in terms of which (42) is equivalent to the vanishing of the polynomial $g(1-y)^L Q(qy) + q^N(1-qy)^L Q(y/q)$ when $y$ is one of the Bethe roots $y_j$. This polynomial must then be divisible by $Q(y)$, which leads to Baxter's equation

$$T(y)Q(y) = g(1-y)^L Q(qy) + q^N(1-qy)^L Q(y/q). \tag{43}$$

The unknown polynomial $T$, of degree $L$, is interpreted in the algebraic formulation of Bethe ansatz as the eigenvalue of a commuting family of transfer matrices which contains $M(g)$. Baxter's equation puts enough constrains on the polynomials $T$ and $Q$ that its solutions are discrete and correspond to the eigenstates of $M(g)$ (for generic values of $g$ and after removing spurious solutions).

ASEP is symmetric under the combined exchange of occupied and empty sites, $p$ and $q$, $g$ and $g^{-1}$. Thus, Bethe ansatz for a given eigenstate of ASEP could also be formulated in terms of a set of $L-N$ Bethe roots $\tilde{y}_j$, associated to another polynomial $P(y) = \prod_{j=1}^{N}(y - \tilde{y}_j)$ and the same polynomial $T$. Considering Baxter's equation (43) as a discrete analogue of a linear differential equation of second order for $Q$ or $P$, the polynomials $Q$ and $P$ can be seen as a basis of two independent solutions, and their discrete Wronskian has the simple expression [375, 597]

$$\frac{g}{g-q^N}\frac{Q(y)P(y/q)}{Q(0)P(0)} - \frac{q^N}{g-q^N}\frac{Q(y/q)P(y)}{Q(0)P(0)} = (1-y)^L. \tag{44}$$

A perturbative solution for the eigenvalue of $M(g)$ near the stationary point $o \in \mathcal{R}_q$ was obtained in [375] using (44), giving the first stationary cumulants $c_k^{\mathrm{st}}(\lambda)$ of the KPZ height, see section 2.4.1. It was shown that $w(y) = \frac{1}{2}\log\left(\frac{q^N Q(y/q)P(y)}{g\,Q(y)P(y/q)}\right)$, considered as a formal power series in $\gamma = \log g$ with coefficients that are Laurent series in $y$, is solution of the closed equation $\sinh w(y) = B\,y^{-N}(1-y)^L\,e^{(Xw)(y)}$, where $B = -(g^{1/2}-q^N g^{-1/2})Q(0)P(0)/2$ and $X$ acts on arbitrary Laurent series $u(y) = \sum_{k=-\infty}^{\infty} u_k y^k$ as $(Xu)(y) = \sum_{k\neq 0} u_k \frac{1+q^{|j|}}{1-q^{|k|}} y^k$. It would be interesting to understand whether this approach can be extended to the whole Riemann surface $\mathcal{R}_q$. It is not clear however that the parameter $B$ above leads to a particularly simple branching structure for $\mathcal{R}_q$ away from the stationary point $o$.

Taking the limit $q \to 1$ of the Bethe equations (42), of Baxter's equation (43) or of the Wronskian (44) requires some care, as it depends crucially on how $1-q$ scales with the system size. Under the scaling $1-q \sim 1/L$, where a dynamical phase transition to a phase-separated state occurs, the eigenvalue of $M(g)$ with largest real part was studied perturbatively in $\log g$ in [377], see also [598] for non-perturbative effects in $\log g$, [376] for a related work at $q = 1$ directly, and [599] for a numerical study of the eigenvectors. A notable feature of the scaling $1-q \simeq \nu/L$ is that at all order in perturbation in $\log g$, that eigenvalue depends on $\nu$ and $g$ only through the combination $(\nu+\log g)\log g$, which is manifestly invariant by the Gallavotti-Cohen symmetry. This is no longer the case on the weakly asymmetric scaling $1-q \sim L^{-1/2}$ corresponding to KPZ fluctuations with finite $\lambda$, where $\lambda$ and $1-q$ are related by (7).

The weakly asymmetric scaling $1-q \sim L^{-1/2}$ was studied for the smallest spectral gap at $g = 1$ in [600]. By considering the variable $y$ in (43), (44) at a distance of order $L^{-1/2}$ of the singular point $-\frac{\rho}{1-\rho}$ of the curve $\mathcal{Y}_\rho$ in figure 16 on which Bethe roots accumulate, Baxter's equation leads to $\mathcal{T}_\lambda(x)\mathcal{Q}_\lambda(x) = e^{2x^2}\mathcal{Q}_\lambda(x+\lambda)+e^{2(x+\lambda)^2}\mathcal{Q}_\lambda(x-\lambda)$ and the Wronskian identity to $\mathcal{Q}_\lambda(x)\mathcal{P}_\lambda(x-\lambda)-\mathcal{Q}_\lambda(x-\lambda)\mathcal{P}_\lambda(x) = \mathcal{C}_\lambda\,e^{2x^2}$. The rescaled functions $\mathcal{T}_\lambda$, $\mathcal{Q}_\lambda$ and $\mathcal{P}_\lambda$ are no longer polynomials, and the equations above must be supplemented by some analyticity

conditions. For the smallest spectral gap, the quantization condition $\mathcal{Q}_\lambda(x) \simeq 1 + \frac{i\pi}{2\lambda x}$ when $x \to -\infty$ then leads to a systematic expansion near the EW fixed point. When $\lambda \to 0$, one has in particular $\mathcal{Q}_\lambda(x) \simeq \frac{(2\pi)^{3/2}}{4i\lambda} e^{2x^2} \text{erfc}(-x\sqrt{2})$, which means that the Bethe roots $y_j$ in the vicinity of the singular point of $\mathcal{Y}_\rho$ converge after rescaling by $L^{1/2}$ to the complex zeroes of $\text{erfc}(-x\sqrt{2})$ when $\lambda \to 0$. Interestingly, the zeroes of truncated Taylor series of analytic functions, which accumulate quite generally on singular curves similar to $\mathcal{Y}_\rho$, also converge close to the singular point to the zeroes of the complementary error function [601, 602].

It would be interesting to extend the approach of [600] to higher eigenstates with arbitrary $\log g \sim L^{-1/2}$ in order to try to understand the branching structure of the eigenstates $e_n(s; \lambda)$ from section 2.4.8. This would hopefully give some information on the elusive Riemann surface $\mathcal{R}_{\text{KPZ}}^\lambda$ describing KPZ fluctuations with finite $\lambda$, obtained as the WASEP scaling limit of $\mathcal{R}_q$ and generalizing the Riemann surface $\mathcal{R}_{\text{KPZ}}$ for polylogarithms with half-integer index. Additionally, more sophisticated approaches could be useful to understand better $\mathcal{R}_q$ and its WASEP scaling limit, for instance the ODE/IM correspondence [603, 604] between ordinary differential equations and integrable models, which connects Baxter's equation to a generalized Sturm-Liouville problem, or K-theory [605], from which the Bethe equations and Baxter's operator of the XXZ spin chain with twisted boundary condition, closely related to ASEP, naturally appear.

We end this section by a small discussion of the Bethe ansatz solution for the delta-Bose gas used in the replica approach to KPZ fluctuations, see section 2.2.4. The Hamiltonian $H_{n,\lambda}$ in (9) is known to be integrable by Bethe ansatz [225]. For periodic boundaries $x = x + 1$, setting $c = 2\lambda^2$, the Bethe equations are $e^{2q_j} = \prod_{k \neq j} \frac{q_j - q_k + c}{q_j - q_k - c}$, $j = 1, \ldots, n$, corresponding to the eigenvalue $-2\sum_{j=1}^n q_j^2$ for $H_{n,\lambda}$. The Bethe equations are not directly suitable for studying the generating function $\langle e^{sh_\lambda(x,t)} \rangle$ of KPZ fluctuations since $s = -2\lambda n$ only takes discrete values. The Riemann surface interpretation is in particular quite obscure at this level, since it would require analytic continuation in the variable $n$. A proper implementation of the replica trick circumventing the issue was worked out in [226, 227], by noting that the quantity $A(u) = e^{\frac{c}{4}(u^2-1)} \sum_{j=1}^N e^{(u-1)q_j} \prod_{k \neq j} \frac{q_j - q_k + c}{q_j - q_k}$ verifies the integral equation [12]

$$A(u+1) - A(u-1) = c \int_0^u dv \, e^{\frac{cv(u-v)}{2}} A(v+1)A(u-v-1). \tag{45}$$

This equation can then be expanded in powers of $c$, with coefficients that are polynomial in $n$, so that an expansion in powers of $n$ can finally be obtained. This procedure allows to treat $n$ as a continuous variable, and led to exact expressions for stationary large deviations, see section 2.4.1. It would be very interesting to understand the branching structure of the integral equation (45) and its consequences for the Riemann surface $\mathcal{R}_{\text{KPZ}}^\lambda$. It would also be nice to understand the relationship between (45) and the WASEP limit of Baxter's equation.

# 4  Conclusions

We have summarized in these lecture notes various known facts about KPZ fluctuations in one dimension, with a special focus on finite volume effects, where a key object in the exact description of height fluctuations is a Riemann surface of infinite genus. This implies in particular that fluctuations at a finite time can to some extent be reconstructed by analytic continuation from the knowledge of how stationary large deviations are approached at late times.

---

[12] This small modification of the integral equation in [226, 227] is valid for all the eigenstates of $H_{n,\lambda}$.

One point we would like to emphasize again is the universality of KPZ fluctuations in finite volume, for the whole evolution in time from an arbitrary initial state to a non-equilibrium stationary state. During this relaxation process, fluctuations interpolate between those for the system on the infinite line, when the finite spatial extension of the system is not felt, and a stationary process depending only on boundary conditions at late times.

Several exact results have already been obtained at the KPZ fixed point in finite volume, especially with periodic boundary condition. Many interesting questions still remain, in particular with regard to other types of boundary conditions or when several conservation laws are present. Describing the full dynamics of the universal $\lambda$-dependent renormalization group flow between the EW and KPZ fixed points is a major open problem.

## Acknowledgements

I am grateful to K. Mallick and H. Spohn for introducing me to exclusion processes and KPZ fluctuations. It is also a pleasure to thank the members of the committee for my habilitation thesis, on which these lecture notes are based, L. Canet, V. Terras, F. van Wijland, S. Cohen and P. Pujol.

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
