# Peer review of "KPZ fluctuations in finite volume"

_SciPost Physics Lecture Notes_

## Round 1 · Referee Report · Anonymous (Referee 1) · 2024-3-7

Report
This is a great review on mathematical physics approaches to characteristic fluctuation properties of the 1D KPZ class. Compared with the existing reviews on this topic, distinctive features of the submitted review lie in its introductory descriptions (in particular Sec. 2.1) intended for physicists unfamiliar with this subject, and its comprehensive coverage of the latest developments in the field. As a review, it puts a particular focus on exact results for finite-size systems as well as the Riemannian surface approach, to which the author has made significant contributions. This review will be part of useful literature of the field. As such, I recommend publication of the submitted manuscript, after the author considers the following remarks and makes changes that the author agrees improve the manuscript.
-
In my opinion, the author could give an explicit form of the KPZ equation much earlier in the introduction, before its nonlinearity parameter $\lambda$ is discussed; otherwise Sec.1 is too hard to follow for layman readers. This is a pity because Sec. 2.1 is much more approachable for those readers.
-
About the Family-Vicsek scaling described in the beginning of Sec. 2.1.2, I think it is useful to remark that $w$ is used as a quantifier of $\ell_\perp$, and that any other possible choices work as well. For example, in the literature, the height difference correlation function has also been used.
-
Cole-Hops -> Cole-Hopf (p.10)
-
It seems to me that the following sentence in p.14 is broken: "Beside shocks, anti-shocks, ..., but only unphysical solutions (also called weak solutions) that do not describe the density $\rho(x,\tau)$ of ASEP, but are unstable and decay instantaneously..."
-
I was a bit confused by the following description in p.15: "Since KPZ fluctuations are only seen for TASEP in a reference frame moving at velocity $1-2\rho$, they do not exist in the low and high density phases." The first part of the sentence suggests that KPZ fluctuations do exist, when observed in the moving frame (the current needs to be integrated at a fixed position in the moving frame then). What does the author mean by "do not exist" here?
-
Whereas the author is successful in not overburdening the review with equations, in some places, equations would help clarify what the author means. Such places include "a functional integral over the realizations..." (p.17) and "a variational formula for general initial condition" (p.20).
-
About Fig.10, I think it's helpful to show the color ordering for the two bottom panels, to show in what direction $t$ increases.
I thank the referee for her/his report. Here is a reply to the points mentioned in the report:
1 - agreed, I will write down the KPZ equation in the introduction.
2 - agreed, I will mention this.
3 - corrected.
4 - Yes, that sentence was not that clear. I will rewrite it.
5 - The sentence was a bit unclear, I will improve. What I meant is that KPZ fluctuations in finite volume, which require times of order $L^{3/2}$, also need a spatial extension of order $L^{3/2}$ so that the moving frame $i = (1-2\rho)t+xL$ makes sense for the system with open boundaries (when $\rho\neq1/2$, i.e. in the high and low density phases). This is the main difference with the model with periodic boundaries, where particles just wrap around the system and there is no problem working in a moving frame at any velocity. On the other hand, with open boundaries, KPZ fluctuations in the early time regime (where the correlation length is much smaller that the system size) can indeed be observed even for boundary rates in the low and high density phases, by taking a time $t$ much smaller than $L$ but still very large, considering the moving frame with velocity $1-2\rho$ where $\rho$ is the local density, and staying away from the boundaries.
6 - I agree that it would be nice to write down explicit formulas in the two cases mentioned. Unfortunately, this would require in both cases to set up somewhat heavy notations. Since these formulas are not used in the following, are only tangentially related to my main point about KPZ fluctuations in finite volume, and can be found in the references, I would rather leave them out.
7 - I will add an indication for the ordering of times in the caption of the figure.

Author: Sylvain Prolhac on 2024-05-02 [id 4466]
(in reply to Report 2 on 2024-04-12)I thank the referee for her/his report. Here is a reply to the points mentioned in the report:
1 - yes, the sentence was not so clear, I will rewrite.
2 - corrected.
3 - corrected.
4 - the height increments are correct as stated. In figure 2, with $N=3$ and $L=9$, the increments $N/L=+1/3$ and $-(1-N/L)=-2/3$ are consistent with the picture (the downward slopes have twice the height of the upward slopes), and the sum from $i=1$ to $L$ of the increments is indeed equal to 0 (the common term $+N/L$ contributes $N$ to the sum, while the terms -1 for all the sites with a particle sum to $-N$). The height increments $L/(N−L)$ and $−L/N$ proposed by the referee, although also consistent with periodicity, would presumably be harder to treat analytically.
5 - agreed, I will correct this.
6 - both are equivalent on the Euler scale. I will reformulated to make the point clearer.
7 - I agree, I will remove the mention of the Riemann surface $R_{KPZ}$ in section 2.4.3, but keep and expand the discussion below (21) to mention how this is connected to analytic continuation between the eigenstates.
8 - corrected.
9 - corrected.
10 - footnote 11 has been rewritten and expanded, and figure 12 has been added to explain visually the difference between singular and exceptional points (or more generally ramified points with respect to some parameter). At an exceptional point, eigenstates "disappear" in the sense that several eigenstates correspond to the very same point on the Riemann surface. At singular points, on the other hand, the desingularization of the algebraic curve splits the singular point into several distinct points on the Riemann surface, preserving the total number of eigenstates.

---

## Round 1 · Referee Report · Anonymous (Referee 2) · 2024-4-12

Strengths
Report
The manuscript "KPZ fluctuations in finite volume" is a review of the state of the art in the field of exactly solvable models of KPZ class with emphasis on the systems in finite volume. The Author starts with gentle introduction into the field of KPZ universality class in one dimension, draws its large scale picture, and then specializes to his own results, which in fact constitute a significant part of the present development of the subject. My impression about this work is very positive. It is an exceptional, if not to say the most complete review on the topic to date, which contains an extensive list of very well systematized formulas and exhaustive collection of references. It will for sure be my table handbook from now on. The part about the Riemann surface approach is a bit difficult to read as it requires at least superficial acquaintance with the theory of Riemann surfaces and algebraic geometry, which, I suppose, the most of the target community do not possess. This, however, is the price for the chosen presentation format that gives the advantage of brevity over explanation of details. Probably, this is the best of what the Author could do in this way. In need of deeper understanding the Reader may consult with the original publications, the references to which are given in the text. To summarize, I highly recommend the manuscript for publication and believe that it will become one of the key sources on the subject. There are a few little typos I found and comments I have listed below.The Author may address them before publication.
1)Page 8, the sentence after formula (3) "... (respectively the infinite line $x\in \mathbb{R}$ and the half-line $c\in \mathbb{R}^+$) with periodic and open boundaries fo $h(x,t)$)..." is confusing. Probably a comma or a conjunction before words "with periodic.." should be added.
2)Page 10, the second paragraph from below: "coincide" ->"coincides".
3)Page 11, the last paragraph, line 8 from below: "...leads in particular for the average height..." -> "...leads in particular to the average height...".
4)Page 13, second paragraph: the height increments $N/L$ and $-(1-N/L)$ seem to be consistent nor with explanation neither with the fig. 2. If we assume that a particle and a hole correspond to the height decrease and increase respectively, as they do in the figure, the height increments should be $-L/N$ and $L/(N-L)$ to be consistent with periodic boundary conditions.
5)Page 13, third paragraph: the traveling wave velocity is $(1-q)(1-2\rho)$ rather than $(1-2\rho)$. It becomes $(1-2\rho)$ in the next paragraph after the time rescaling. However, we are still in the original time scale in the third paragraph, which in particular means that the velocity vanishes at $q=1$.
6)page 14, line 2 of third paragraph: $L^{2/3}\gg L$ -> $t^{2/3}\gg L$.
7) page 28: here and in a few places below the Author refers to the Riemann surface $\mathcal{R}_{KPZ}$ before introducing it or even describing it in any way. In my opinion this is not very informative, but can be confusing for the Reader. I would try to keep from using it until it is really necessary, or at least try to give some rough explanation of what happens.
8)page 31, the last paragraph, line four from below: where -> were.
9) page 32, line 2: was->were
10) Page 37: I did not get a clear understanding of the difference between singular and exceptional points from the text, except that the latter are associated with Jordan cells and the former with generic spectrum degeneracy. Why the latter are ramification points and the former are singular points? It seems that the characteristic polynomial does not distinguish between theses two cases. Then, what is the difference in terms of the spectral curve?
Recommendation
Publish (surpasses expectations and criteria for this Journal; among top 10%)

---

## Round 3 · List of Changes

Changes in response to referee 1:
1 - the KPZ equation has been written already in the introduction.
2 - the fact that other estimators for the width of the interface do verify the Family-Vicsek scaling is now mentioned.
3 - the misprint "Cole-Hops" has been corrected.
4 - the sentence about anti-shocks was too convoluted. It has been rewritten.
5 - for TASEP with open boundaries, it should now be clearer why the low and high density phases must be excluded, on the time scale $t\sim L^{3/2}$.
6 - I agree with the referee that writing down explicit formulas here would be nice, but this would require to set up somewhat heavy notations. Since these formulas are not used in the following, are only tangentially related to my main point about KPZ fluctuations in finite volume, and can be found in the references, I would rather leave them out.
7 - an indication for the ordering of times has been added in the caption of the figure.
Changes in response to referee 2:
1 - the sentence was not so clear, I have rewritten.
2 - misprint corrected.
3 - formulation improved.
4 - none, the height increments are correct as stated.
5 - misprint corrected.
6 - the formulation has been improved.
7 - the mention of the Riemann surface $R_{KPZ}$ has been removed from section 2.4.3, but kept and expanded in the discussion below (21). It is now mentioned there how this is connected to analytic continuation between the eigenstates.
8 - misprint corrected.
9 - misprint corrected.
10 - footnote 11 has been rewritten and expanded, and figure 12 has been added to explain visually the difference between singular and exceptional points (or more generally ramified points with respect to some parameter).
Other changes:
- page 9: reference [71], where finite volume effects are discussed for condensates out of equilibrium, has been added.
- page 17: reference [227] has been updated (the discrepancy with the replica result has been resolved in the newer version) and moved to section 2.4.1, as [372].
- reference [461] has been added.
- figures 18 and 19 have been corrected (problem with the LaTeX interpreter)

---

## Editorial Decision

editorial_decision: